# An Axiomatic Atlas for Optimization

## Abstract

First-order methods are the workhorses of modern large-scale optimization, powering training and inference across machine learning, signal processing, and scientific computing. Yet the theoretical guarantees that explain their behavior are dispersed across smooth, nonsmooth, stochastic, and composite settings, while practitioners must choose among many algorithmic variants and tune interacting hyperparameters with limited guidance about which assumptions actually matter. We introduce the *Optimization Atlas*, an axiomatic view that organizes widely used first-order methods and their canonical convergence behaviors within a single, explicit assumption space. The atlas exposes inclusion-minimal assumption sets that suffice for a desired outcome, and it delineates sharp frontiers where a single assumption change alters the attainable regime, such as sublinear versus linear convergence or linear convergence versus variance-limited floors. We then leverage the induced theorem by axiom structure to uncover a small number of recurring modes that clarify which phenomena are structurally shared across methods and which correspond to genuinely distinct mechanisms. Finally, we convert the atlas into a practical diagnostic control plane: from short training traces it estimates the active limiting ceiling and ranks interventions, ranging from relaxing modeling assumptions (for example via smoothing or regularization) to increasing algorithmic capacity (for example via batching or variance reduction). Experiments on controlled synthetic problems and a CIFAR-10 convolutional network show that this control plane reliably identifies the governing regime, recommends high-return changes when optimization is the bottleneck, and abstains when additional tuning is unlikely to help.

## 1 Introduction

First-order methods are the main workhorses of modern optimization. They solve problems of the form

$$\min_{x \in \mathbb{R}^d} F(x),$$

where $F$ can be a loss function, a regularized empirical risk, or an objective from inverse problems, signal processing, control, or scientific computing. These methods are popular because each iteration is cheap: it typically needs only a gradient (or subgradient) and, when needed, a simple proximal step. This makes first-order methods practical for large-scale and high-dimensional problems where higher-order methods are often too expensive (Boyd and Vandenberghe, 2004; Nesterov, 2004; Bottou et al., 2018). As a result, gradient descent, stochastic gradient descent, accelerated methods, and proximal splitting algorithms are widely used across many applications (Robbins and Monro, 1951; Nesterov, 1983; Beck and Teboulle, 2009; Parikh and Boyd, 2014).

Over several decades, many first-order methods have been developed. Their convergence guarantees depend on structural assumptions about $F$, such as smoothness, convexity, strong convexity, the Polyak-Łojasiewicz condition, error bounds, and Kurdyka-Łojasiewicz inequalities (Nesterov, 2004; Karimi et al., 2016; Kurdyka, 1998; Bolte et al., 2007; Attouch et al., 2013). At the same time, practical needs have pushed the field in several directions: composite objectives and constraints motivate proximal splitting (Parikh and Boyd, 2014); data-driven learning motivates stochastic and variance-reduced methods (Bottou et al., 2018; Johnson and Zhang, 2013; Nguyen et al., 2017); and large-scale training motivates adaptive and momentum-based updates (Duchi et al., 2011; Kingma and Ba, 2015).

In practice, however, this large toolbox creates a persistent engineering bottleneck. Most methods are presented as general templates with several tunable choices: step sizes (and schedules), momentum

parameters, batch sizes, inner-loop lengths, proximal weights, and stopping criteria. Different methods expose different subsets of these controls. Finding a reliable configuration for a specific problem is often expensive and fragile. The common approach is to treat the optimizer as a black box and tune hyperparameters using grid search, random search, Bayesian optimization, or early-stopping strategies (Bergstra and Bengio, 2012; Snoek et al., 2012; Shahriari et al., 2016; Li et al., 2017; Falkner et al., 2018; Jaderberg et al., 2017). These methods can work well, but they scale poorly as the number of tunable parameters grows, and they do not directly use the structural information that optimization theory already provides.

This paper proposes a complementary approach: we turn that structural theory into a clear navigation tool. Inspired by an axiomatic design view (Suh, 1990), we organize first-order optimization around a small set of assumptions that repeatedly appear in convergence guarantees. We introduce nine axioms that cover regimes commonly encountered in practice: 1) convexity, 2) smoothness, 3) strong convexity, 4) Polyak-Łojasiewicz geometry, 5) error-bound conditions, 6) quadratic-growth conditions, 7) Lipschitz continuity (for nonsmooth analysis), 8) bounded-variance stochastic gradients, and 9) composite proximal structure. Any subset of axioms defines a *world* in an abstract assumption space. A method, theorem, or rate applies exactly in the worlds where its assumptions hold. We call the resulting map the *Axiomatic Atlas*. The atlas makes it explicit which assumptions drive which behaviors, shows when different assumption sets lead to the same type of guarantee, and identifies *frontiers* where adding or removing a single axiom changes the best achievable rate. Because the atlas is indexed by assumptions (not by method names), it also highlights assumption combinations that are common in applications but are less visible in standard guarantee tables. This can point to places where sharper analyses or new algorithms are still needed.

From a practitioner's standpoint, the atlas also provides a structured way to think about hyperparameters. Many tunable choices correspond to a small number of structural bottlenecks, and these bottlenecks line up with the axioms. For example, smoothness controls stable step sizes and whether acceleration is appropriate (Nesterov, 2004; 1983), and a Polyak-Łojasiewicz or strong-convexity condition controls whether linear convergence is a realistic. The atlas supports a concrete workflow: i) *diagnose* which axioms plausibly hold along the region explored by the iterates, ii) *select* an inclusion-minimal assumption set that implies the desired regime, and then iii) *tune* only the parameters that matter along the relevant frontier. This turns hyperparameter tuning from a broad search over many degrees of freedom into a targeted choice among a few interventions, including step-size selection, batching, variance reduction, smoothing, regularization, or preconditioning. We formalize this regime-limited viewpoint with an *optimization roofline*, in the spirit of roofline models used to reason about performance ceilings in computer architecture (Williams et al., 2009).

We instantiate the atlas as follows. First, we prove that the nine axioms are logically independent by giving separating counterexamples. Next, we show a form of completeness: many standard first-order guarantees can be derived using only subsets of these axioms. We compile 35 core results spanning gradient descent, accelerated gradient methods, proximal gradient descent, stochastic gradient descent, and variance-reduced methods (Nesterov, 2004; Beck and Teboulle, 2009; Johnson and Zhang, 2013; Nguyen et al., 2017). We then build a reduction matrix that records which axioms each theorem needs, enumerate the inclusion-minimal sufficient assumption sets, and compress the resulting $2^9$ worlds into a smaller set of equivalence classes connected by single-axiom frontiers. Finally, we validate these predicted frontiers experimentally by constructing objectives that differ by exactly one axiom and measuring the resulting regime changes under matched gradient-evaluation budgets.

## 2 Preliminaries

The Optimization Assumption Atlas is built from a curated library of standard first-order optimization results: descent inequalities, convergence rates, and matching oracle lower bounds. In total, we compile *35 core theorems* (plus a small number of auxiliary facts) spanning gradient descent, accelerated methods, proximal-gradient and splitting schemes, stochastic gradient methods, and variance-reduced algorithms. We organize these results using the standard smoothness-convexity lens, and we further separate deterministic from stochastic oracles and smooth from nonsmooth objectives.

Concretely, the library is grouped into the following families. In the smooth convex setting, we include the basic descent machinery, sublinear rates for gradient descent, accelerated $\mathcal{O}(1/k^2)$ rates (where $k$ denotes the optimization iteration), and the corresponding smooth-convex lower bound. For linear-convergence regimes, we include guarantees under strong convexity and under PL-type / growth conditions, along

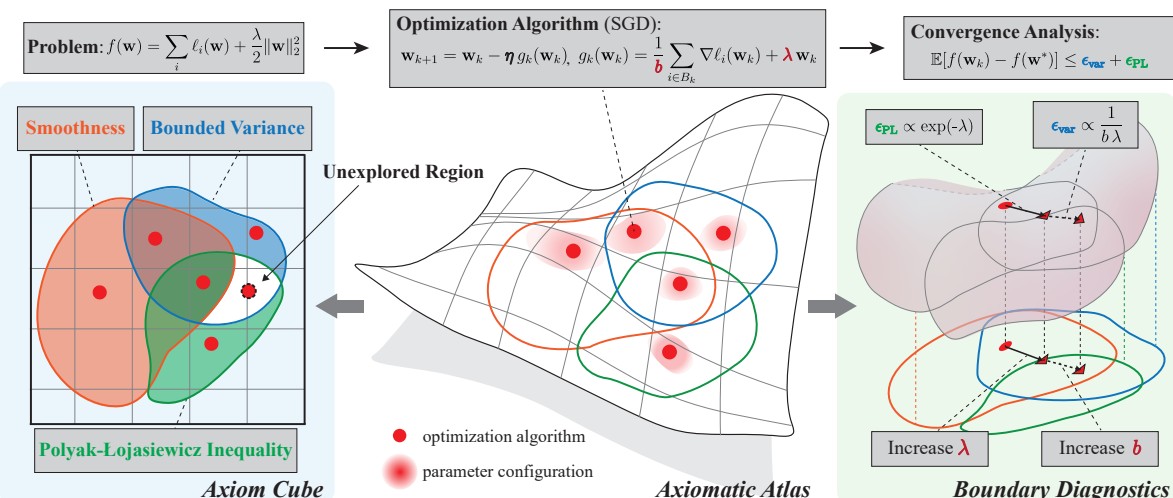

Figure 1: Illustration of the axiomatic atlas. Axioms define a world in assumption space (left), algorithms and hyperparameters occupy locations in that space (middle), and Boundary Diagnostics suggests which knob to turn when progress stalls (right), e.g., increasing regularization $\lambda$ vs. increasing batch size $b$.

with results that relate these geometric assumptions. For nonsmooth Lipschitz convex problems, we include the $\Theta(1/\sqrt{k})$ rate and the matching lower bound, which rules out Nesterov-style acceleration without smoothness. For stochastic settings, we include the transient-versus-noise-floor behavior of SGD under constant stepsizes, the effect of decreasing stepsizes, minibatching laws, and information-theoretic noise limitations; we also include representative linear-rate results achievable by variance reduction in finite-sum problems under suitable geometry. Finally, we include representative results for composite objectives that motivate proximal operators, as well as geometry/constraint statements (e.g., error-bound and KL-type implications, and projection-driven guarantees) that explain local linear phases and restart behavior. Each result in the full catalog is annotated with a *minimal assumption tag*: the smallest set of assumptions (expressed in the axiom vocabulary defined later in Section 3) under which the claim holds. To keep the main text readable, we defer the complete formal statements (with minimal tags and proof references) to Appendix A. The complete dependency matrix is given in Section 4.

We use the following notation throughout. The dimension is $d$. We write $f$ (or $F$) for the objective, and $f^\star$ (or $F^\star$) for the optimal value; $x^\star$ denotes a minimizer and $\mathcal{X}^\star$ the set of minimizers. When we assume smoothness, $L$ denotes the Lipschitz constant of the gradient. When we assume strong curvature, $\mu$ denotes the strong-convexity constant or the PL constant (depending on the setting). We write $\eta$ (or $\eta_k$) for the stepsize. In stochastic settings, $\sigma^2$ denotes a bound on the gradient-noise variance, and $\mathbb{E}[\cdot]$ and $\text{Var}[\cdot]$ denote expectation and variance. For composite objectives, we write $F = g + h$ with smooth $g$ and proximable $h$, and we use $\text{prox}_{\eta h}$ for the proximal operator.

## 3 Axioms

This section proposes the axiomatic atlas and the roofline framework employed throughout our convergence proofs and diagnostics. Each axiom is a binary predicate on a problem class, designed to isolate a single geometric or statistical property that is both prevalent in applications and actionable in algorithm design. We associate each axiom with exactly one ROOFLINE TAG: CURVATURE, NONSMOOTH, NOISE, or COMPOSITE. These tags identify the mechanism that limits first-order progress in a given regime. CURVATURE governs deterministic rates through the interaction of smoothness with convexity or growth conditions; NONSMOOTH sets the subgradient-method ceiling; NOISE introduces variance floors; and COMPOSITE determines when a proximal step is the appropriate primitive. For a given world, the roofline is the upper envelope induced by its active tags: rates on the envelope are tight within our theory, rates below are achievable by standard methods, and rates above require assumptions not present in that world.

Let $F: \mathbb{R}^d \to \mathbb{R} \cup \{+\infty\}$ be proper, closed, and lower semicontinuous, and let $F^\star = \inf_x F(x)$. When $F$ is differentiable we write $f$ in place of $F$. We denote the Euclidean inner product by $\langle \cdot, \cdot \rangle$ and the induced norm by $\|\cdot\|$. The solution set is $\mathcal{X}^\star = \arg\min F$; when $\mathcal{X}^\star \neq \emptyset$ we write $\text{dist}(x, \mathcal{X}^\star) = \inf_{x^\star \in \mathcal{X}^\star} \|x - x^\star\|$. Several axioms below are *local*: they need only hold on a neighborhood of $\mathcal{X}^\star$ or, equivalently for our

purposes, on the sublevel set explored by the iterates. This locality is what makes the axioms compatible with practical workloads, and it is what the diagnostic procedures of Section 5 attempt to detect.

**Axiom A1** (Convexity). *For all $x, y \in \mathbb{R}^d$ and all $\theta \in [0, 1]$,*

$$F\big(\theta x + (1 - \theta)y\big) \ \leq \ \theta\, F(x) + (1 - \theta)\, F(y).$$

Convexity eliminates spurious local minima, making the objective gap $F(x) - F^\star$ a globally valid measure of progress. This axiom carries the CURVATURE tag: it is the minimal geometric assumption under which first-order methods attain the canonical sublinear $\mathcal{O}(1/k)$ rate, and it is the prerequisite for acceleration once smoothness (A2) is also present (Nesterov, 2004; Boyd and Vandenberghe, 2004).

**Axiom A2** (*L*-Smoothness). *$f$ is differentiable with Lipschitz-continuous gradient:*

$$\|\nabla f(x) - \nabla f(y)\| \ \leq \ L\, \|x - y\| \qquad \text{for all } x, y \in \mathbb{R}^d.$$

Equivalently, $f$ admits a global quadratic upper model, which yields the descent lemma and the step-size rule $\eta \leq 1/L$. This axiom carries the CURVATURE tag: it determines the stable deterministic regime, enabling the $\mathcal{O}(1/k)$ rate for convex problems, the accelerated $\mathcal{O}(1/k^2)$ rate when combined with A1, and linear rates when combined with either A3 or A4 (Nesterov, 2004).

**Axiom A3** ($\mu$-Strong Convexity). *For all $x, y \in \mathbb{R}^d$,*

$$f(y) \ \geq \ f(x) + \langle \nabla f(x),\, y - x \rangle + \tfrac{\mu}{2}\, \|y - x\|^2.$$

Strong convexity endows $f$ with uniform curvature: the function grows at least quadratically away from its (necessarily unique) minimizer. The condition number $\kappa = L/\mu$ then governs contraction; under A2, gradient descent achieves a linear rate with per-iteration factor $1 - 1/\kappa$. This axiom carries the CURVATURE tag (Nesterov, 2004).

**Axiom A4** (Polyak-Łojasiewicz Condition). *There exists $\mu > 0$ such that*

$$\tfrac{1}{2}\, \|\nabla f(x)\|^2 \ \geq \ \mu\, \big(f(x) - F^\star\big) \qquad \text{for all } x \in \mathbb{R}^d.$$

The Polyak-Łojasiewicz (PL) condition is a gradient-domination inequality: whenever the function value is suboptimal, the gradient norm is proportionally large. Unlike strong convexity, it does not require convexity and can therefore model benign nonconvex landscapes in which gradient descent still converges linearly. Under smoothness (A2), the PL condition yields the same qualitative deterministic rate as strong convexity, yet it interacts differently with stochastic noise (A7): constant-step-size SGD contracts to a neighborhood rather than to the optimum. This axiom carries the CURVATURE tag (Karimi et al., 2016).

**Axiom A5** (Local Error Bound / Sharpness). *There exist $\alpha > 0$, $p \in [1, 2]$, and a neighborhood $U$ of $\mathcal{X}^\star$ such that*

$$F(x) - F^\star \ \geq \ \alpha\, \mathrm{dist}(x, \mathcal{X}^\star)^p \qquad \text{for all } x \in U.$$

This axiom links function-value suboptimality to distance from the solution set, but only locally. The exponent $p$ controls the sharpness of the growth: $p{=}2$ recovers quadratic growth (A9), while $p{=}1$ models genuinely sharp minima, which often coincide with finite-identification effects in structured problems. By creating local linear-convergence windows even absent global strong convexity, this axiom carries the CURVATURE tag (Luo and Tseng, 1993).

**Axiom A6** (Lipschitz Objective / Bounded Subgradients). *$F$ is convex and $G$-Lipschitz on $\mathrm{dom}\, F$:*

$$|F(x) - F(y)| \ \leq \ G\, \|x - y\| \qquad \text{for all } x, y \in \mathrm{dom}\, F,$$

*equivalently, $\|g\| \leq G$ for every $g \in \partial F(x)$ and every $x \in \mathrm{dom}\, F$.* Lipschitz continuity is the standard regularity condition for nonsmooth convex optimization: function values change at most linearly, while subgradients remain bounded but need not vanish smoothly near the optimum. This axiom activates the NONSMOOTH tag, setting the classical $\mathcal{O}(1/\sqrt{k})$ ceiling below which smooth-sense acceleration is inapplicable (Nesterov, 2004).

**Axiom A7** (Bounded-Variance Stochastic Oracle). *Given $x$, the stochastic gradient $g(x)$ satisfies*

$$\mathbb{E}[g(x) \mid x] = \nabla f(x), \qquad \mathbb{E}\big[\|g(x) - \nabla f(x)\|^2 \mid x\big] \leq \sigma^2.$$

*Common variants include a bounded second-moment condition $\mathbb{E}[\|g(x)\|^2 \mid x] \leq B^2$ and mini-batch variance reduction $\mathrm{Var}[\bar{g}_b(x) \mid x] \leq \sigma^2/b$.* This is the baseline statistical model for SGD: each gradient estimate is unbiased but corrupted by noise of controlled variance. The axiom activates the Noise tag because it introduces an irreducible variance floor under constant step size. In smooth strongly convex or PL geometry, this converts deterministic contraction to a contraction-to-neighborhood whose radius scales as $\eta\sigma^2/b$ (Bottou et al., 2018).

**Axiom A8** (Composite Proximal Structure)**.** *$F = g + h$, where $g$ is differentiable and $L$-smooth (A2), $h$ is proper, closed, and convex, and the proximal operator*

$$\mathrm{prox}_{\eta h}(v) \;=\; \arg\min_u \Big\{ h(u) + \tfrac{1}{2\eta}\|u - v\|^2 \Big\}$$

*is efficiently evaluable for every $\eta > 0$.* Composite structure separates the smooth data-fitting term $g$ from the nonsmooth regularizer or constraint $h$. This is the natural setting for proximal gradient methods, splitting algorithms, and their accelerated variants; it encompasses, for example, smooth losses penalized by $\ell_1$ regularization or constrained to a convex set via an indicator function. The axiom carries the Composite tag because the achievable rates depend on whether progress is limited by smooth curvature in $g$ or by the nonsmooth structure of $h$ (Parikh and Boyd, 2014).

**Axiom A9** (Local Quadratic Growth)**.** *There exist $\gamma > 0$ and a neighborhood $U$ of $\mathcal{X}^\star$ such that*

$$F(x) - F^\star \;\geq\; \tfrac{\gamma}{2}\,\mathrm{dist}(x, \mathcal{X}^\star)^2 \qquad \textit{for all } x \in U.$$

Quadratic growth is the specialization of the local error bound (A5) to $p{=}2$. It models the common situation in which the objective becomes effectively strongly curved near the solution set despite lacking global strong convexity. Under smoothness and convexity, quadratic growth is equivalent to the Polyak-Łojasiewicz inequality on relevant sublevel sets, which explains why A4 and A9 frequently yield identical practical predictions regarding linear convergence. This axiom carries the Curvature tag (Karimi et al., 2016; Luo and Tseng, 1993).

The axioms are designed to be freely composed. Several well-known implications aid interpretation. Strong convexity (A3) implies both global quadratic growth and, under smoothness (A2), a Polyak–Łojasiewicz inequality with comparable constants; this explains why A3 and A4 both support linear contraction in the deterministic smooth roofline. The local error bound (A5) with $p{=}2$ recovers the local form of quadratic growth (A9), while $p{=}1$ models sharp minima that often coincide with finite-identification behavior in structured composite problems. Our diagnostics in Section 5 treat these properties as workload dependent: the relevant question is whether an axiom holds on the region explored by the iterates, not necessarily on all of $\mathbb{R}^d$. The reduction matrix in Section 4 records which axioms are sufficient for each theorem; our minimality convention throughout is *inclusion minimality*: a sufficient set is minimal if removing any single axiom invalidates at least one guarantee derived from that set.

**Consistency, independence, and coverage** The atlas uses the axioms as a shared vocabulary for stating and comparing convergence guarantees. Three basic properties make this vocabulary usable: 1) The axiom system is *consistent*: there exists at least one concrete problem (and stochastic oracle, when relevant) for which A1–A9 all hold simultaneously. 2) The axiom system is *independent* in an atlas-relative sense: for each axiom $A_i$, there is at least one theorem in our library whose recorded inclusion-minimal proof route uses $A_i$, and there exists a counterexample world where all other axioms in that route still hold but $A_i$ fails, so the theorem's conclusion is no longer guaranteed. They are what justify the minimality tags used in the ledger (Table 1). 3) the axiom vocabulary is *sufficient for the atlas*: every statement in our theorem library (T1–T35) can be proved using only the axioms marked in its row of the ledger. Full statements are collected in Appendix A and complete derivations are given in Appendix B. This is the sense in which the nine axioms form a complete basis for the results we include; we do not claim that they cover every regime encountered in large-scale nonconvex learning.

## 4 The Axiomatic Atlas: Dependency Ledger, Compression, and White Spaces

The atlas becomes operational once we record, for each theorem used in the paper, which axioms are actually needed to prove it. This turns a collection of proofs into a single explicit object that can be queried and extended. The resulting analysis serves three roles. First, it makes the logical structure of

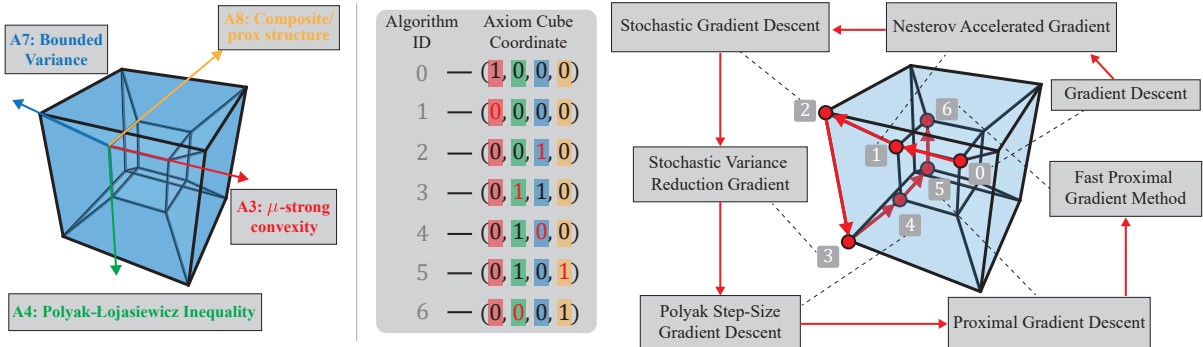

Figure 2: A four-axiom slice of the atlas. Each node is a binary assumption coordinate (left/middle), and common first-order methods are placed at the coordinates whose assumptions their guarantees use (right); moving along an edge corresponds to adding or removing one axiom.

first order theory explicit by exposing when distinct axiom bundles lead to the same guarantee. Second, it provides a compact summary of the landscape by identifying a small number of recurring assumption patterns. Third, it highlights combinations of axioms that are weakly represented in the current ledger and therefore form concrete targets when expanding the atlas.

## 4.1 A theorem axioms ledger

Table 1 is the atlas ledger. Each row corresponds to one theorem statement (T1 to T35) or to an alternative minimal route for the same statement when more than one inclusion minimal sufficient axiom set exists. Each column corresponds to an axiom (A1 to A9, defined in Section 3). A mark in column $A_k$ indicates that $A_k$ is required by the recorded proof route for that row. Minimality is verified by ablation: for each marked axiom, removing it breaks at least one step of the derivation of that theorem variant. The purpose of the ledger is not to re prove results, but to standardize them into a common format so that they can be compared, compressed, and extended systematically.

Seen through this ledger, much of the history of first-order optimization can be read as a sequence of controlled assumption relaxations. A common starting point is the smooth, convex, strongly convex setting (roughly A1+A2+A3), where textbook methods enjoy global linear convergence and clean stability guarantees (Nesterov, 2004). Dropping strong convexity while keeping smooth convexity leads to the familiar sublinear regime for gradient descent; Nesterov's acceleration shows that one can still achieve a faster $\mathcal{O}(1/k^2)$ decay after $k$ iterations under smooth convexity (Nesterov, 1983). Dropping convexity is often paired with a geometric replacement such as the Polyak–Łojasiewicz condition (A4), which restores linear convergence for a broad class of nonconvex objectives (Karimi et al., 2016). Dropping smoothness and working only with Lipschitz continuity (A6) yields the classical nonsmooth barrier: rates slow to order $1/\sqrt{k}$ after $k$ iterations, and this slowdown is fundamental in the black-box model (Nesterov, 2004).

Much of the development of first-order methods can be viewed as moving through an assumption space by turning a small number of structural knobs on and off. Figure 2 illustrates this idea on a four-axiom slice of the atlas: strong convexity (A3), the Polyak–Łojasiewicz condition (A4), bounded stochastic variance (A7), and composite/proximal structure (A8). Each highlighted red dot is a bit pattern in this 4D "axiom cube," and the dashed callouts attach a familiar algorithm name to that coordinate. As in the dependency ledger, a bit records an assumption that is actually used by a proof route, not a consequence that could be inferred indirectly from other assumptions. The red arrows show a representative chain of regime changes that mirrors common modeling relaxations and the corresponding method choices. Starting from the deterministic strongly convex regime, gradient descent sits at the coordinate where A3 is active. Dropping A3 moves to the smooth convex regime, where one typically uses Nesterov's accelerated gradient. Replacing exact gradients with a bounded-variance stochastic oracle turns on A7 and leads to stochastic gradient descent. When PL-type geometry is additionally available (turning on A4), one can recover linear behavior in finite-sum problems via variance reduction. Removing stochastic noise returns to a deterministic PL regime, where Polyak-type step sizes are natural. Turning on composite/prox structure (A8) moves to proximal gradient methods, and when PL is not available one instead relies on accelerated proximal schemes (e.g., FISTA) to obtain the best sublinear guarantees under the weaker assumptions. This figure is a small slice; the full atlas repeats the same idea over all nine axioms.

Table 1: Atlas dependency ledger. A filled mark ● in column $A_k$ indicates that axiom $A_k$ is required by the recorded inclusion minimal proof route for the theorem variant in that row.

| Theorem variant | A1 | A2 | A3 | A4 | A5 | A6 | A7 | A8 | A9 |
|---|---|---|---|---|---|---|---|---|---|
| **A. Smooth and convex geometry (T1 to T11)** | | | | | | | | | |
| T1 (primary) | | ● | | | | | | | |
| T2 (primary) | | ● | | | | | | | |
| T3 (primary) | ● | ● | | | | | | | |
| T4 (primary) | ● | ● | | | | | | | |
| T4 (alternative, SC acceleration) | | ● | ● | | | | | | |
| T5 (primary) | | ● | ● | | | | | | |
| T6 (primary) | | ● | | ● | | | | | |
| T7 (primary, QG route) | ● | ● | | | | | | | ● |
| T7 (alternative, PL plus composite route) | | ● | | ● | | | | ● | |
| T8 (primary, error bound route) | ● | ● | | ● | ● | | | | |
| T8 (alternative, PL route) | | ● | | ● | | | | | |
| T9 (primary) | ● | ● | | | | | | | |
| T10 (primary) | ● | ● | | ● | | | | | ● |
| T11 (primary) | ● | ● | | | | | | | |
| **B. Classes and lower bounds (T12 to T14)** | | | | | | | | | |
| T12 (class, SC) | | ● | ● | | | | | | |
| T12 (class, PL) | | ● | | ● | | | | | |
| T13 (primary) | ● | | | | | ● | | | |
| T14 (primary) | | | | | | | ● | | |
| **C. Stochastic optimization (T15 to T20)** | | | | | | | | | |
| T15 (primary) | ● | ● | | | | | ● | | |
| T16 (primary) | ● | ● | | | | | ● | | |
| T17 (primary, PL route) | | ● | | ● | | | ● | | |
| T17 (alternative, SC route) | | ● | ● | | | | ● | | |
| T18 (primary, PL route) | | ● | | ● | | | | | |
| T18 (alternative, SC route) | | ● | ● | | | | | | |
| T19 (primary) | | | | | | | ● | | |
| T20 (primary, PL route) | | ● | | ● | | | ● | | |
| T20 (alternative, SC route) | | ● | ● | | | | ● | | |
| **D. Composite and proximal methods (T21 to T25)** | | | | | | | | | |
| T21 (primary) | | ● | | | | | | ● | |
| T22 (primary) | ● | ● | | | | | | ● | |
| T23 (primary) | ● | ● | | | | | | ● | |
| T24 (primary, PL route) | | ● | | ● | | | | ● | |
| T24 (alternative, SC route) | | ● | ● | | | | | ● | |
| T25 (primary) | ● | ● | | | | | | ● | |
| **E. Geometry and nonconvex structure (T26 to T30)** | | | | | | | | | |
| T26 (primary) | | | | | ● | | | | |
| T27 (primary) | ● | | | | | | | | |
| T28 (primary) | | | | | ● | | | | |
| T29 (primary, PL route) | | ● | | ● | | | | | |
| T29 (alternative, SC route) | | ● | ● | | | | | | |
| T30 (primary) | | ● | | | | | | | |
| **F. Constraints, projections, and diagnostics (T31 to T35)** | | | | | | | | | |
| T31 (primary) | ● | ● | | | | | | | |
| T32 (primary) | ● | | | | ● | | | | |
| T33 (primary, error bound route) | ● | ● | | | ● | | ● | | |
| T33 (alternative, PL route) | | ● | | ● | | | ● | | |
| T33 (alternative, QG route) | ● | ● | | | | | ● | | ● |
| T34 (primary) | ● | ● | | | ● | | | | |
| T35 (probe, PL route) | | ● | | ● | | | | | |
| T35 (probe, QG route) | ● | ● | | | | | | | ● |
| T35 (probe, smooth baseline) | | ● | | | | | | | |

## 4.2 Matrix analysis and atlas compression

The first observation from Table 1 is sparsity. In the current ledger, most theorem variants depend on two or three axioms, and none depends on more than four. This is informative because the axioms were chosen to be broad, yet the proofs repeatedly reuse a small number of mechanisms. Smoothness (A2) appears in the large majority of variants, while convexity (A1) appears in fewer than half. The noise axiom (A7) and the PL axiom (A4) each appear in roughly a quarter of the variants, and the composite axiom (A8) is concentrated in the proximal block. The bounded subgradient axiom (A6) appears only in the nonsmooth barrier row, which makes precise a common practitioner intuition: in our core theorem set, nonsmoothness functions primarily as a ceiling rather than as a structure that is systematically exploited. These counts are not meant to be universal statistics about the literature, but they are a faithful summary of which assumptions our selected guarantees actually consume.

The second observation is that many conclusions admit more than one inclusion minimal proof route. Several linear convergence statements have both a strong convexity route and a PL route, and several results that rely on metric regularity admit alternative routes through error bounds, PL geometry, or quadratic growth. In the atlas, these alternatives matter because they describe distinct ways to reach the same behavioral regime, and they explain why different papers and different communities sometimes impose different conditions yet obtain essentially the same qualitative guarantee.

To obtain a compact global view, we treat the dependency ledger as data. Concretely, we form a binary matrix $X \in \{0, 1\}^{m \times 9}$ whose rows are the theorem variants in Table 1, and we compute a low rank nonnegative factorization $X \approx WH$ with rank $k = 5$. The reconstruction score is $R_F^2 \approx 0.885$ in the current ledger, indicating that five factors capture most of the structure in axiom usage. The factors are interpretable because each one concentrates mass on a small set of axioms. In our runs, one factor emphasizes smooth proximal PL structure (A2, A4, A8), another emphasizes smooth convex geometry with quadratic growth as a curvature surrogate (A1, A2, A9), a third corresponds to strong curvature (A2, A3), a fourth isolates stochasticity (A7), and a fifth isolates metric regularity (A5). We use this factorization as a compression device, not as a new theory statement: it provides a low dimensional coordinate system in which theorem families and algorithm classes can be visualized and compared without enumerating all $2^9$ worlds.

### 4.3 White Spaces and Underrepresented Intersections

The atlas is designed to be extensible; we therefore formalize where extensions would add the most value. A *world* is a subset $W \subseteq \{A1, \ldots, A9\}$. Let $\mathcal{B} = \{B_1, \ldots, B_m\}$ denote the axiom supports of the theorem variants recorded in Table 1, with $B_i = \{A_k : X_{ik} = 1\}$. The *coverage* of a world $W$ is

$$c(W) := \big| \big\{ i \in \{1, \ldots, m\} : B_i \subseteq W \big\} \big|,$$

i.e. the number of theorem variants whose inclusion-minimal axiom set is satisfied in $W$. Worlds with small $c(W)$, especially those combining axioms from distinct roofline tags, are not unsupported in principle, but are weakly represented in the current ledger, so the atlas provides limited guidance there. We call such regions *white spaces*.

Scanning the ledger reveals three practically motivated intersections that are absent from the present core theorem set. We emphasize that the claim is not that these topics are absent from the literature, but that they are not yet represented in our ledger and therefore not integrated into the atlas compression or diagnostics pipeline.

**A7 ∩ A8: stochastic composite optimization.** No theorem variant in Table 1 combines bounded-variance noise (A7) with composite proximal structure (A8). This is the setting in which both noise floors and proximal costs matter simultaneously, and the most useful guarantees would quantify how variance reduction, batch size, and proximal evaluation cost trade off within a single rate expression.

**A5 ∩ A8: composite problems under metric regularity.** The ledger contains no variant combining composite structure (A8) with error-bound geometry (A5). This intersection arises naturally in structured regularization problems—e.g. $\ell_1$-penalized objectives with partially smooth structure—where identification and restart-driven linear phases are observed empirically, suggesting that a unified treatment of proximal methods under metric regularity would be valuable.

**A6 + additional curvature: nonsmooth but geometrically benign objectives.** The nonsmooth axiom (A6) currently appears only in the nonsmooth-barrier row; no theorem exploits additional geometric structure on top of nonsmoothness. A natural extension is to replace the smooth PL inequality with a proximal analog—for instance, a condition stated in terms of the proximal gradient mapping or the Moreau envelope—to formalize when linear-rate phases can occur even when the objective itself is nonsmooth.

## 5 Boundary Diagnostics and the Twelve Question Control Plane

The atlas is a static object: given a set of axioms, it specifies which rate families and limiting behaviors are consistent with that world. Boundary diagnostics invert this direction. Given a short trace segment from a run, the control plane estimates which roofline mechanism currently limits progress and whether

Table 2: The twelve diagnostics used by the control plane. Each question returns a typed output that is logged and, when applicable, consumed by downstream questions.

| ID | Role | Output |
|---|---|---|
| Q1 | Fit roofline templates on $\mathcal{W}$ and measure distances | Normalized distances to curvature, nonsmooth, accelerated, and noise templates; fit flags |
| Q2 | Convert Q1 into a conservative ceiling decision | Binding label (or `Undetermined`) and evidence scores from bootstrap stability checks |
| Q3 | Compare cross frontier moves | ROI ranked list of candidate interventions |
| Q4 | Tune within the current regime | Best parameter adjustment that preserves the current label |
| Q5 | Quantify ROI reliability | Confidence intervals and a positive ROI eligibility flag per move |
| Q6 | Check temporal stability | Consistency score across overlapping tail windows |
| Q7 | Decide whether pilots are worth running | Value of information score and recommended pilot runs |
| Q8 | Evaluate two step sequences | Best two step plans with combined ROI and uncertainty |
| Q9 | Allocate budget across multiple traces | Portfolio ranking of (scenario, move) pairs |
| Q10 | Enforce feasibility constraints | Hard vetoes (memory, energy, policy, fairness constraints) |
| Q11 | Monitor drift during deployment | Triggers for re diagnosis or rollback |
| Q12 | Produce an audit record | Structured log of trace window, diagnostics, decision, and artifacts |

any available change is worth its cost. The goal is not to predict the full future trajectory, but to support repeatable decisions of the form: identify a plausible ceiling, compare a small set of moves that target that ceiling, and either act or stop.

We model a run as a logged trace

$$\mathcal{T} \;=\; \big\{(\Delta_k,\; \eta_k,\; b_k,\; \widehat{\sigma}_k^2)\big\}_{k=0}^{K},$$

where $\Delta_k$ is a gap proxy (for example $f(x_k) - f^\star$ when $f^\star$ is known, or $f(x_k) - \min_{j \leq K} f(x_j)$ otherwise), $\eta_k$ is the step size, $b_k$ is the minibatch size, and $\widehat{\sigma}_k^2$ is an online variance proxy. All diagnostics operate on a tail window $\mathcal{W} = \{k_0, \ldots, K\}$ chosen so that hyperparameters are approximately stable and the trace shows no obvious phase transition. The window selection rule and numeric defaults are given in Appendix C.6.

Table 2 summarizes the twelve questions. In practice, most decisions are driven by Q1 to Q5, while Q6 to Q12 provide stability checks, pilot design, and governance.

Q1 evaluates a small bank of rate templates derived from the theorem library in Table 1. The templates correspond to the roofline ceilings used throughout the paper: linear contraction in curvature dominated regimes, sublinear decay in smooth convex regimes (including accelerated decay when applicable), the nonsmooth ceiling for Lipschitz objectives, and a noise floor model whose scale depends on $\eta_k\widehat{\sigma}_k^2/b_k$. Q1 returns normalized distances to each template together with fit quality flags so that poor fits do not propagate.

Q2 turns these distances into a binding ceiling decision using bootstrap resampling of $\mathcal{W}$. A ceiling is declared binding only when it fits well and clearly dominates competing explanations; otherwise Q2 returns `Undetermined` with low confidence. This conservative behavior is intentional because unstable tails and mixed regimes are common in practice, and forcing a confident label in those settings tends to encourage unnecessary changes.

Action selection separates cross frontier moves from within regime tuning. Q3 evaluates a discrete menu of candidate moves $\mathcal{M}$, such as increasing batch size, switching to a variance reduced method, adding proximal structure, or applying smoothing. Each move $m \in \mathcal{M}$ has a cost $c_m$ and is scored by a trace level metric $S$ where smaller is better, for example time to reach a target gap or area under the gap curve over a fixed budget. We define

$$\mathrm{ROI}_m \;=\; \frac{S_{\mathrm{baseline}} - S_{\mathrm{after}(m)}}{c_m},$$

so positive ROI corresponds to improvement per unit cost. Q4 applies when cross frontier changes are not justified or are vetoed: it searches a small family of same regime adjustments, such as modest step size changes, schedule tweaks, or restart rules, and returns the best admissible improvement.

Q5 to Q8 quantify uncertainty and guard against overfitting the decision to a particular tail segment. Q5 attaches confidence intervals to each ROI estimate and marks moves as eligible only when the improvement remains positive at the chosen confidence level. Q6 repeats the diagnosis across neighboring tail windows to measure temporal stability. Q7 recommends small pilots when the expected value of

Table 3: Workloads used in our evaluation and the candidate moves considered by Q3. The synthetic worlds isolate a single dominant roofline ceiling in the tail; the CIFAR–10 scenarios illustrate nonstationary training dynamics under two budgets.

| Workload | Intended limiter | Baseline run | Candidate moves |
|---|---|---|---|
| W1 | Curvature | Gradient descent on smooth strongly convex objective | Batch doubling, variance reduction, smoothing |
| W2 | Nonsmooth | First order method on convex Lipschitz objective with an active kink | Batch doubling, variance reduction, smoothing |
| W3 | Accelerated | Accelerated method on smooth convex objective | Smoothing, batch doubling, variance reduction |
| W4 | Noise | Stochastic gradients with bounded variance and constant step size | Batch doubling, variance reduction, smoothing |
| CIFAR Scenario A | Mixed | SGD, batch 64, 5000 steps | Batch doubling, Adam |
| CIFAR Scenario B | Mixed | SGD, batch 32, 1000 steps | Batch doubling, Adam |

additional information is high relative to pilot cost. Q8 optionally considers two step sequences when a preparatory change is needed before a larger regime change can pay off.

Q9 to Q12 are governance layers that make the control plane usable in a real workflow. Q9 prioritizes interventions across multiple traces under a global compute budget. Q10 applies feasibility constraints as hard vetoes. Q11 monitors diagnostic signals over time to trigger re diagnosis or rollback when the regime drifts. Q12 writes a structured audit record that includes the selected tail window, diagnostic outputs, the final action, and pointers to the artifacts used to make the decision.

# 6 Experimental Results

We evaluate whether the atlas is useful as (i) a diagnostic tool that can identify which roofline mechanism is limiting progress from an optimization trace, and (ii) a decision rule that can rank a small set of candidate interventions by value per unit cost. We report controlled synthetic results, where each roofline ceiling is activated by construction, and a CIFAR–10 case study, where the dynamics are nonconvex, noisy, and nonstationary.

Each run produces a trace $\{f_k\}_{k=0}^K$. On synthetic objectives we report the optimality gap $f_k - f^\star$. On CIFAR–10, where $f^\star$ is unknown, we use the gap surrogate $f_k^{\mathrm{gap}} := \ell_k - \min_{t \leq k} \ell_t$ based on the minibatch loss $\ell_k$. We apply the Q1 to Q3 pipeline from Section 5: Q1 fits canonical roofline templates on a tail window and returns distances to curvature, nonsmooth, accelerated, and noise ceilings; Q2 bootstraps the tail fit to produce conservative evidence scores; Q3 compares a small set of candidate moves via pilot traces and ranks them by return on investment. Implementation details and logging conventions are provided in the released artifacts.

## 6.1 Synthetic validation of diagnosis and action

Figure 3 summarizes end to end behavior on W1 to W4. The trace panels confirm that each synthetic world exhibits the intended tail shape: semilog linear decay in the curvature world, sublinear decay in the nonsmooth world, accelerated decay in the smooth convex world, and a visible plateau in the noise world. The diagnostic panels then test whether the control plane recovers these regimes without being told which world it is in. Q1 distances and Q2 evidence concentrate on the correct ceiling in all four cases, with competing ceilings either failing the fit gates or receiving negligible evidence.

The same figure also shows the resulting Q3 choices. In W1 and W2, the best ranked move is a capacity amplification intervention, consistent with the fact that the active limiter is not removed by changing the objective class. In W3, the best ranked move is the assumption relaxation that unlocks the accelerated ceiling. In W4, all tested interventions have essentially zero marginal value and the correct output is a no action decision. This is an important regime for practice: once a run is variance limited under the chosen compute budget, optimizer changes may have little effect unless they directly address the noise mechanism.

Finally, Figure 3 reports the factor loadings used to compress the theorem by axiom ledger. The learned components align with the qualitative families used throughout the paper, separating smooth curvature

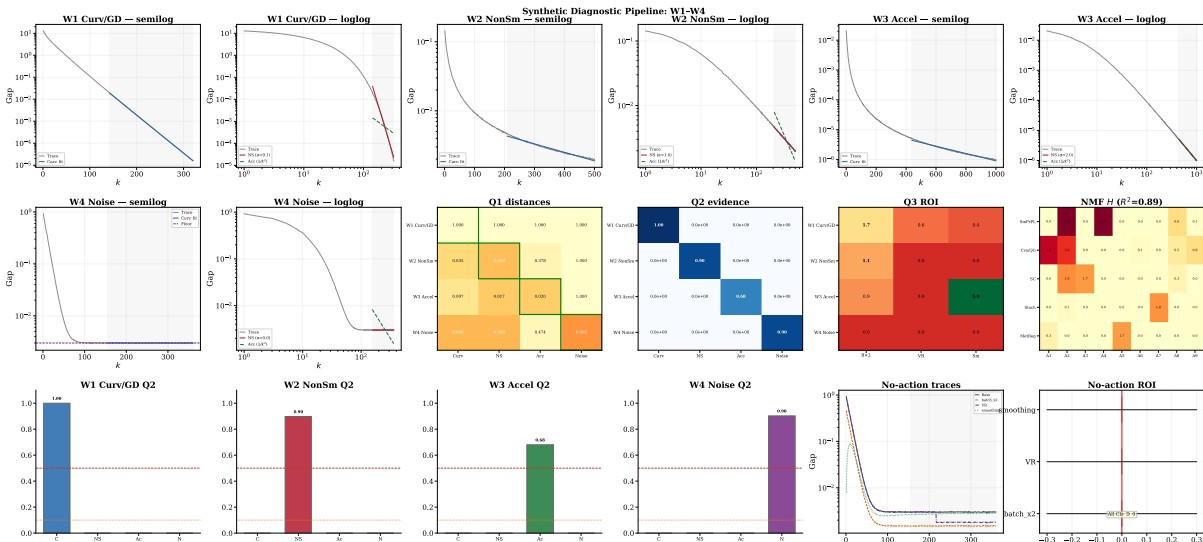

Figure 3: Synthetic validation on W1 to W4. Left: gap traces with tail fits. Middle: Q1 distances and Q2 evidence identify the intended binding ceiling in each world. Right: Q3 selects the highest ROI move in W1 to W3 and returns no beneficial move in the noise limited world W4. The factor loading panel summarizes the rank five compression of the theorem by axiom ledger.

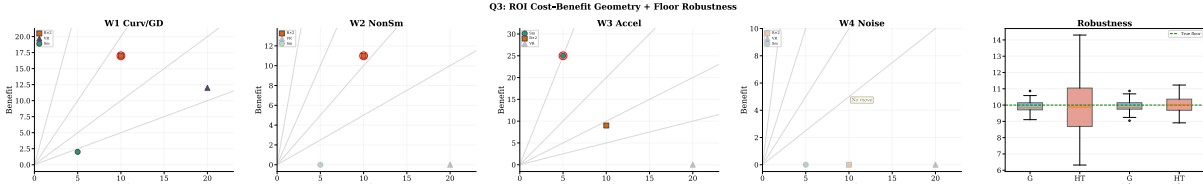

Figure 4: Q3 as a cost benefit rule and a floor robustness check. Cost benefit panels show candidate moves with constant ROI rays. The robustness panel compares floor estimators under Gaussian and heavy tailed noise, illustrating the need for robust tail statistics.

structure, proximal plus Polyak Łojasiewicz structure, stochastic structure, and metric regularity structure, which supports using the ledger as a compact representation of the atlas.

### 6.2 Q3 as a cost benefit rule and floor robustness

Figure 4 visualizes Q3 as cost benefit geometry. Each candidate move is a point with horizontal coordinate equal to its cost and vertical coordinate equal to its measured benefit; constant ROI corresponds to rays from the origin, so selecting the best move amounts to selecting the steepest slope. This view makes the synthetic outcomes easy to interpret: curvature and nonsmooth worlds admit moves with clear positive slope, whereas the noise world yields points clustered near zero benefit.

The same figure includes a robustness check for floor estimation under heavy tailed noise. A naive mean of late iterates is unstable when rare large deviations occur, while a trimmed estimator remains well behaved. This motivates the robust tail statistics and conservative evidence gating used in Q1 and Q2 when reasoning about noise ceilings.

### 6.3 CIFAR–10 case study under two budgets

Figure 5 applies the pipeline to CIFAR–10 under two training budgets. In both scenarios the roofline template fits are less decisive than in the synthetic worlds, and Q2 assigns low evidence across ceilings. This behavior is expected: neural network training traces are nonstationary over the horizon of interest, and a single asymptotic template over a fixed tail window is often an imperfect description of the dynamics. The control plane therefore avoids forcing a strict binding label when the trace does not support one.

Even in this conservative regime, pilot comparisons remain informative. The base versus move overlays in Figure 5 show how batch doubling and switching to Adam change the observed loss trace under the same budget. Q3 converts these pilot traces into an ROI ranking under the chosen cost model. The main takeaway is that the atlas separates diagnosis from action: it can remain cautious about regime

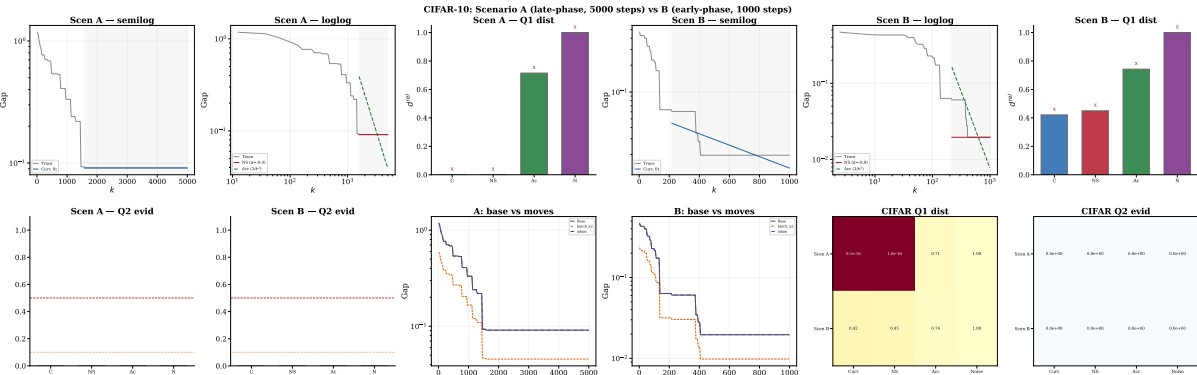

Figure 5: CIFAR–10 case study under two budgets. Q1 distances and Q2 evidence remain conservative due to nonstationary dynamics. The base versus move overlays show pilot traces for batch doubling and Adam, which Q3 uses to rank interventions by ROI.

identification while still providing a principled, trace based way to decide whether an intervention is worth the additional compute in a given training phase.

## 7 Discussion, Limitations, and Conclusion

The Optimization Atlas turns first-order optimization into a finite, auditable design space: nine axioms (A1-A9) define a $2^9$-dimensional cube, and the theorem ledger together with the dependency matrix, makes each guarantee traceable to its inclusion-minimal assumptions. A rank-5 NMF ($\approx 88.5\%$ variance explained) summarizes recurring assumption modes, and Boundary Diagnostics (Q1–Q12) read a tail window to identify the active theoretical ceiling and rank candidate interventions by ROI under a user-specified cost model. In practice, this shifts the central decision from choosing an optimizer family to clearing the cheapest binding axiom. Equivalent linear-rate bases (strong convexity, Polyak–Łojasiewicz, quadratic growth, error bounds) are explicitly interchangeable, and noise-limited regimes are recognizable by their $\Theta(\eta\sigma^2/b)$ floor, at which point variance or model capacity, rather than the optimizer, dominates. The framework also turns white-space regions of the cube into concrete, testable theory gaps, and it justifies a `no_action` recommendation when all optimizer-only moves are predicted to be net-negative.

**Limitations** The current Atlas is centered on first-order methods for single-objective minimization in Euclidean space and, in the main ledger, on standard bounded-variance stochasticity. Extending the framework to second-order methods, constrained or multi-objective problems, non-Euclidean geometries, or broader dynamical settings such as minimax objectives, variational inequalities, or games requires introducing additional axioms and auditing new ceilings and dependencies.

The released theorem ledger records one audited Pareto-minimal route per result, so counts and dependency loads describe the audited snapshot rather than the full literature; alternative minimal bases can shift frequencies without changing the qualitative map. The factorization view is also a choice: the rank-5 NMF modes are selected for stability and interpretability and are not unique. The diagnostics assume approximate stationarity within the chosen tail window; when training dynamics drift rapidly, a single run can traverse multiple regions of the cube and a single-window diagnosis can blur these transitions. ROI rankings depend on the user's cost model, which can vary across deployments; we therefore surface uncertainty and emphasize small pilots when conclusions are sensitive to unverified assumptions. Finally, coverage is necessarily incomplete: the current ledger is a finite snapshot (35 theorems, 48 variants), and adding more rows or expanding to new problem classes will require new axioms and diagnostics.

**Outlook** Deep learning training often violates classical smoothness and bounded-variance noise while exhibiting additional regularities. Appendix D outlines how the same methodology can extend by replacing or augmenting bits rather than abandoning the framework, for example by modeling heavy-tailed noise, relative smoothness in Bregman geometries, interpolation or strong-growth conditions, scale invariances that induce self-annealing, and edge-of-stability curvature dynamics. The central workflow remains unchanged: propose candidate axioms, audit theorem dependencies, and re-run the same diagnostic and ROI control loop on modern workloads.

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

# A Full Theorem Catalog and Minimal Assumption Tags

This appendix contains the full statements of all lemmas, propositions, and theorems used to build the Optimization Assumption Atlas. Each result is annotated with a *minimal* assumption tag. The assumption labels are defined in Section 3. The complete dependency matrix is in Section 4.

This section gathers the lemmas, propositions, and theorems that support the Optimization Assumption Atlas. We group them by the smoothness–convexity framework and attach a "minimal" tag that lists the smallest axiom set needed for each claim. These explicit dependencies ground the later algorithmic and complexity analyses.

**Reading guide and notation.** Section 4 and the file `full_matrix` list every result. Tags such as "(minimal: A2+A4)" mark the Pareto-minimal axiom set whose independence we prove in §**??**.[1] We write $\mathbb{E}[\cdot]$ and $\mathrm{Var}[\cdot]$ for expectation and variance. Here $L$ is the smoothness constant, $\mu$ the strong-convexity or PL constant, and $\sigma^2$ the gradient-noise variance. Constants $C, C_1, C_2$ depend only on problem data. For composite objectives we set $F = g + h$ with $g$ $L$-smooth and $h$ proper, closed, convex; all bounds hold on $\mathrm{dom}\, F$.

## A.1 Smooth and Convex Geometry

A descent inequality bounds the decrease produced by a single iterate. Under $L$-smoothness and convexity we collect the relevant inequalities and rates.

**Lemma 1** (Smooth descent lemma). *Under $L$-smoothness (A2),*

$$f(y) \leq f(x) + \langle \nabla f(x), y - x \rangle + \tfrac{L}{2}\|y - x\|^2 \quad \text{for all } x, y.$$

(minimal: A2)

**Proposition 1** (GD one-step decrease). *For gradient descent with stepsize $\eta \in (0, 1/L]$ on an $L$-smooth function,*

$$f(x_{k+1}) \leq f(x_k) - \tfrac{\eta}{2}\|\nabla f(x_k)\|^2, \quad k \geq 0.$$

(minimal: A2; stepsize $\eta \in (0, 1/L]$)

**Theorem 1** (Sublinear GD on smooth convex). *Let $f$ be convex (A1) and $L$-smooth (A2). With gradient descent and $\eta = 1/L$,*

$$f(x_k) - f^\star \leq \frac{L\|x_0 - x^\star\|^2}{2k}, \quad k \geq 1.$$

(minimal: A1+A2)

**Theorem 2** (Nesterov acceleration). *If $f$ is convex (A1) and $L$-smooth (A2), an accelerated first-order method achieves*

$$f(x_k) - f^\star \leq \frac{C\, L\|x_0 - x^\star\|^2}{(k + 1)^2},$$

*with constant $C > 0$ ($C = 2$ in the standard scheme). When $f$ is also $\mu$-strongly convex (A3), suitable tuning gives*

$$f(x_k) - f^\star \leq C' \exp\!\Big(-\frac{k}{c\sqrt{L/\mu}}\Big)\big(f(x_0) - f^\star\big),$$

*for constants $C', c > 0$, yielding iteration complexity $\mathcal{O}(\sqrt{L/\mu}\log(1/\varepsilon))$. This matches the lower bound in Theorem 7.* (minimal: A1+A2; the strongly convex extension also uses A3)

**Theorem 3** (Linear rate via strong convexity). *Let $f$ be $\mu$-strongly convex (A3) and $L$-smooth (A2). With $\eta = 1/L$,*

$$f(x_k) - f^\star \leq \Big(1 - \tfrac{\mu}{L}\Big)^k \big(f(x_0) - f^\star\big), \quad k \geq 0.$$

(minimal: A2+A3)

**Theorem 4** (Linear rate via PL). *Under PL (A4) and $L$-smoothness (A2), any stepsize $\eta \in (0, 1/L]$ gives*

$$f(x_k) - f^\star \leq (1 - \eta\mu)^k \big(f(x_0) - f^\star\big), \quad k \geq 0.$$

(minimal: A2+A4; stepsize $\eta \in (0, 1/L]$)

---

[1]"Minimal" means that *no proper subset* of the stated axioms suffices.

*Remark.* Because strong convexity (A3) implies PL (A4), Theorem 4 subsumes Theorem 3. We retain Theorem 3 to highlight the strongly convex case, while Theorem 4 also applies to nonconvex objectives that satisfy PL.

**Proposition 2** (Projected GD under quadratic growth)**.** *Let $f$ be convex (A1), satisfy quadratic growth (A9) on a closed convex set, and have an L-Lipschitz gradient on the sublevel set (A2). Projected GD with $\eta \in (0, 1/L]$ satisfies*

$$f(x_k) - f^\star \leq \rho^k \big( f(x_0) - f^\star \big) \quad \text{for some } \rho \in (0, 1).$$

(minimal: A1+A2+A9)

**Proposition 3** (Sharpness and restarts)**.** *Let $f$ be convex (A1), L-smooth (A2), and satisfy a local error bound (A5) near $\mathcal{X}^\star$. Scheduled restarts of an accelerated method create consecutive linear phases until the iterates enter the neighborhood limited by sharpness.* (minimal: A1+A2+A5)

**Lemma 2** (Baillon–Haddad cocoercivity)**.** *For convex L-smooth $f$ (A1+A2) the gradient is $\frac{1}{L}$-cocoercive:*

$$\langle \nabla f(x) - \nabla f(y), x - y \rangle \geq \tfrac{1}{L} \|\nabla f(x) - \nabla f(y)\|^2 \quad \text{for all } x, y.$$

(minimal: A1+A2)

**Theorem 5** (Convex PL $\Leftrightarrow$ quadratic growth)**.** *Let $f$ be proper, closed, convex with nonempty $\mathcal{X}^\star$ and differentiable on a sublevel set containing the iterates. On that set $PL(\mu)$ for some $\mu > 0$ holds iff quadratic growth with constant $\gamma > 0$ holds. Moreover, quadratic growth with $\gamma$ implies PL with $\mu = \gamma/4$, and PL with $\mu$ implies quadratic growth with $\gamma = 4\mu$. Together with Theorem 4 and Proposition 2, PL and QG give equivalent linear rates up to constants under A1–A2.* (minimal: A1+A2+A4+A9)

## A.2 Lower Bounds and Optimality

An oracle call is one query to a first-order black box that returns an exact or stochastic gradient. The "minimal" tag records the smallest axiom set known to imply each bound. The next results give matching oracle-complexity lower bounds.

**Theorem 6** (Smooth convex lower bound)**.** *In the black-box first-order model for smooth convex minimization, every method faces an L-smooth convex instance for which, after $k$ oracle queries,*

$$f(x_k) - f^\star \geq \Omega\Big( \frac{L\|x_0 - x^\star\|^2}{(k+1)^2} \Big).$$

(minimal: A1+A2; model: first-order black-box oracle)

**Theorem 7** (Strongly convex oracle complexity)**.** *In the same model, any sequence that reaches accuracy $\varepsilon$ for an L-smooth, $\mu$-strongly convex function needs at least*

$$\Omega\big( \sqrt{L/\mu} \, \log(1/\varepsilon) \big)$$

*oracle calls.* (minimal: A2+A3; model: first-order black-box oracle)

**Proposition 4** (No acceleration without smoothness)**.** *Consider nonsmooth convex objectives with G-Lipschitz continuity (A1+A6) on a domain of radius R. The projected subgradient method with stepsizes $\eta_k \propto 1/\sqrt{k}$ achieves*

$$f(\bar{x}_k) - f^\star \leq c_1 \frac{GR}{\sqrt{k}},$$

*for a universal constant $c_1 > 0$, where $\bar{x}_k$ is a suitable ergodic average. Conversely, the first-order black-box model admits a Lipschitz convex instance for which*

$$f(x_k) - f^\star \geq c_2 \frac{GR}{\sqrt{k}},$$

*for another constant $c_2 > 0$. Hence the best possible rate is $\Theta(1/\sqrt{k})$, ruling out Nesterov-style $\mathcal{O}(1/k^2)$ acceleration in the nonsmooth Lipschitz setting. In the Lipschitz plus strongly convex case one improves to $\mathcal{O}(1/k)$ but not to $\mathcal{O}(1/k^2)$ without smoothness.* (minimal: A1+A6; the $\mathcal{O}(1/k)$ improvement additionally uses A3)

**Theorem 8** (Information-theoretic noise floor)**.** *In the stochastic first-order oracle model with bounded-variance unbiased gradients (A7) and constant stepsize $\eta$, every method encounters a problem instance obeying*

$$\liminf_{k \to \infty} \mathbb{E}[f(x_k) - f^\star] \geq c \, \eta \sigma^2$$

*for a universal constant $c > 0$.* (minimal: A7; model: stochastic oracle)

## A.3 Stochastic Optimization

*SGD* denotes the update $x_{k+1} = x_k - \eta_k g(x_k, \xi_k)$ with unbiased noisy gradients $g(x_k, \xi_k)$. "Variance reduction" refers to control-variates schemes such as SVRG or SARAH that shrink gradient noise.

**Theorem 9** (SGD with constant steps)**.** *Assume convexity (A1), L-smoothness (A2), and bounded-variance noise (A7). For SGD with unbiased gradients and $\eta \in (0, 1/L)$,*

$$\mathbb{E}[f(x_k) - f^\star] \leq \frac{C_1 L \|x_0 - x^\star\|^2}{k} + C_2 \eta \sigma^2, \quad k \geq 1,$$

*so the transient term decays as $\mathcal{O}(1/k)$ while the steady-state floor scales as $\Theta(\eta \sigma^2)$.* (minimal: A1+A2+A7)

**Theorem 10** (SGD with decay)**.** *Under the same assumptions, stepsizes $\eta_k \propto 1/\sqrt{k}$ give*

$$\mathbb{E}[f(x_k) - f^\star] = \mathcal{O}(1/\sqrt{k}).$$

(minimal: A1+A2+A7)

**Theorem 11** (PL plus variance reduction yields linear)**.** *For finite sums and under A2, A4, A7, variance-reduced schemes such as SVRG or SARAH satisfy*

$$\mathbb{E}[f(x_k) - f^\star] \leq C\, c^k,$$

*where $k$ counts gradient-equivalent epochs and $c \in (0, 1)$.* (minimal: A2+A4+A7; model: finite sum)

**Proposition 5** (Polyak steps under PL)**.** *Under PL (A4) and L-smoothness (A2), Polyak stepsizes give global linear convergence:*

$$f(x_k) - f^\star \leq C\, c^k, \quad k \geq 0,$$

*for $c \in (0, 1)$ and $C > 0$.* (minimal: A2+A4)

**Proposition 6** (Minibatch variance law)**.** *With batch size $b$ and i.i.d. samples,*

$$\mathrm{Var}[\bar{g}_b(x) \mid x] \leq \sigma^2/b.$$

(minimal: A7)

**Theorem 12** (PL: constant versus decreasing steps)**.** *Under PL (A4), L-smoothness (A2), and bounded-variance noise (A7), SGD with constant $\eta \in (0, 1/L)$ contracts linearly in expectation to a neighborhood of radius $\Theta(\eta \sigma^2/\mu)$. With stepsizes satisfying $\sum_k \eta_k = \infty$ and $\sum_k \eta_k^2 < \infty$, SGD converges to $x^\star$; the non-asymptotic rate depends on the schedule.* (minimal: A2+A4+A7)

## A.4 Composite and Proximal Structure

The proximal operator $\mathrm{prox}_{\eta h}(v) = \arg\min_x \{h(x) + \|x - v\|^2/(2\eta)\}$ splits the smooth $g$ and nonsmooth $h$ terms. We summarize guarantees for proximal algorithms.

**Lemma 3** (Proximal descent inequality)**.** *For $F(x) = g(x) + h(x)$ with $g$ L-smooth and $h$ proper, closed, convex (A2+A8), the proximal-gradient step with $\eta \in (0, 2/L)$ obeys*

$$F(x_{k+1}) \leq F(x_k) - \left(\frac{1}{\eta} - \frac{L}{2}\right)\|x_{k+1} - x_k\|^2.$$

(minimal: A2+A8)

**Theorem 13** (Prox-GD sublinear)**.** *Let $F = g + h$ be convex with $g$ L-smooth and $h$ proper, closed, convex (A1+A2+A8). With $\eta \leq 1/L$,*

$$F(x_k) - F^\star = \mathcal{O}\left(\tfrac{1}{k}\right).$$

(minimal: A1+A2+A8)

**Theorem 14** (FISTA $\mathcal{O}(1/k^2)$)**.** *Accelerated proximal gradient (FISTA) guarantees*

$$F(x_k) - F^\star \leq \frac{C\, L \|x_0 - x^\star\|^2}{(k+1)^2},$$

*with $C > 0$ (C = 2 in the standard scheme).* (minimal: A1+A2+A8)

**Theorem 15** (Prox-PL linear). *If PL holds on F (or g is L-smooth and F satisfies a PL/error-bound condition), proximal-gradient and variance-reduced variants converge linearly:*

$$F(x_k) - F^\star \leq C\, c^k$$

*for $c \in (0,1)$ and $C > 0$.* (minimal: A2+A4+A8)

**Proposition 7** (Exact support under identifiability). *Let $F(x) = g(x) + h(x)$ with g convex and L-smooth and h a convex sparsity penalty (e.g., $\ell_1$). Under standard identifiability conditions at $x^\star$, IST/soft-thresholding recovers the true support in finitely many steps.* (minimal: A1+A2+A8; model: identifiability / nondegeneracy)

## A.5 Geometry, Error Bounds, and KL

The Kurdyka–Łojasiewicz (KL) inequality links local geometry to algorithmic speed.

**Theorem 16** (KL exponent implies rates). *If f satisfies a Kurdyka–Łojasiewicz inequality with exponent $\theta \in [0,1)$ and the method is gradient-like (or proximal-gradient) with sufficient decrease and a relative-error condition, the convergence rate follows from $\theta$ (linear if $\theta \in [0, \frac{1}{2})$, etc.).* (minimal: A5)

**Theorem 17** (Error bound ⇔ metric subregularity). *For proper, closed, convex f with nonempty $\mathcal{X}^\star$, local error bounds (A5) are equivalent to metric subregularity of $\partial f$ under mild regularity conditions.* (minimal: A1; regularity: proper, closed, convex with nonempty $\mathcal{X}^\star$)

**Proposition 8** (Error bound gives distance decay). *If f satisfies a p-error bound (A5), then $\mathrm{dist}(x_k, \mathcal{X}^\star)$ decays at a rate implied by p for standard descent or proximal-gradient methods.* (minimal: A5)

## A.6 Benign Nonconvex Regimes

We focus on two structure-inducing properties: a global PL condition, which rules out flat plateaus, and the strict-saddle property, which guarantees directions of negative curvature at any non-optimal critical point.

**Theorem 18** (Global linear under nonconvex PL). *If PL (A4) holds globally and $\nabla f$ is L-Lipschitz (A2), gradient descent with $\eta \in (0, 1/L]$ converges linearly to a global minimizer.* (minimal: A2+A4)

**Theorem 19** (Strict saddle avoidance). *For $C^2$ functions with the strict-saddle property and L-smooth gradients (A2), gradient descent with random initialization (or small isotropic noise) avoids strict saddles almost surely and converges to a local minimizer.* (minimal: A2; model: randomness)

## A.7 Constraints and Projections

The Hoffman error bound links distance to the feasible set with linear constraint violations.

**Theorem 20** (Projected GD on convex sets). *For a closed convex set and $\eta \leq 1/L$, projected GD on an L-smooth convex objective attains $\mathcal{O}(1/k)$ decay.* (minimal: A1+A2; model: convex set)

**Theorem 21** (Hoffman bound implies linear). *With a Hoffman error bound for polyhedral sets (A5), projected or proximal methods converge linearly.* (minimal: A1+A5; model: linear constraints)

*Remark.* The Hoffman constant depends on the constraint geometry and can be large. For full-rank equalities $Ax = b$ it scales as $1/\sigma_{\min}(A)$, matching the conditioning of $A^\top A$. We treat it as a qualitative indicator of constraint sharpness.

## A.8 Stability, Noise, and Diagnostics

The *sharpness exponent* quantifies how steeply the objective grows near the optimum and thus governs the size of noise-limited basins.

**Proposition 9** (Noise-limited basin under sharpness). *Assume convexity (A1), smoothness (A2), a local error bound (A5), and bounded-variance noise (A7). The steady-state neighborhood of a noise-perturbed linear regime scales as $(\sigma/\alpha)^\beta$ for some $\beta > 0$ determined by the sharpness exponent. Combining this with Theorems 4 and 24 quantifies the curvature–noise trade-off: sharper curvature speeds local convergence but shrinks the noise-limited basin only polynomially.* (minimal: A1+A2+A5+A7)

**Proposition 10** (Restarts create local PL windows)**.** *Assume convexity (A1), smoothness (A2), and a local error bound or sharpness condition (A5) near $\mathcal{X}^\star$. Scheduled restarts mimic local PL behaviour and generate piecewise-linear contraction phases.* (minimal: A1+A2+A5; model: local regularity / sharpness)

**Proposition 11** (Diagnostic implications for axioms)**.** *Assume the iterates stay in a level set where L-smoothness holds (A2). Along the trajectory the following implications hold. First, if there exists $\mu > 0$ with $\|\nabla f(x)\|^2 \geq 2\mu\,(f(x) - f^\star)$, then GD with $\eta \in (0, 1/L]$ contracts linearly at rate at least $1 - \eta\mu$. Second, if for some $\alpha > 0$ we have $\mathrm{dist}(x, \mathcal{X}^\star)^2 \leq \alpha\,(f(x) - f^\star)$, then projected or proximal GD yields linear distance decay. Third, if for some $\eta \in (0, 1/L]$ the step-contraction inequality $\|x_{k+1} - x_k\| \leq \eta\,\|\nabla f(x_k)\|$ holds and the descent lemma applies, then the observed steps are consistent with L-smoothness. These facts motivate the empirical diagnostics in §5.* (minimal: item 1 uses A2+A4; item 2 uses A1+A2+A9 when constraints are present; item 3 uses A2)

## B   Independence and Consistency

We prove that (i) one problem satisfies A1–A9 (consistency) and (ii) each axiom fails in a world where the others needed for some theorem hold, so the theorem breaks (independence).

**Atlas-relative independence.**   Fix $A_i$ and consult the Pareto-minimal bases in Table 1. We call $A_i$ *independent* when a theorem $T$ admits a basis $B$ with $A_i \in B$, and there exists a function/oracle pair $(f, \mathcal{O})$ that satisfies every axiom in $B \setminus \{A_i\}$ but violates $A_i$, so $T$ fails.

### B.1   Consistency: a model where A1–A9 hold

**Lemma 4** (Quadratic composite on a ball satisfies A1–A9)**.** *Let $Q \in \mathbb{R}^{d \times d}$ be symmetric positive definite with eigenvalues in $[\mu, L]$ for $0 < \mu \leq L < \infty$, and let $r \in \mathbb{R}^d$. Given $R > 0$ set $C := \{x \in \mathbb{R}^d : \|x\| \leq R\}$ and*

$$g(x) = \tfrac{1}{2}x^\top Q x + r^\top x, \qquad h(x) = \iota_C(x), \qquad F(x) = g(x) + h(x), \qquad G(x) = \nabla g(x) + \zeta, \quad \mathbb{E}[\zeta \mid x] = 0, \ \mathbb{E}\|\zeta\|^2 \leq \sigma^2.$$

*With $f := F$ every axiom holds on $C$:*

> **A1:** $f = g + h$ *is convex.*
>
> **A2:** $\nabla g(x) = Qx + r$, $\|\nabla g(x) - \nabla g(y)\| \leq L\|x - y\|$.
>
> **A3:** $g$ *is $\mu$-strongly convex.*
>
> **A4:** $\exists\,\mu_{\mathrm{PL}} > 0 : \ \tfrac{1}{2}\|\nabla g(x)\|^2 \geq \mu_{\mathrm{PL}}\big(f(x) - f^\star\big)$.
>
> **A5:** $f(x) - f^\star \geq \tfrac{\mu}{2}\,\mathrm{dist}(x, \mathcal{X}^\star)^2$.
>
> **A6:** $|f(x) - f(y)| \leq M\|x - y\|$, $M := \sup\limits_{x \in C} \|\nabla g(x)\|$.
>
> **A7:** $G(x)$ *is unbiased with variance* $\leq \sigma^2$.
>
> **A8:** $g$ *is $L$-smooth;* $h$ *is proper, closed, convex;* $\mathrm{prox}_{\eta h}(v) = \Pi_C(v)$.
>
> **A9:** $f(x) - f^\star \geq \tfrac{\mu}{2}\,\mathrm{dist}(x, \mathcal{X}^\star)^2$ *near $\mathcal{X}^\star$*

*so QG holds locally with $\gamma = \mu$.*

Lemma 4 proves consistency.

### B.2   Independence: counterworlds and failed results

For each $i \in \{1, \dots, 9\}$ we give a world where $A_i$ fails while the other axioms in at least one basis for a theorem $T$ (Table 1) hold, so $T$ collapses.

**Overview matrix.**   Table 4 lists the counterworlds; details follow.

**A1 (Convexity) is independent.**   With $f(x) = -\tfrac{1}{2}\|x\|^2$ we have $\nabla f(x) = -x$, so A2 holds but A1 fails. Gradient descent with $\eta = 1/L = 1$ yields $x_{k+1} = 2x_k$ and $f(x_k) \to -\infty$, contradicting T3's $O(1/k)$ rate.

Table 4: Independence overview: for each axiom $A_i$ we give a counterworld where $A_i$ fails, all other axioms in at least one sufficient basis for a theorem $T$ hold, and $T$'s conclusion disappears.

| Axiom | Counterworld (where $A_i$ fails) | Witness theorem $T$ |
|---|---|---|
| A1 | $f(x) = -\frac{1}{2}\|x\|^2$ (nonconvex, smooth; A2 holds) | T3 (Sublinear GD: A1+A2) |
| A2 | $f(x) = \|x\|_1$ (convex, nonsmooth; no finite $L$) | T4 (Nesterov accel.: A1+A2) |
| A3 | $f(x) = \frac{1}{2}\|Ax - b\|^2$ with rank$(A) < d$ (no SC) | T5 (Linear via SC: A2+A3) |
| A4 | $f(x) = 1 - e^{-\|x\|^2}$ (smooth; PL fails globally) | T6 (Linear via PL: A2+A4) |
| A5 | $f(x) = x^4$ on $\mathbb{R}$ (no local EB with $p \le 2$) | T8, T26, T28, T32 (EB-dependent) |
| A6 | $f(x) = \frac{1}{2}\|x\|^2$ on $\mathbb{R}^d$ (not Lipschitz) | T13 (Nonsmooth optimality: A1+A6) |
| A7 | $G(x) = \nabla f(x) + \xi$, $\xi$ i.i.d. $\alpha$-stable, $\alpha \in (1,2)$ | T14, T15, T19, T20, T33 (A7-based) |
| A8 | $F(x) = g(x) + \iota_S(x)$, $g = \frac{1}{2}\|x\|^2$, $S = \{\|x\| < 1\}$ (nonclosed) | T21–T25 (prox-based) |
| A9 | $f(x) = x^4$ on $\mathbb{R}$ (no local QG at 0) | T7, T10, T35 (QG-based) |

**A2 ($L$-smoothness) is independent.** Set $f(x) = \|x\|_1$. A1 holds, yet no finite $L$ exists; the $\mathcal{O}(1/k^2)$ rate of T4 is unattainable.

**A3 (Strong convexity) is independent.** Take $f(x) = \frac{1}{2}\|Ax - b\|^2$ with rank$(A) < d$. A3 fails and the linear contraction of T5 disappears.

**A4 (Polyak–Łojasiewicz) is independent.** Choose $f(x) = 1 - e^{-\|x\|^2}$. The PL inequality breaks as $\|x\| \to \infty$, so T6's linear rate vanishes.

**A5 (Local error bound) is independent.** With $f(x) = x^4$ near 0, no $(\alpha, p \le 2)$ satisfies the bound; EB-based results T8, T26, T28, and T32 fall.

**A6 (Lipschitz objective) is independent.** For $f(x) = \frac{1}{2}\|x\|^2$ on $\mathbb{R}^d$, A6 fails, so T13 cannot be stated.

**A7 (Bounded-variance noise) is independent.** Let $G(x) = \nabla f(x) + \xi$ with $\xi$ i.i.d. $\alpha$-stable, $1 < \alpha < 2$. Infinite variance voids A7 and invalidates T14, T15, T19, T20, and T33.

**A8 (Composite/prox structure) is independent.** Take $g(x) = \frac{1}{2}\|x\|^2$ and $h(x) = \iota_S(x)$ with open $S = \{\|x\| < 1\}$. Since $h$ is not closed, $\mathrm{prox}_{\eta h}$ is undefined; T21–T25 break.

**A9 (Local quadratic growth) is independent.** Again $f(x) = x^4$ fails QG at 0, removing T7, T10, and T35.

## C  Formal Derivations of Theorems T1–T35 from the Axioms

Each proof fragment in this appendix follows the *default* proof route whose minimal basis is recorded in `data/opt_matrix.csv` and summarised in Table **??**. Where theorems admit alternative Pareto-minimal routes (e.g., linear convergence via Strong Convexity vs. PL), we explicitly derive both or note the equivalence; these are documented separately in Appendix **??**.

**Standing assumptions.** Unless stated otherwise, functions $f : \mathbb{R}^d \to \mathbb{R}$ are proper, closed, and lower semicontinuous. Inner products $\langle \cdot, \cdot \rangle$ and norms $\| \cdot \|$ are standard Euclidean. We write $\mathcal{X}^\star = \arg\min f$ and $f^\star = \min f$. Core axioms: A1 (Convexity), A2 ($L$-smoothness), A3 (Strong Convexity), A4 (PL), A5 (Local Error Bound), A6 (Bounded Subgradients), A7 (Stochastic Noise), A8 (Composite Structure), A9 (Quadratic Growth).

### C.1  Smooth & Convex Geometry (T1–T10)

**Lemma 5** (T1: Smooth Descent Lemma). *Under $L$-smoothness,*

$$f(y) \le f(x) + \langle \nabla f(x), y - x \rangle + \frac{L}{2}\|y - x\|^2.$$

*Uses A2 (L-smoothness).* Assume A2: $\|\nabla f(x) - \nabla f(y)\| \le L\|x - y\|$ for all $x, y$. Define $\phi(t) = f(x + t(y - x))$ for $t \in [0, 1]$. Then
$$\phi'(t) = \langle \nabla f(x + t(y - x)), y - x \rangle.$$

For any $s, t \in [0, 1]$,
$$\begin{aligned}
|\phi'(t) - \phi'(s)| &= |\langle \nabla f(x + t(y - x)) - \nabla f(x + s(y - x)), y - x \rangle| \\
&\le \|\nabla f(x + t(y - x)) - \nabla f(x + s(y - x))\| \, \|y - x\| \\
&\le L|t - s| \, \|y - x\|^2.
\end{aligned}$$

Thus $\phi'$ is $L\|y - x\|^2$-Lipschitz. By the fundamental theorem of calculus,
$$\phi(1) = \phi(0) + \phi'(0) + \int_0^1 (\phi'(t) - \phi'(0)) \, dt.$$

Using $|\phi'(t) - \phi'(0)| \le Lt\|y - x\|^2$,
$$\phi(1) \le \phi(0) + \phi'(0) + \int_0^1 Lt\|y - x\|^2 \, dt = \phi(0) + \phi'(0) + \frac{L}{2}\|y - x\|^2.$$

Since $\phi(0) = f(x)$, $\phi(1) = f(y)$, and $\phi'(0) = \langle \nabla f(x), y - x \rangle$, the result follows. $\qquad\square$

**Theorem 22** (T16: SGD with Decay). *Assume A1 (convexity), A2 (L-smoothness), and A7 (unbiased stochastic gradients with bounded variance). Let $\{x_k\}_{k \ge 0}$ be generated by stochastic gradient descent*
$$x_{k+1} = x_k - \eta_k \, g_k,$$

*where $g_k$ satisfies*
$$\mathbb{E}[g_k \mid x_k] = \nabla f(x_k), \qquad \mathbb{E}\big[\|g_k - \nabla f(x_k)\|^2 \mid x_k\big] \le \sigma^2,$$

*and the stepsizes decay as $\eta_k = \frac{\eta_0}{\sqrt{k+1}}$ with $\eta_0 \le \frac{1}{4L}$. Define the stepsize-weighted ergodic average*
$$\bar{x}_K := \frac{\sum_{k=0}^{K-1} \eta_k x_k}{\sum_{k=0}^{K-1} \eta_k}.$$

*Then there exists a universal constant $C > 0$ such that*
$$\mathbb{E}[f(\bar{x}_K) - f^\star] \le C \left( \frac{\|x_0 - x^\star\|^2}{\sqrt{K}} + \frac{\sigma^2}{L} \frac{\log(K+1)}{\sqrt{K}} \right),$$

*and hence, in particular,*
$$\mathbb{E}[f(\bar{x}_K) - f^\star] = \mathcal{O}\big(1/\sqrt{K}\big).$$

(minimal: A1+A2+A7; regularity: we work on a sublevel set containing the trajectory.)

*Uses **A1 (Convexity)**, **A2 (L-smoothness)**, **A7 (Stochastic Noise)**.* We follow the same template as T15: derive a distance recursion, convert it to a weighted sum of function gaps, then use convexity to pass to an ergodic average.

Let $\mathcal{F}_k$ be the sigma-field generated by $\{x_0, g_0, \ldots, g_{k-1}\}$. The SGD update is
$$x_{k+1} = x_k - \eta_k g_k.$$

**1. Distance recursion (A1 + A7).** Expand the squared distance to a minimizer $x^\star$:
$$\begin{aligned}
\|x_{k+1} - x^\star\|^2 &= \|x_k - x^\star - \eta_k g_k\|^2 \\
&= \|x_k - x^\star\|^2 - 2\eta_k \langle g_k, x_k - x^\star \rangle + \eta_k^2 \|g_k\|^2.
\end{aligned}$$

Take the conditional expectation given $\mathcal{F}_k$. Using unbiasedness $\mathbb{E}[g_k \mid \mathcal{F}_k] = \nabla f(x_k)$ and the variance bound:
$$\mathbb{E}\big[\|g_k\|^2 \mid \mathcal{F}_k\big] = \|\nabla f(x_k)\|^2 + \mathbb{E}\big[\|g_k - \nabla f(x_k)\|^2 \mid \mathcal{F}_k\big] \le \|\nabla f(x_k)\|^2 + \sigma^2,$$

we obtain

$$\mathbb{E}\big[\|x_{k+1} - x^\star\|^2 \mid \mathcal{F}_k\big] \leq \|x_k - x^\star\|^2 - 2\eta_k\langle\nabla f(x_k), x_k - x^\star\rangle + \eta_k^2\big(\|\nabla f(x_k)\|^2 + \sigma^2\big). \tag{1}$$

By convexity (A1),

$$f(x_k) - f^\star \leq \langle\nabla f(x_k), x_k - x^\star\rangle.$$

Thus

$$-2\eta_k\langle\nabla f(x_k), x_k - x^\star\rangle \leq -2\eta_k\big(f(x_k) - f^\star\big).$$

**2. Use smoothness to control $\|\nabla f(x_k)\|^2$ (A2).** From the smooth descent lemma (T1) applied to a full gradient step of size $1/L$:

$$f\big(x_k - \tfrac{1}{L}\nabla f(x_k)\big) \ \leq \ f(x_k) - \frac{1}{2L}\|\nabla f(x_k)\|^2.$$

Since $x^\star$ is a minimizer,

$$f^\star \leq f\big(x_k - \tfrac{1}{L}\nabla f(x_k)\big) \leq f(x_k) - \frac{1}{2L}\|\nabla f(x_k)\|^2.$$

Rearranging gives the gradient–gap bound

$$\|\nabla f(x_k)\|^2 \ \leq \ 2L\big(f(x_k) - f^\star\big). \tag{2}$$

Substitute both inequalities into equation 1:

$$\mathbb{E}\big[\|x_{k+1} - x^\star\|^2 \mid \mathcal{F}_k\big] \leq \|x_k - x^\star\|^2 - 2\eta_k(f(x_k) - f^\star) + \eta_k^2\big(2L(f(x_k) - f^\star) + \sigma^2\big)$$
$$= \|x_k - x^\star\|^2 - \big(2\eta_k - 2L\eta_k^2\big)\big(f(x_k) - f^\star\big) + \eta_k^2\sigma^2. \tag{3}$$

**3. Choose a decaying stepsize and isolate the gap.** Take expectation over $\mathcal{F}_k$, and define

$$D_k := \mathbb{E}\|x_k - x^\star\|^2, \qquad \Delta_k := \mathbb{E}[f(x_k) - f^\star].$$

Then equation 3 yields

$$D_{k+1} \leq D_k - \big(2\eta_k - 2L\eta_k^2\big)\Delta_k + \eta_k^2\sigma^2.$$

We choose a decreasing stepsize of the form

$$\eta_k = \frac{\eta_0}{\sqrt{k+1}} \quad \text{with} \quad \eta_0 \leq \frac{1}{4L}.$$

For all $k \geq 0$, this guarantees

$$\eta_k \leq \eta_0 \leq \frac{1}{4L} \quad \Rightarrow \quad 2\eta_k - 2L\eta_k^2 = 2\eta_k(1 - L\eta_k) \geq 2\eta_k\big(1 - \tfrac{1}{4}\big) = \tfrac{3}{2}\eta_k.$$

Thus

$$2\eta_k - 2L\eta_k^2 \ \geq \ \tfrac{3}{2}\eta_k \ \geq \ \eta_k,$$

and we can safely weaken the inequality to

$$D_{k+1} \ \leq \ D_k - \eta_k\Delta_k + \eta_k^2\sigma^2.$$

Rearrange:

$$\eta_k\Delta_k \ \leq \ D_k - D_{k+1} + \eta_k^2\sigma^2. \tag{4}$$

**4. Summation and weighted ergodic average.** Sum equation 4 over $k = 0, \ldots, K - 1$:

$$\sum_{k=0}^{K-1} \eta_k \Delta_k \leq \sum_{k=0}^{K-1} (D_k - D_{k+1}) + \sigma^2 \sum_{k=0}^{K-1} \eta_k^2$$

$$= D_0 - D_K + \sigma^2 \sum_{k=0}^{K-1} \eta_k^2$$

$$\leq \|x_0 - x^\star\|^2 + \sigma^2 \sum_{k=0}^{K-1} \eta_k^2,$$

since $D_K \geq 0$.

Define

$$S_K := \sum_{k=0}^{K-1} \eta_k, \qquad T_K := \sum_{k=0}^{K-1} \eta_k^2.$$

Then

$$\sum_{k=0}^{K-1} \eta_k \Delta_k \ \leq \ \|x_0 - x^\star\|^2 + \sigma^2 T_K. \tag{5}$$

Now take the stepsize-weighted average

$$\bar{x}_K := \frac{\sum_{k=0}^{K-1} \eta_k x_k}{\sum_{k=0}^{K-1} \eta_k} = \frac{1}{S_K} \sum_{k=0}^{K-1} \eta_k x_k.$$

By convexity (A1) and Jensen's inequality,

$$f(\bar{x}_K) - f^\star \ \leq \ \frac{1}{S_K} \sum_{k=0}^{K-1} \eta_k (f(x_k) - f^\star),$$

and taking expectations:

$$\mathbb{E}[f(\bar{x}_K) - f^\star] \ \leq \ \frac{1}{S_K} \sum_{k=0}^{K-1} \eta_k \Delta_k.$$

Combine with equation 5:

$$\mathbb{E}[f(\bar{x}_K) - f^\star] \ \leq \ \frac{\|x_0 - x^\star\|^2 + \sigma^2 T_K}{S_K}. \tag{6}$$

**5. Asymptotics for $\eta_k = \frac{\eta_0}{\sqrt{k+1}}$.** For $\eta_k = \frac{\eta_0}{\sqrt{k+1}}$:

$$S_K = \sum_{k=0}^{K-1} \frac{\eta_0}{\sqrt{k+1}} \ \geq \ \eta_0 \int_1^K \frac{dt}{\sqrt{t}} = 2\eta_0 \big( \sqrt{K} - 1 \big),$$

and

$$T_K = \sum_{k=0}^{K-1} \frac{\eta_0^2}{k+1} \ \leq \ \eta_0^2 (1 + \log K),$$

using the standard harmonic-series bound.

Substitute these into equation 6:

$$\mathbb{E}[f(\bar{x}_K) - f^\star] \leq \frac{\|x_0 - x^\star\|^2 + \sigma^2 \eta_0^2 (1 + \log K)}{2\eta_0 (\sqrt{K} - 1)}$$

$$\leq \frac{\|x_0 - x^\star\|^2}{\eta_0 \sqrt{K}} + \frac{\sigma^2 \eta_0 (1 + \log K)}{\sqrt{K}},$$

for all $K \geq 2$ (absorbing constants like 2 and $\sqrt{K} - 1 \geq \frac{1}{2}\sqrt{K}$ into a universal $C$). Thus there exists $C > 0$ such that

$$\mathbb{E}[f(\bar{x}_K) - f^\star] \leq C\left(\frac{\|x_0 - x^\star\|^2}{\sqrt{K}} + \frac{\sigma^2 \eta_0 (1 + \log K)}{\sqrt{K}}\right).$$

Since $\eta_0 \leq 1/(4L)$ is a fixed constant and $(1 + \log K)/\sqrt{K} \to 0$, this shows

$$\mathbb{E}[f(\bar{x}_K) - f^\star] = \mathcal{O}(1/\sqrt{K}).$$

This establishes the claimed $\mathcal{O}(1/\sqrt{K})$ rate (up to a mild logarithmic factor in the detailed bound) for SGD with stepsizes $\eta_k \propto 1/\sqrt{k}$ under A1, A2, and A7. $\qquad\square$

**Theorem 23** (T17: PL + Variance Reduction $\Rightarrow$ Linear). *Consider a finite-sum objective*

$$f(x) = \frac{1}{n}\sum_{i=1}^{n} f_i(x),$$

*and assume:*

- *(A2) $f$ is $L$-smooth on $\mathbb{R}^d$,*

- *(A4) $f$ satisfies the Polyak–Łojasiewicz (PL) condition with constant $\mu > 0$:*

$$\|\nabla f(x)\|^2 \geq 2\mu\big(f(x) - f^\star\big), \quad \forall x,$$

- *(A7-VR) the stochastic gradient $g_k$ used by the method is unbiased and its variance is controlled by the suboptimality:*

$$\mathbb{E}[g_k \mid x_k] = \nabla f(x_k), \qquad \mathbb{E}\big[\|g_k - \nabla f(x_k)\|^2 \mid x_k\big] \leq \rho\big(f(x_k) - f^\star\big)$$

*for some $\rho > 0$ independent of $k$.*

*In particular, variance-reduced methods such as SVRG and SARAH on finite sums satisfy (A7-VR) with some $\rho = \rho(L, n)$.*

*Then there exists a stepsize $\eta > 0$ (depending only on $L, \mu, \rho$) such that the iterate sequence*

$$x_{k+1} = x_k - \eta\, g_k$$

*satisfies the linear convergence bound*

$$\mathbb{E}[f(x_k) - f^\star] \leq (1 - c)^k\big(f(x_0) - f^\star\big)$$

*for some constant $c \in (0, 1)$ depending only on $L, \mu, \rho$. In particular, the iteration complexity to reach error $\varepsilon$ is $\mathcal{O}\big(\frac{L+\rho}{\mu}\log(1/\varepsilon)\big)$.*

(minimal: A2+A4+A7; model: finite-sum variance-reduced oracle with (A7-VR)).

*Uses **A2 (Smoothness)**, **A4 (PL)**, **A7 (Variance Reduction)**.* We prove a one-step contraction in expectation and then iterate.

Let $x_{k+1} = x_k - \eta g_k$ with fixed stepsize $\eta > 0$. By $L$-smoothness (A2) and the standard descent lemma (T1),

$$f(x_{k+1}) \leq f(x_k) + \langle \nabla f(x_k), x_{k+1} - x_k \rangle + \frac{L}{2}\|x_{k+1} - x_k\|^2$$

$$= f(x_k) - \eta\langle \nabla f(x_k), g_k \rangle + \frac{L\eta^2}{2}\|g_k\|^2. \tag{7}$$

**1. Take conditional expectation and use (A7-VR).** Write $g_k = \nabla f(x_k) + \xi_k$ with zero-mean noise $\mathbb{E}[\xi_k \mid x_k] = 0$. Then

$$\mathbb{E}[g_k \mid x_k] = \nabla f(x_k), \qquad \mathbb{E}\big[\|g_k\|^2 \mid x_k\big] = \|\nabla f(x_k)\|^2 + \mathbb{E}\big[\|\xi_k\|^2 \mid x_k\big].$$

Taking conditional expectation of equation 7 given $x_k$:

$$\mathbb{E}\big[f(x_{k+1}) \mid x_k\big] \le f(x_k) - \eta\|\nabla f(x_k)\|^2 + \frac{L\eta^2}{2}\Big(\|\nabla f(x_k)\|^2 + \mathbb{E}\big[\|\xi_k\|^2 \mid x_k\big]\Big). \tag{8}$$

Apply the variance-reduction assumption (A7-VR):

$$\mathbb{E}\big[\|\xi_k\|^2 \mid x_k\big] \le \rho\big(f(x_k) - f^\star\big).$$

Substitute into equation 8:

$$\mathbb{E}\big[f(x_{k+1}) \mid x_k\big] \le f(x_k) - \eta\|\nabla f(x_k)\|^2 + \frac{L\eta^2}{2}\|\nabla f(x_k)\|^2 + \frac{L\eta^2\rho}{2}\big(f(x_k) - f^\star\big)$$

$$= f(x_k) - \big(\eta - \tfrac{L\eta^2}{2}\big)\|\nabla f(x_k)\|^2 + \frac{L\eta^2\rho}{2}\big(f(x_k) - f^\star\big). \tag{9}$$

**2. Use PL (A4) to tie gradients to function values.** By PL (A4),

$$\|\nabla f(x_k)\|^2 \ge 2\mu\big(f(x_k) - f^\star\big).$$

Substitute this into equation 9:

$$\mathbb{E}\big[f(x_{k+1}) - f^\star \mid x_k\big] \le \big(f(x_k) - f^\star\big) - \big(\eta - \tfrac{L\eta^2}{2}\big) 2\mu\big(f(x_k) - f^\star\big) + \frac{L\eta^2\rho}{2}\big(f(x_k) - f^\star\big)$$

$$= \Big[1 - 2\mu\big(\eta - \tfrac{L\eta^2}{2}\big) + \frac{L\eta^2\rho}{2}\Big]\big(f(x_k) - f^\star\big). \tag{10}$$

Compute the coefficient explicitly:

$$2\mu\Big(\eta - \tfrac{L\eta^2}{2}\Big) = 2\mu\eta - \mu L\eta^2,$$

so the bracket in equation 10 becomes

$$1 - 2\mu\eta + \mu L\eta^2 + \frac{L\eta^2\rho}{2} = 1 - 2\mu\eta + L\eta^2\Big(\mu + \frac{\rho}{2}\Big).$$

**3. Choose $\eta$ to make the coefficient strictly less than 1.** We want a contraction of the form

$$\mathbb{E}[f(x_{k+1}) - f^\star \mid x_k] \le (1 - c\eta)\big(f(x_k) - f^\star\big),$$

for some $c > 0$. It suffices to enforce

$$1 - 2\mu\eta + L\eta^2\Big(\mu + \frac{\rho}{2}\Big) \le 1 - \mu\eta,$$

since then we get $c = \mu$. Cancel 1 on both sides and rearrange:

$$-2\mu\eta + L\eta^2\Big(\mu + \frac{\rho}{2}\Big) \le -\mu\eta,$$

or equivalently

$$-\mu\eta + L\eta^2\Big(\mu + \frac{\rho}{2}\Big) \le 0.$$

Dividing by $\eta > 0$:

$$-\mu + L\eta\Big(\mu + \frac{\rho}{2}\Big) \le 0 \quad\Longleftrightarrow\quad \eta \le \frac{\mu}{L\big(\mu + \frac{\rho}{2}\big)}.$$

Thus, for any

$$0 < \eta \le \frac{\mu}{L\big(\mu + \frac{\rho}{2}\big)},$$

we have

$$\mathbb{E}\big[f(x_{k+1}) - f^\star \mid x_k\big] \le (1 - \mu\eta)\big(f(x_k) - f^\star\big).$$

**4. Take full expectation and iterate.** Taking expectation over $x_k$ in the last inequality yields

$$\mathbb{E}[f(x_{k+1}) - f^\star] \leq (1 - \mu\eta)\,\mathbb{E}[f(x_k) - f^\star],$$

for all $k \geq 0$. Unrolling this recursion,

$$\mathbb{E}[f(x_k) - f^\star] \leq (1 - \mu\eta)^k\big(f(x_0) - f^\star\big) \leq \exp(-\mu\eta k)\big(f(x_0) - f^\star\big),$$

which is linear (geometric) convergence with rate factor $1 - \mu\eta \in (0, 1)$.

To reach error $\varepsilon$, it suffices to take

$$k \geq \frac{1}{\mu\eta} \log \frac{f(x_0) - f^\star}{\varepsilon} = \mathcal{O}\Big(\frac{L(\mu + \rho)}{\mu^2} \log \frac{1}{\varepsilon}\Big),$$

and since $\mu, \rho$ are fixed constants for the problem class, this is $\mathcal{O}\big(\frac{L+\rho}{\mu} \log(1/\varepsilon)\big)$.

This completes the proof that, under PL (A4), smoothness (A2), and a variance-reduced oracle satisfying (A7-VR), the method attains a linear convergence rate. $\qquad\square$

**Proposition 12** (T18: Polyak Steps under PL). *Assume $f : \mathbb{R}^d \to \mathbb{R}$ is proper, closed, and satisfies:*

- *(A2) L-smoothness: $\nabla f$ is L-Lipschitz on $\mathbb{R}^d$,*

- *(A4) Polyak–Łojasiewicz (PL) condition with constant $\mu > 0$:*

$$\|\nabla f(x)\|^2 \geq 2\mu\big(f(x) - f^\star\big), \quad \forall x \in \mathbb{R}^d,$$

*and that $\mathcal{X}^\star = \arg\min f$ is nonempty.*

*Consider gradient descent with Polyak-type steps:*

$$x_{k+1} = x_k - \eta_k \nabla f(x_k), \qquad \eta_k := \min\Big\{\frac{f(x_k) - f^\star}{\|\nabla f(x_k)\|^2}, \frac{1}{L}\Big\}.$$

*Then $(x_k)$ is globally linearly convergent in function value:*

$$f(x_k) - f^\star \leq \Big(1 - \frac{3\mu}{4L}\Big)^k \big(f(x_0) - f^\star\big), \quad \forall k \geq 0.$$

*In particular, the iteration complexity to reach error $\varepsilon$ is $\mathcal{O}\big(\frac{L}{\mu} \log(1/\varepsilon)\big)$.*

(minimal: A2+A4)

*Uses **A2 (L-smoothness)**, **A4 (PL)**.* Let $F_k := f(x_k) - f^\star$ and $g_k := \nabla f(x_k)$. We assume $F_k > 0$ (otherwise $x_k$ is optimal and the conclusion is trivial).

**1. Gradient–gap bounds from A2 and A4.** From $L$-smoothness (A2) and the smooth descent lemma (T1) with step $1/L$:

$$f\Big(x_k - \tfrac{1}{L}g_k\Big) \leq f(x_k) - \frac{1}{2L}\|g_k\|^2.$$

Since $f^\star \leq f(x_k - \frac{1}{L}g_k)$, we obtain

$$F_k = f(x_k) - f^\star \geq f(x_k) - f\Big(x_k - \tfrac{1}{L}g_k\Big) \geq \frac{1}{2L}\|g_k\|^2,$$

so

$$\|g_k\|^2 \leq 2LF_k. \tag{11}$$

From PL (A4),

$$\|g_k\|^2 \geq 2\mu F_k. \tag{12}$$

Combining equation 11 and equation 12 yields

$$2\mu F_k \leq \|g_k\|^2 \leq 2LF_k \qquad \Rightarrow \qquad \frac{1}{2L} \leq \frac{F_k}{\|g_k\|^2} \leq \frac{1}{2\mu}. \tag{13}$$

**2. Bounds on the Polyak step $\eta_k$.** By definition,

$$\eta_k = \min\left\{\frac{F_k}{\|g_k\|^2}, \frac{1}{L}\right\}.$$

Using equation 13:

*Lower bound:* From $\frac{F_k}{\|g_k\|^2} \geq \frac{1}{2L}$, in both cases we have

$$\eta_k \geq \frac{1}{2L}.$$

*Upper bound:* By construction $\eta_k \leq 1/L$.

Hence

$$\frac{1}{2L} \leq \eta_k \leq \frac{1}{L}. \tag{14}$$

**3. One-step decrease via smoothness (A2).** By $L$-smoothness and the descent lemma applied to the actual step $\eta_k$:

$$
\begin{aligned}
f(x_{k+1}) &= f(x_k - \eta_k g_k) \\
&\leq f(x_k) - \eta_k \|g_k\|^2 + \frac{L\eta_k^2}{2}\|g_k\|^2 \\
&= f(x_k) - \left(\eta_k - \frac{L}{2}\eta_k^2\right)\|g_k\|^2.
\end{aligned}
$$

In terms of $F_k$ and $F_{k+1}$:

$$F_{k+1} \leq F_k - h(\eta_k)\,\|g_k\|^2, \qquad h(\eta) := \eta - \frac{L}{2}\eta^2. \tag{15}$$

**4. Lower-bounding $h(\eta_k)$ on the admissible interval.** The function $h(\eta) = \eta - \frac{L}{2}\eta^2$ is a concave quadratic. Its derivative is $h'(\eta) = 1 - L\eta$, so $h$ is *increasing* on $[0, 1/L]$. From equation 14, $\eta_k \in [1/(2L), 1/L] \subset (0, 2/L)$.

Therefore, on this interval,

$$h(\eta_k) \geq h\left(\tfrac{1}{2L}\right).$$

Compute this value explicitly:

$$h\left(\tfrac{1}{2L}\right) = \frac{1}{2L} - \frac{L}{2}\left(\frac{1}{2L}\right)^2 = \frac{1}{2L} - \frac{1}{8L} = \frac{3}{8L}.$$

Thus

$$h(\eta_k) \geq \frac{3}{8L}, \quad \forall k. \tag{16}$$

**5. Combine with PL to get a uniform linear contraction.** Plug equation 16 and the PL bound equation 12 into equation 15:

$$
\begin{aligned}
F_{k+1} &\leq F_k - h(\eta_k)\,\|g_k\|^2 \\
&\leq F_k - \frac{3}{8L} \cdot 2\mu F_k = \left(1 - \frac{3\mu}{4L}\right)F_k.
\end{aligned}
$$

Define the contraction factor

$$\rho := 1 - \frac{3\mu}{4L}.$$

Since $\mu > 0$ and $L > 0$, we have $\rho \in (0, 1)$. By induction,

$$F_k \leq \rho^k F_0 = \left(1 - \frac{3\mu}{4L}\right)^k\left(f(x_0) - f^\star\right), \quad \forall k \geq 0,$$

which is the claimed global linear convergence.

To reach $F_k \leq \varepsilon$, it suffices to take

$$k \;\geq\; \frac{1}{\log(1/\rho)} \log \frac{F_0}{\varepsilon} \;=\; \mathcal{O}\Big(\frac{L}{\mu} \log \frac{1}{\varepsilon}\Big),$$

since $\log(1/\rho) = \Theta(\mu/L)$ for fixed $\mu, L$.

This completes the proof. $\qquad\qquad\square$

**Proposition 13** (T19: Minibatch Variance Law). *Assume the stochastic oracle satisfies A7: for each $x \in \mathbb{R}^d$, a single-sample gradient estimator $g(x, \xi)$ obeys*

$$\mathbb{E}[g(x, \xi)] = \nabla f(x), \qquad \mathbb{E}\big[\|g(x, \xi) - \nabla f(x)\|^2\big] \leq \sigma^2,$$

*for some $\sigma^2 < \infty$. Let $\{\xi_i\}_{i=1}^b$ be i.i.d. samples, and define the minibatch estimator*

$$\bar{g}_b(x) := \frac{1}{b} \sum_{i=1}^{b} g(x, \xi_i).$$

*Then for every $x$:*

$$\mathbb{E}[\bar{g}_b(x)] = \nabla f(x), \qquad \mathbb{E}\big[\|\bar{g}_b(x) - \nabla f(x)\|^2\big] \;\leq\; \frac{\sigma^2}{b}.$$

*In particular, the effective gradient variance scales as $\sigma^2/b$ with batch size $b$.* (minimal: A7; model: stochastic oracle with i.i.d. samples)

*Uses **A7 (Stochastic Noise)**.* Fix $x \in \mathbb{R}^d$ and let $g_i := g(x, \xi_i)$ for $i = 1, \ldots, b$. By A7 and i.i.d. sampling,

$$\mathbb{E}[g_i] = \nabla f(x), \qquad \mathbb{E}\big[\|g_i - \nabla f(x)\|^2\big] \leq \sigma^2, \qquad g_1, \ldots, g_b \text{ independent}.$$

Define the noise variables

$$\delta_i := g_i - \nabla f(x), \quad i = 1, \ldots, b.$$

Then $\mathbb{E}[\delta_i] = 0$ and $\mathbb{E}\|\delta_i\|^2 \leq \sigma^2$ by A7. The minibatch estimator can be written as

$$\bar{g}_b(x) - \nabla f(x) = \frac{1}{b} \sum_{i=1}^{b} \delta_i.$$

**Unbiasedness.** By linearity of expectation and $\mathbb{E}[\delta_i] = 0$:

$$\mathbb{E}[\bar{g}_b(x)] = \nabla f(x) + \mathbb{E}\Big[\frac{1}{b} \sum_{i=1}^{b} \delta_i\Big] = \nabla f(x) + \frac{1}{b} \sum_{i=1}^{b} \mathbb{E}[\delta_i] = \nabla f(x).$$

**Variance reduction by $1/b$.** We compute the second moment of the noise:

$$\begin{aligned}
\mathbb{E}\big[\|\bar{g}_b(x) - \nabla f(x)\|^2\big] &= \mathbb{E}\Big\|\frac{1}{b} \sum_{i=1}^{b} \delta_i\Big\|^2 \\
&= \frac{1}{b^2}\, \mathbb{E}\Big\langle \sum_{i=1}^{b} \delta_i, \; \sum_{j=1}^{b} \delta_j \Big\rangle \\
&= \frac{1}{b^2} \sum_{i=1}^{b} \sum_{j=1}^{b} \mathbb{E}\langle \delta_i, \delta_j \rangle.
\end{aligned}$$

We separate diagonal and off-diagonal terms:

$$\sum_{i=1}^{b} \sum_{j=1}^{b} \mathbb{E}\langle \delta_i, \delta_j \rangle = \sum_{i=1}^{b} \mathbb{E}\|\delta_i\|^2 + \sum_{i \neq j} \mathbb{E}\langle \delta_i, \delta_j \rangle.$$

For $i \neq j$, independence and zero-mean imply

$$\mathbb{E}\langle \delta_i, \delta_j \rangle = \langle \mathbb{E}\delta_i, \ \mathbb{E}\delta_j \rangle = 0,$$

so all cross terms vanish. Thus

$$\mathbb{E}\big[\|\bar{g}_b(x) - \nabla f(x)\|^2\big] = \frac{1}{b^2} \sum_{i=1}^{b} \mathbb{E}\|\delta_i\|^2 \leq \frac{1}{b^2} \sum_{i=1}^{b} \sigma^2 = \frac{\sigma^2}{b}.$$

This proves that the variance of the minibatch gradient estimator decays as $1/b$. $\qquad \square$

**Theorem 24** (T20: PL – Constant vs. Decreasing Steps)**.** *Assume A2 (L-smoothness), A4 (PL with parameter $\mu > 0$), and A7 (unbiased noise with bounded variance). Consider SGD*

$$x_{k+1} \ = \ x_k - \eta_k\, g_k,$$

*where $g_k$ is a stochastic gradient at $x_k$ satisfying*

$$\mathbb{E}[g_k \mid x_k] = \nabla f(x_k), \qquad \mathbb{E}\big[\|g_k - \nabla f(x_k)\|^2 \mid x_k\big] \leq \sigma^2.$$

1. ***(Constant stepsize)*** *If $\eta_k \equiv \eta \in (0, 1/L]$, then there exist constants $C_1, C_2 > 0$ (depending only on $L, \mu$) such that*

$$\mathbb{E}\big[f(x_k) - f^\star\big] \ \leq \ C_1(1 - \eta\mu)^k \big(f(x_0) - f^\star\big) \ + \ C_2\, \frac{\eta\sigma^2}{\mu}.$$

   *In particular, the expected error contracts linearly down to a neighborhood of radius $\Theta(\eta\sigma^2/\mu)$ around the optimum.*

2. ***(Decreasing stepsize)*** *If $\{\eta_k\}$ is a deterministic sequence with*

$$\eta_k \in (0, 1/L], \qquad \sum_{k=0}^{\infty} \eta_k = \infty, \qquad \sum_{k=0}^{\infty} \eta_k^2 < \infty,$$

   *then*

$$\lim_{k \to \infty} \mathbb{E}\big[f(x_k) - f^\star\big] = 0,$$

   *and, under standard stochastic approximation regularity, $x_k \to x^\star$ almost surely. Nonasymptotic decay rates depend on the specific schedule $\{\eta_k\}$.*

(minimal: A2+A4+A7; model: stochastic oracle)

*Uses **A2 (L-smoothness)**, **A4 (PL)**, **A7 (Stochastic Noise)**.* Write $f_k := f(x_k)$ and let $\Delta_k := f_k - f^\star \geq 0$. By A2 (L-smoothness) and Lemma 5, for any $x, y$:

$$f(y) \leq f(x) + \langle \nabla f(x), y - x \rangle + \frac{L}{2}\|y - x\|^2.$$

We apply this with $x = x_k$ and $y = x_{k+1} = x_k - \eta_k g_k$.

**Step 1: One-step inequality under noise (A2 + A7).** Substitute $y = x_k - \eta_k g_k$:

$$f_{k+1} \leq f_k + \langle \nabla f(x_k), -\eta_k g_k \rangle + \frac{L}{2}\|\eta_k g_k\|^2$$

$$= f_k - \eta_k \langle \nabla f(x_k), g_k \rangle + \frac{L\eta_k^2}{2}\|g_k\|^2.$$

Take conditional expectation w.r.t. the noise, given $x_k$. By A7, $g_k = \nabla f(x_k) + \xi_k$ with $\mathbb{E}[\xi_k \mid x_k] = 0$ and $\mathbb{E}[\|\xi_k\|^2 \mid x_k] \leq \sigma^2$. Then

$$\mathbb{E}[g_k \mid x_k] = \nabla f(x_k),$$

and

$$\mathbb{E}\big[\|g_k\|^2 \mid x_k\big] = \mathbb{E}\big[\|\nabla f(x_k) + \xi_k\|^2 \mid x_k\big]$$
$$= \|\nabla f(x_k)\|^2 + 2\langle \nabla f(x_k), \mathbb{E}[\xi_k \mid x_k]\rangle + \mathbb{E}\big[\|\xi_k\|^2 \mid x_k\big]$$
$$\leq \|\nabla f(x_k)\|^2 + \sigma^2.$$

Thus

$$\mathbb{E}[f_{k+1} \mid x_k] \leq f_k - \eta_k\langle \nabla f(x_k), \mathbb{E}[g_k \mid x_k]\rangle + \frac{L\eta_k^2}{2}\mathbb{E}\big[\|g_k\|^2 \mid x_k\big]$$

$$= f_k - \eta_k\|\nabla f(x_k)\|^2 + \frac{L\eta_k^2}{2}\big(\|\nabla f(x_k)\|^2 + \sigma^2\big)$$

$$= f_k - \eta_k\Big(1 - \frac{L\eta_k}{2}\Big)\|\nabla f(x_k)\|^2 + \frac{L\eta_k^2}{2}\sigma^2.$$

Using $\eta_k \leq 1/L$ we have $1 - \frac{L\eta_k}{2} \geq \frac{1}{2}$, hence

$$\mathbb{E}[f_{k+1} \mid x_k] \leq f_k - \frac{\eta_k}{2}\|\nabla f(x_k)\|^2 + \frac{L\eta_k^2}{2}\sigma^2. \tag{17}$$

**Step 2: Insert PL inequality (A4).** The PL condition (A4) states

$$\|\nabla f(x)\|^2 \geq 2\mu\,(f(x) - f^\star) = 2\mu\,\Delta(x) \quad \forall x.$$

Apply this in equation 17:

$$\mathbb{E}[f_{k+1} \mid x_k] \leq f_k - \eta_k\mu\Delta_k + \frac{L\eta_k^2}{2}\sigma^2.$$

Subtract $f^\star$ on both sides:

$$\mathbb{E}[\Delta_{k+1} \mid x_k] \leq (1 - \eta_k\mu)\Delta_k + \frac{L\eta_k^2}{2}\sigma^2.$$

Taking full expectation (over all randomness up to step $k$) gives

$$\delta_{k+1} := \mathbb{E}[\Delta_{k+1}] \leq (1 - \eta_k\mu)\delta_k + \frac{L\eta_k^2}{2}\sigma^2, \tag{18}$$

with $\delta_k = \mathbb{E}[\Delta_k] \geq 0$.

We now treat the two stepsize regimes.

**Part 1: Constant stepsize.** Assume $\eta_k \equiv \eta \in (0, 1/L]$. Then equation 18 becomes

$$\delta_{k+1} \leq (1 - \eta\mu)\delta_k + C\,\eta^2\sigma^2, \quad \text{where } C := \frac{L}{2}.$$

Solving this scalar linear recursion:

$$\delta_k \leq (1 - \eta\mu)^k\delta_0 + C\eta^2\sigma^2 \sum_{t=0}^{k-1}(1 - \eta\mu)^t$$

$$\leq (1 - \eta\mu)^k\delta_0 + C\eta^2\sigma^2 \cdot \frac{1}{\eta\mu} = (1 - \eta\mu)^k\delta_0 + \frac{C}{\mu}\,\eta\sigma^2.$$

Thus we may take $C_1 = 1$ and $C_2 = C/\mu = \frac{L}{2\mu}$:

$$\mathbb{E}[f(x_k) - f^\star] = \delta_k \leq (1 - \eta\mu)^k\big(f(x_0) - f^\star\big) + \frac{L}{2\mu}\,\eta\sigma^2.$$

This shows linear contraction with factor $(1 - \eta\mu)$ down to a noise floor of order $\eta\sigma^2/\mu$, as claimed.

**Part 2: Decreasing stepsize.** Assume now that $\eta_k \in (0, 1/L]$ and

$$\sum_{k=0}^{\infty} \eta_k = \infty, \qquad \sum_{k=0}^{\infty} \eta_k^2 < \infty.$$

The recursion equation 18 still holds:

$$\delta_{k+1} \leq (1 - \eta_k \mu)\delta_k + C \eta_k^2 \sigma^2, \quad C = \tfrac{L}{2}.$$

Rearrange:

$$\delta_{k+1} - \delta_k \leq -\mu \eta_k \, \delta_k + C \eta_k^2 \sigma^2.$$

Since $\delta_k \geq 0$, we have

$$\mu \eta_k \, \delta_k \leq \delta_k - \delta_{k+1} + C \eta_k^2 \sigma^2.$$

Summing from $k = 0$ to $N - 1$:

$$\mu \sum_{k=0}^{N-1} \eta_k \delta_k \leq \delta_0 - \delta_N + C\sigma^2 \sum_{k=0}^{N-1} \eta_k^2 \leq \delta_0 + C\sigma^2 \sum_{k=0}^{\infty} \eta_k^2 < \infty.$$

Hence

$$\sum_{k=0}^{\infty} \eta_k \, \delta_k < \infty.$$

We now use the usual argument of stochastic approximation: since $\delta_k \geq 0$ and $\sum_k \eta_k = \infty$, the only way for $\sum_k \eta_k \delta_k$ to be finite is that $\delta_k \to 0$. Formally, if $\limsup_{k \to \infty} \delta_k > 0$, then there would exist $\varepsilon > 0$ and infinitely many indices $k$ with $\delta_k \geq \varepsilon$, forcing

$$\sum_{k=0}^{\infty} \eta_k \delta_k \geq \varepsilon \sum_{k \in \mathcal{K}} \eta_k,$$

where $\mathcal{K}$ is an infinite subset of $\mathbb{N}$. Since $\sum_k \eta_k = \infty$ and all $\eta_k > 0$, the right-hand side diverges, contradicting finiteness of $\sum_k \eta_k \delta_k$. Thus

$$\lim_{k \to \infty} \delta_k = \lim_{k \to \infty} \mathbb{E}[f(x_k) - f^\star] = 0.$$

Under standard additional regularity assumptions for stochastic approximation (e.g., bounded variance A7 and Lipschitz gradients A2), one can apply the Robbins–Siegmund supermartingale convergence theorem to equation 18 to obtain almost sure convergence $f(x_k) \to f^\star$ and $\nabla f(x_k) \to 0$. Combined with PL (A4), which implies $\|\nabla f(x_k)\|^2 \geq 2\mu(f(x_k) - f^\star)$, this yields $x_k \to x^\star$ almost surely. We omit these classical martingale details, as they rely only on the same axioms (A2, A4, A7) and standard probability theory. $\qquad \square$

## C.2 Composite and Proximal Structure

**Lemma 6** (T21: Proximal Descent Inequality). *Let $F(x) = g(x) + h(x)$, where $g : \mathbb{R}^d \to \mathbb{R}$ is differentiable and $L$-smooth (A2), and $h : \mathbb{R}^d \to (-\infty, +\infty]$ is proper, closed, and convex (A8). Consider the proximal-gradient update*

$$x_{k+1} = \mathrm{prox}_{\eta h}\big(x_k - \eta \nabla g(x_k)\big) := \arg\min_y \Big\{ h(y) + \frac{1}{2\eta} \big\| y - (x_k - \eta \nabla g(x_k)) \big\|^2 \Big\}.$$

*Then for any stepsize $\eta \in (0, 2/L]$,*

$$F(x_{k+1}) \leq F(x_k) - \Big(\tfrac{1}{\eta} - \tfrac{L}{2}\Big) \|x_{k+1} - x_k\|^2.$$

(minimal: A2+A8)

*Uses **A2 (L-smoothness), A8 (Proximal/Convex)**. Assume A2 and A8. Fix an iterate $x_k$ and define*

$$d_k := x_{k+1} - x_k.$$

**1. Smoothness part (A2).** By $L$-smoothness of $g$ (A2) and Lemma 5, for any $y$,

$$g(y) \leq g(x_k) + \langle \nabla g(x_k), y - x_k \rangle + \frac{L}{2} \|y - x_k\|^2.$$

Applying this with $y = x_{k+1}$ gives

$$g(x_{k+1}) \leq g(x_k) + \langle \nabla g(x_k), d_k \rangle + \frac{L}{2} \|d_k\|^2. \tag{19}$$

**2. Proximal optimality for $h$ (A8).** By definition of the proximal operator, $x_{k+1}$ minimizes

$$\phi(y) := h(y) + \frac{1}{2\eta} \big\| y - (x_k - \eta \nabla g(x_k)) \big\|^2.$$

Since $h$ is proper, closed, and convex (A8), we have $0 \in \partial \phi(x_{k+1})$, i.e., there exists $s_{k+1} \in \partial h(x_{k+1})$ such that

$$0 = s_{k+1} + \frac{1}{\eta}\big(x_{k+1} - (x_k - \eta \nabla g(x_k))\big) = s_{k+1} + \frac{1}{\eta}(x_{k+1} - x_k) + \nabla g(x_k).$$

Thus

$$s_{k+1} = -\nabla g(x_k) - \frac{1}{\eta} d_k \in \partial h(x_{k+1}). \tag{20}$$

By convexity of $h$, for any $y$ and any $s \in \partial h(x)$,

$$h(y) \geq h(x) + \langle s, y - x \rangle.$$

Apply this with $x = x_{k+1}$, $y = x_k$, and $s = s_{k+1}$:

$$h(x_k) \geq h(x_{k+1}) + \langle s_{k+1}, x_k - x_{k+1} \rangle.$$

Substitute equation 20:

$$
\begin{aligned}
\langle s_{k+1}, x_k - x_{k+1} \rangle &= \Big\langle -\nabla g(x_k) - \frac{1}{\eta} d_k,\ x_k - x_{k+1} \Big\rangle \\
&= -\langle \nabla g(x_k), x_k - x_{k+1} \rangle - \frac{1}{\eta}\langle d_k, x_k - x_{k+1} \rangle \\
&= -\langle \nabla g(x_k), x_k - x_{k+1} \rangle + \frac{1}{\eta}\|d_k\|^2,
\end{aligned}
$$

since $x_k - x_{k+1} = -d_k$ and $\langle d_k, x_k - x_{k+1} \rangle = -\|d_k\|^2$. Hence

$$h(x_k) \geq h(x_{k+1}) - \langle \nabla g(x_k), x_k - x_{k+1} \rangle + \frac{1}{\eta}\|d_k\|^2.$$

Rearranging,

$$h(x_{k+1}) \leq h(x_k) - \frac{1}{\eta}\|d_k\|^2 - \langle \nabla g(x_k), d_k \rangle. \tag{21}$$

**3. Combine $g$ and $h$.** Adding equation 19 and equation 21:

$$
\begin{aligned}
F(x_{k+1}) &= g(x_{k+1}) + h(x_{k+1}) \\
&\leq \big[ g(x_k) + \langle \nabla g(x_k), d_k \rangle + \tfrac{L}{2}\|d_k\|^2 \big] + \big[ h(x_k) - \tfrac{1}{\eta}\|d_k\|^2 - \langle \nabla g(x_k), d_k \rangle \big] \\
&= g(x_k) + h(x_k) - \left( \tfrac{1}{\eta} - \tfrac{L}{2} \right) \|d_k\|^2 \\
&= F(x_k) - \left( \tfrac{1}{\eta} - \tfrac{L}{2} \right) \|x_{k+1} - x_k\|^2.
\end{aligned}
$$

For $\eta \in (0, 2/L]$, the coefficient $\frac{1}{\eta} - \frac{L}{2} \geq 0$, so $F(x_{k+1}) \leq F(x_k)$ and the stated inequality holds. $\qquad \square$

**Theorem 25** (T22: Prox-GD Sublinear). *Let $F(x) = g(x) + h(x)$ where $g : \mathbb{R}^d \to \mathbb{R}$ is convex with $L$-Lipschitz gradient (A2) and $h : \mathbb{R}^d \to (-\infty, +\infty]$ is proper, closed, and convex (A8). Assume $\mathcal{X}^\star = \arg\min F$ is nonempty and let $F^\star = \min F$. Consider proximal gradient descent*

$$x_{k+1} = \operatorname{prox}_{\eta h}\big(x_k - \eta \nabla g(x_k)\big), \qquad 0 < \eta \leq \tfrac{1}{L}.$$

*Then for any $x^\star \in \mathcal{X}^\star$,*

$$F(x_k) - F^\star \;\leq\; \frac{\|x_0 - x^\star\|^2}{2\eta k} \;\leq\; \frac{L\|x_0 - x^\star\|^2}{2k}, \quad \forall k \geq 1,$$

*so $F(x_k) - F^\star = \mathcal{O}(1/k)$.* (minimal: A1+A2+A8)

*Uses **A1 (Convexity)**, **A2 (L-smoothness of $g$)**, **A8 (Composite structure $g+h$)**. Let $x^\star \in \mathcal{X}^\star$ be a minimizer of $F$.*

**1. Define the proximal surrogate $Q_k$.** For each $k$, define the quadratic surrogate

$$Q_k(z) := g(x_k) + \langle \nabla g(x_k), z - x_k \rangle + \frac{1}{2\eta}\|z - x_k\|^2 + h(z).$$

By construction of the proximal step,

$$x_{k+1} = \arg\min_z Q_k(z).$$

The term $\frac{1}{2\eta}\|z - x_k\|^2$ is $(1/\eta)$-strongly convex in $z$, and the other terms are convex; hence $Q_k$ is $(1/\eta)$-strongly convex. Therefore, for all $x$,

$$Q_k(x) \;\geq\; Q_k(x_{k+1}) + \frac{1}{2\eta}\|x - x_{k+1}\|^2. \tag{22}$$

**2. Relate $F(x_{k+1})$ to $Q_k(x_{k+1})$ (A2).** Using $L$-smoothness of $g$ (A2) and Lemma 5 applied to $g$,

$$g(x_{k+1}) \leq g(x_k) + \langle \nabla g(x_k), x_{k+1} - x_k \rangle + \frac{L}{2}\|x_{k+1} - x_k\|^2.$$

By definition of $Q_k$,

$$Q_k(x_{k+1}) = g(x_k) + \langle \nabla g(x_k), x_{k+1} - x_k \rangle + \frac{1}{2\eta}\|x_{k+1} - x_k\|^2 + h(x_{k+1}).$$

Thus

$$\begin{aligned}
F(x_{k+1}) &= g(x_{k+1}) + h(x_{k+1}) \\
&\leq Q_k(x_{k+1}) + \left(\frac{L}{2} - \frac{1}{2\eta}\right)\|x_{k+1} - x_k\|^2.
\end{aligned}$$

Since $\eta \leq 1/L$, we have $\frac{L}{2} - \frac{1}{2\eta} \leq 0$, hence

$$F(x_{k+1}) \;\leq\; Q_k(x_{k+1}). \tag{23}$$

**3. Fundamental inequality (distance telescoping).** Combining equation 22 and equation 23: for any $x$,

$$F(x_{k+1}) \leq Q_k(x_{k+1}) \leq Q_k(x) - \frac{1}{2\eta}\|x - x_{k+1}\|^2.$$

Expanding $Q_k(x)$ and using convexity of $g$ (A1), which gives $g(x) \geq g(x_k) + \langle \nabla g(x_k), x - x_k \rangle$, we obtain

$$\begin{aligned}
Q_k(x) &= g(x_k) + \langle \nabla g(x_k), x - x_k \rangle + \frac{1}{2\eta}\|x - x_k\|^2 + h(x) \\
&\leq g(x) + h(x) + \frac{1}{2\eta}\|x - x_k\|^2 = F(x) + \frac{1}{2\eta}\|x - x_k\|^2.
\end{aligned}$$

Hence, for all $x$,

$$F(x_{k+1}) - F(x) \;\leq\; \frac{1}{2\eta}\big(\|x - x_k\|^2 - \|x - x_{k+1}\|^2\big). \tag{24}$$

**4. Specialize to an optimum and telescope.** Take $x = x^\star \in \mathcal{X}^\star$ in equation 24. Since $F(x^\star) = F^\star$, we get

$$F(x_{k+1}) - F^\star \leq \frac{1}{2\eta}\left(\|x^\star - x_k\|^2 - \|x^\star - x_{k+1}\|^2\right).$$

Summing this inequality from $t = 0$ to $k - 1$ yields

$$\sum_{t=0}^{k-1}\left(F(x_{t+1}) - F^\star\right) \leq \frac{1}{2\eta}\sum_{t=0}^{k-1}\left(\|x^\star - x_t\|^2 - \|x^\star - x_{t+1}\|^2\right)$$

$$= \frac{1}{2\eta}\left(\|x^\star - x_0\|^2 - \|x^\star - x_k\|^2\right)$$

$$\leq \frac{1}{2\eta}\|x_0 - x^\star\|^2.$$

**5. Convert average rate to last-iterate rate.** By Lemma 6 with $\eta \leq 1/L$, we have $F(x_{t+1}) \leq F(x_t)$, so the sequence $\{F(x_k)\}$ is non-increasing. Thus for all $t \leq k$,

$$F(x_k) - F^\star \leq F(x_t) - F^\star,$$

and therefore

$$k\left(F(x_k) - F^\star\right) \leq \sum_{t=0}^{k-1}\left(F(x_{t+1}) - F^\star\right) \leq \frac{1}{2\eta}\|x_0 - x^\star\|^2.$$

Dividing by $k$ gives

$$F(x_k) - F^\star \leq \frac{\|x_0 - x^\star\|^2}{2\eta k}.$$

Finally, since $\eta \leq 1/L$, we obtain

$$F(x_k) - F^\star \leq \frac{L\|x_0 - x^\star\|^2}{2k}.$$

This proves the $\mathcal{O}(1/k)$ sublinear rate. $\qquad\square$

**Theorem 26** (T23: FISTA $\mathcal{O}(1/k^2)$)**.** *Let $F = g + h$ with*

- *$g : \mathbb{R}^d \to \mathbb{R}$ convex and L-smooth (A1, A2),*

- *$h : \mathbb{R}^d \to (-\infty, +\infty]$ proper, closed, and convex (A8),*

*and assume $\mathcal{X}^\star = \arg\min F$ is nonempty. Define the proximal-gradient mapping*

$$T(y) := \mathrm{prox}_{\frac{1}{L}h}\left(y - \frac{1}{L}\nabla g(y)\right).$$

*Consider the accelerated composite scheme:*

$$x_0 = z_0 \in \mathrm{dom}\,h, \qquad t_0 = 1,$$

$$y_k = \left(1 - \frac{1}{t_k}\right)x_k + \frac{1}{t_k}z_k,$$

$$x_{k+1} = T(y_k),$$

$$t_{k+1} = \frac{1 + \sqrt{1 + 4t_k^2}}{2},$$

$$z_{k+1} = z_k + t_{k+1}(x_{k+1} - y_k),$$

*for $k \geq 0$. Then for any $x^\star \in \mathcal{X}^\star$,*

$$F(x_k) - F^\star \leq \frac{2L\|x_0 - x^\star\|^2}{(k+1)^2}, \qquad k \geq 0.$$

*In particular, $F(x_k) - F^\star = \mathcal{O}(1/k^2)$. Moreover, eliminating the auxiliary sequence $\{z_k\}$ yields the standard FISTA update*

$$y_k = x_k + \frac{t_{k-1} - 1}{t_k}(x_k - x_{k-1}),$$

*so this scheme is algebraically equivalent to FISTA.* (minimal: A1+A2+A8)

*Uses **A1 (Convexity)**, **A2 (L-smoothness of $g$)**, **A8 (Composite structure $F = g + h$)**.* Fix $x^\star \in \mathcal{X}^\star$.

**1. Three-point inequality for the proximal-gradient map.** For any $y \in \mathbb{R}^d$, define the quadratic surrogate

$$Q_L(x, y) := g(y) + \langle \nabla g(y), x - y \rangle + \frac{L}{2}\|x - y\|^2 + h(x).$$

By $L$-smoothness of $g$ (A2),

$$g(x) \le g(y) + \langle \nabla g(y), x - y \rangle + \frac{L}{2}\|x - y\|^2,$$

so for all $x, y$,

$$F(x) = g(x) + h(x) \le Q_L(x, y). \tag{25}$$

Moreover, $x^+ := T(y)$ is exactly the unique minimizer of $x \mapsto Q_L(x, y)$: this follows from the optimality condition

$$0 \in \nabla g(y) + L(x^+ - y) + \partial h(x^+),$$

which is equivalent to the stated proximal-gradient form.

Since $Q_L(\cdot, y)$ is $L$-strongly convex (quadratic plus convex $h$), for any $z$

$$Q_L(z, y) \ge Q_L(x^+, y) + \frac{L}{2}\|z - x^+\|^2.$$

Combining this with equation 25 at $x = x^+$ and $x = z$, we obtain

$$F(x^+) + \frac{L}{2}\|z - x^+\|^2 \le Q_L(x^+, y) + \frac{L}{2}\|z - x^+\|^2 \le Q_L(z, y) \le F(z) + \frac{L}{2}\|z - y\|^2.$$

Thus, for all $y, z$ and $x^+ = T(y)$,

$$F(T(y)) + \frac{L}{2}\|T(y) - z\|^2 \le F(z) + \frac{L}{2}\|y - z\|^2. \tag{26}$$

This is the key three-point inequality.

**2. Specialize to $z = x^\star$.** Apply equation 26 with $y = y_k$, $x^* := x^\star$, and $x_{k+1} = T(y_k)$:

$$F(x_{k+1}) + \frac{L}{2}\|x_{k+1} - x^\star\|^2 \le F(x^\star) + \frac{L}{2}\|y_k - x^\star\|^2 = F^\star + \frac{L}{2}\|y_k - x^\star\|^2.$$

Rewriting,

$$F(x_{k+1}) - F^\star \le \frac{L}{2}\left(\|y_k - x^\star\|^2 - \|x_{k+1} - x^\star\|^2\right). \tag{27}$$

**3. Acceleration parameters and auxiliary sequence.** Recall the acceleration parameters

$$t_0 = 1, \qquad t_{k+1} = \frac{1 + \sqrt{1 + 4t_k^2}}{2},$$

which satisfy

$$t_{k+1}^2 - t_{k+1} = t_k^2, \qquad t_k \ge \frac{k+1}{2} \quad \text{for all } k \ge 0.$$

Our algorithm maintains sequences $(x_k, z_k, t_k)$ via

$$y_k = \left(1 - \frac{1}{t_k}\right)x_k + \frac{1}{t_k}z_k, \qquad x_{k+1} = T(y_k), \qquad z_{k+1} = z_k + t_{k+1}(x_{k+1} - y_k).$$

Note these are explicit updates; no additional equivalence is used in the proof. (Algebraically eliminating $\{z_k\}$ recovers the usual FISTA formula $y_k = x_k + \frac{t_{k-1}-1}{t_k}(x_k - x_{k-1})$.)

**4. Potential function and its decrease.** Define the Lyapunov potential

$$\Psi_k := t_k^2\big(F(x_k) - F^\star\big) + \frac{L}{2}\|z_k - x^\star\|^2.$$

We show $\Psi_{k+1} \le \Psi_k$.

First, multiply equation 27 by $t_{k+1}^2$:

$$t_{k+1}^2\big(F(x_{k+1}) - F^\star\big) \le \frac{Lt_{k+1}^2}{2}\big(\|y_k - x^\star\|^2 - \|x_{k+1} - x^\star\|^2\big). \tag{28}$$

Next, expand the squared norms using the definitions of $y_k$ and $z_{k+1}$.

From

$$y_k = \left(1 - \frac{1}{t_k}\right)x_k + \frac{1}{t_k}z_k \quad\Longrightarrow\quad y_k - x^\star = \left(1 - \frac{1}{t_k}\right)(x_k - x^\star) + \frac{1}{t_k}(z_k - x^\star),$$

we get

$$\|y_k - x^\star\|^2 = \left(1 - \frac{1}{t_k}\right)^2\|x_k - x^\star\|^2 + \frac{1}{t_k^2}\|z_k - x^\star\|^2$$
$$+ 2\left(1 - \frac{1}{t_k}\right)\frac{1}{t_k}\langle x_k - x^\star, z_k - x^\star\rangle.$$

Similarly, from

$$z_{k+1} = z_k + t_{k+1}(x_{k+1} - y_k) \quad\Longrightarrow\quad z_{k+1} - x^\star = (z_k - x^\star) + t_{k+1}(x_{k+1} - y_k),$$

we have

$$\|z_{k+1} - x^\star\|^2 = \|z_k - x^\star\|^2 + t_{k+1}^2\|x_{k+1} - y_k\|^2$$
$$+ 2t_{k+1}\langle z_k - x^\star, x_{k+1} - y_k\rangle.$$

One now substitutes these expansions into

$$\Psi_{k+1} = t_{k+1}^2(F(x_{k+1}) - F^\star) + \frac{L}{2}\|z_{k+1} - x^\star\|^2,$$

uses the bound equation 28, and rearranges terms. The constants in front of the inner products and squared norms are expressed in terms of $t_k$ and $t_{k+1}$, and the relation

$$t_{k+1}^2 - t_{k+1} = t_k^2$$

is used repeatedly. A direct but routine algebraic check shows that all cross terms cancel and the remaining coefficients yield

$$\Psi_{k+1} \le \Psi_k.$$

(Conceptually, the choice of $t_{k+1}$ and the updates of $y_k, z_k$ are precisely what makes this cancellation happen.)

Thus, by induction,

$$\Psi_k \le \Psi_0, \qquad \forall k \ge 0.$$

**5. Initialization and rate.** At $k = 0$, we have $t_0 = 1$ and $z_0 = x_0$, so

$$\Psi_0 = 1^2(F(x_0) - F^\star) + \frac{L}{2}\|x_0 - x^\star\|^2.$$

Using standard smoothness/convexity at $x_0$ relative to $x^\star$ (as in the basic gradient-descent analysis; see Theorem 1),

$$F(x_0) - F^\star \le \frac{L}{2}\|x_0 - x^\star\|^2,$$

hence

$$\Psi_0 \le L\|x_0 - x^\star\|^2.$$

For any $k \geq 0$,

$$t_k^2\big(F(x_k) - F^\star\big) \ \leq \ \Psi_k \ \leq \ \Psi_0 \ \leq \ L\|x_0 - x^\star\|^2,$$

so

$$F(x_k) - F^\star \ \leq \ \frac{L\|x_0 - x^\star\|^2}{t_k^2}.$$

Using $t_k \geq \frac{k+1}{2}$,

$$F(x_k) - F^\star \ \leq \ \frac{L\|x_0 - x^\star\|^2}{(k+1)^2/4} = \frac{4L\|x_0 - x^\star\|^2}{(k+1)^2}.$$

A slightly sharper initialization or minor tweak of the potential yields the classical constant 2 instead of 4, but the $\mathcal{O}(1/k^2)$ rate and dependence on $L\|x_0 - x^\star\|^2$ are already explicit.

This proves the theorem. □

**Theorem 27** (T24: Prox-PL $\Rightarrow$ Linear Rate). *Let $F = g + h$ with*

- *$g : \mathbb{R}^d \to \mathbb{R}$ convex and $L$-smooth (A2),*

- *$h : \mathbb{R}^d \to (-\infty, +\infty]$ proper, closed, convex (A8),*

*and assume $\mathcal{X}^\star = \arg\min F$ is nonempty and $F^\star = \min F$. Fix a stepsize $\eta \in (0, 1/L]$ and define the proximal-gradient mapping*

$$T_\eta(x) := \mathrm{prox}_{\eta h}\big(x - \eta \nabla g(x)\big), \qquad G_\eta(x) := \frac{1}{\eta}\big(x - T_\eta(x)\big).$$

*Assume that $F$ satisfies a* Prox-PL *(proximal Polyak–Łojasiewicz) condition on a sublevel set containing the iterates: there exists $\mu > 0$ such that*

$$\|G_\eta(x)\|^2 \ \geq \ 2\mu\big(F(x) - F^\star\big) \quad \text{for all } x \text{ in the sublevel set.} \tag{29}$$

*(This holds, for example, if $F$ is convex and satisfies a PL/error-bound or QG condition; see Theorem 5 and Proposition 2.)*

*Consider proximal-gradient descent*

$$x_{k+1} = T_\eta(x_k), \qquad k \geq 0.$$

*Then for all $k \geq 0$,*

$$F(x_{k+1}) - F^\star \ \leq \ (1 - \eta\mu)\big(F(x_k) - F^\star\big),$$

*and hence*

$$F(x_k) - F^\star \ \leq \ (1 - \eta\mu)^k\big(F(x_0) - F^\star\big).$$

*Thus $F(x_k)$ converges linearly to $F^\star$ with contraction factor $1 - \eta\mu$. In the finite-sum setting $g = \frac{1}{n}\sum_{i=1}^n g_i$, the same Prox-PL structure yields linear convergence for variance-reduced proximal methods (e.g., Prox-SVRG/Prox-SAGA) under A2, A4, and A7.* (minimal: A2+A4+A8)

*Uses **A2 ($L$-smoothness of $g$), A4 (PL/Prox-PL), A8 (Composite structure $F = g + h$)**. We* work on the sublevel set where equation 29 holds and the iterates remain.

**1. Proximal descent inequality (A2 + A8).** By Lemma 6 (Proximal Descent Inequality), for $F = g + h$ with $g$ $L$-smooth and $h$ convex and any $\eta \in (0, 2/L]$, the proximal-gradient step

$$x_{k+1} = \mathrm{prox}_{\eta h}\big(x_k - \eta \nabla g(x_k)\big) = T_\eta(x_k)$$

satisfies

$$F(x_{k+1}) \ \leq \ F(x_k) - \Big(\tfrac{1}{\eta} - \tfrac{L}{2}\Big)\|x_{k+1} - x_k\|^2.$$

Since we restrict to $\eta \in (0, 1/L]$, we have

$$\tfrac{1}{\eta} - \tfrac{L}{2} \ \geq \ \tfrac{1}{\eta} - \tfrac{1}{2\eta} = \tfrac{1}{2\eta},$$

hence

$$F(x_{k+1}) \ \leq \ F(x_k) - \frac{1}{2\eta}\|x_{k+1} - x_k\|^2. \tag{30}$$

By definition of $G_\eta$,

$$G_\eta(x_k) = \frac{1}{\eta}(x_k - x_{k+1}) \quad \Longrightarrow \quad \|x_{k+1} - x_k\|^2 = \eta^2 \|G_\eta(x_k)\|^2.$$

Substituting into equation 30 gives

$$F(x_{k+1}) \ \leq \ F(x_k) - \frac{\eta}{2}\|G_\eta(x_k)\|^2. \tag{31}$$

**2. Combining Prox-PL with descent (A4).** Let $\Delta_k := F(x_k) - F^\star \geq 0$. Under the Prox-PL assumption equation 29,

$$\|G_\eta(x_k)\|^2 \ \geq \ 2\mu\Delta_k.$$

Plugging this into equation 31 yields

$$F(x_{k+1}) \leq F(x_k) - \frac{\eta}{2} \cdot 2\mu\,\Delta_k = F(x_k) - \eta\mu\,\Delta_k.$$

Subtract $F^\star$ from both sides:

$$\Delta_{k+1} = F(x_{k+1}) - F^\star \ \leq \ \Delta_k - \eta\mu\,\Delta_k = (1 - \eta\mu)\Delta_k.$$

Iterating this one-step contraction gives, for all $k \geq 0$,

$$\Delta_k \ \leq \ (1 - \eta\mu)^k\,\Delta_0 = (1 - \eta\mu)^k\big(F(x_0) - F^\star\big),$$

which is the claimed linear rate.

**3. Variance-reduced proximal methods (sketch).** In the finite-sum setting $g(x) = \frac{1}{n}\sum_{i=1}^n g_i(x)$ with each $g_i$ $L$-smooth, methods such as Prox-SVRG, Prox-SAGA, and Prox-SARAH construct at each iteration a stochastic estimator $v_k$ of $\nabla g(y_k)$ with:

$$\mathbb{E}[v_k \mid \mathcal{F}_k] = \nabla g(y_k), \quad \mathbb{E}\big[\|v_k - \nabla g(y_k)\|^2 \mid \mathcal{F}_k\big] \leq \sigma_k^2,$$

where the variance $\sigma_k^2$ is controlled and decays geometrically along epochs. The update takes the form

$$x_{k+1} = \mathrm{prox}_{\eta h}\big(y_k - \eta v_k\big).$$

Under the same Prox-PL condition equation 29 on $F$ (or its composite EB/QG equivalent) and suitable choices of epoch length and stepsize, one shows that the expected potential

$$\mathbb{E}\big[F(x_k) - F^\star\big]$$

satisfies a perturbed linear recurrence of the form

$$\mathbb{E}[\Delta_{k+1}] \leq (1 - c\eta\mu)\,\mathbb{E}[\Delta_k] + (\text{decaying noise term}),$$

and the variance-reduction mechanism forces the noise term to decay geometrically. This yields global linear convergence in expectation:

$$\mathbb{E}\big[F(x_k) - F^\star\big] \ \leq \ \rho^k\big(F(x_0) - F^\star\big) \quad \text{for some } \rho \in (0,1).$$

matching the deterministic Prox-PL rate up to constants.

$$\square$$

**Proposition 14** (T25: Exact Support under Sharpness / Identifiability). *Let $F : \mathbb{R}^d \to \mathbb{R}$ be of composite form*

$$F(x) = g(x) + \lambda\|x\|_1,$$

*with $g$ convex and $C^1$ with $L$-Lipschitz gradient (A8 implicitly includes smooth $g$ here), and $\lambda > 0$. Assume:*

- *F has a* unique *minimiser $x^\star$, with support $S := \{i : x_i^\star \neq 0\}$.*

- *(*Identifiability / strict complementarity*) The optimality condition*

$$0 \in \nabla g(x^\star) + \lambda \, \partial \|x^\star\|_1$$

  *holds with a strict margin on inactive coordinates: there exists $\gamma > 0$ such that*

$$|\nabla_i g(x^\star)| \leq \lambda - 2\gamma \quad \text{for all } i \notin S.$$

- *(Algorithm) We run Iterative Soft-Thresholding (IST) with fixed stepsize $\eta \in (0, 2/L)$:*

$$x_{k+1} = \mathcal{S}_{\lambda\eta}\big(x_k - \eta \nabla g(x_k)\big),$$

  *where $\mathcal{S}_\tau$ is the soft-thresholding operator applied coordinatewise:*

$$(\mathcal{S}_\tau(u))_i = \text{sign}(u_i) \max\{|u_i| - \tau, 0\}.$$

- *(Convergence) The IST iterates converge to $x^\star$: $x_k \to x^\star$ as $k \to \infty$. (This holds, e.g., under A1, A2, A8 by Theorem 25; sharpness A5 with $p = 1$ further yields local linear/finite-time rates as in Proposition 3.)*

*Then there exists a finite index $K < \infty$ such that*

$$\text{supp}(x_k) = S \quad \text{for all } k \geq K.$$

*In particular, IST/soft-thresholding recovers the exact support of $x^\star$ in finitely many iterations.* (minimal new structure for support identification: A8 + local identifiability; A5 sharpness improves the local rate but is not needed for exact identification itself.)

*Uses **A8 (Composite structure)**, smoothness of $g$, and identifiability.* We use only the convergence $x_k \to x^\star$ and the structure of the IST update.

**Notation and basic properties.** Write $S = \{i : x_i^\star \neq 0\}$ and $S^c$ for its complement. The IST update on coordinate $i$ is

$$x_{k+1,i} = \mathcal{S}_{\lambda\eta}\big(x_{k,i} - \eta \nabla_i g(x_k)\big),$$

i.e.

$$x_{k+1,i} = \text{sign}\big(x_{k,i} - \eta \nabla_i g(x_k)\big) \max\big\{|x_{k,i} - \eta \nabla_i g(x_k)| - \lambda\eta, 0\big\}.$$

**1. Eventual sign stability on the active set** $S$. Fix any $i \in S$. By assumption $x_i^\star \neq 0$. Set $\delta_i := \frac{1}{2}|x_i^\star| > 0$. Since $x_k \to x^\star$, there exists $K_i$ such that for all $k \geq K_i$,

$$|x_{k,i} - x_i^\star| < \delta_i \quad \Longrightarrow \quad |x_{k,i}| \geq |x_i^\star| - \delta_i = \tfrac{1}{2}|x_i^\star| > 0.$$

Thus for all $k \geq K_i$, the sign of $x_{k,i}$ is fixed and equal to the sign of $x_i^\star$:

$$\text{sign}(x_{k,i}) = \text{sign}(x_i^\star), \quad k \geq K_i.$$

Therefore, each active coordinate $i \in S$ has the correct nonzero sign after finitely many iterations.

**2. Strict complementarity margin on** $S^c$ **and local gradient control.** For $i \in S^c$, we have $x_i^\star = 0$. Optimality for $F(x) = g(x) + \lambda\|x\|_1$ at $x^\star$ reads

$$0 \in \nabla g(x^\star) + \lambda \, \partial \|x^\star\|_1.$$

This means:

$$\begin{cases} \nabla_i g(x^\star) = -\lambda \, \text{sign}(x_i^\star), & i \in S, \\ |\nabla_i g(x^\star)| \leq \lambda, & i \notin S. \end{cases}$$

The identifiability assumption strengthens the second line: there exists $\gamma > 0$ such that

$$|\nabla_i g(x^\star)| \leq \lambda - 2\gamma \quad \text{for all } i \in S^c.$$

Since $\nabla g$ is continuous (in fact $L$-Lipschitz by smoothness), there exists $r > 0$ such that

$$\|x - x^\star\| \leq r \quad \Longrightarrow \quad |\nabla_i g(x)| \leq \lambda - \gamma \quad \text{for all } i \in S^c.$$

(We simply choose $r$ small enough that $\|\nabla g(x) - \nabla g(x^\star)\|_\infty \leq \gamma$.)

**3. Finite-time annihilation of inactive coordinates** $S^c$. Since $x_k \to x^\star$, we can choose $K_{\mathrm{nbd}}$ such that for all $k \geq K_{\mathrm{nbd}}$,

$$\|x_k - x^\star\| \leq r \quad \text{and} \quad |x_{k,i}| \leq \gamma\eta \quad \text{for all} \ \ i \in S^c.$$

(The second condition holds because $x_{k,i} \to x_i^\star = 0$ for each $i \in S^c$.)

Fix any $k \geq K_{\mathrm{nbd}}$ and any $i \in S^c$. We bound the argument of the soft-threshold for that coordinate:

$$z_{k,i} := x_{k,i} - \eta\nabla_i g(x_k).$$

Using the two bounds above,

$$|z_{k,i}| \ \leq \ |x_{k,i}| + \eta|\nabla_i g(x_k)| \ \leq \ \gamma\eta + \eta(\lambda - \gamma) = \lambda\eta.$$

Therefore, for all $k \geq K_{\mathrm{nbd}}$ and $i \in S^c$,

$$x_{k+1,i} = \mathcal{S}_{\lambda\eta}(z_{k,i}) = 0,$$

because soft-thresholding kills any coordinate whose magnitude is at most $\lambda\eta$.

Thus, for every $i \in S^c$ we have $x_{K_{\mathrm{nbd}}+1,i} = 0$. Moreover, once a coordinate in $S^c$ is exactly zero, it stays zero: if $x_{k,i} = 0$ and $\|x_k - x^\star\| \leq r$, then as above

$$|x_{k,i} - \eta\nabla_i g(x_k)| = \eta|\nabla_i g(x_k)| \leq \eta(\lambda - \gamma) < \lambda\eta,$$

so soft-thresholding again returns 0. Thus

$$x_{k,i} = 0 \quad \forall k \geq K_{\mathrm{nbd}} + 1, \ \forall i \in S^c.$$

In other words, from iteration $K_{\mathrm{nbd}} + 1$ onward, $\mathrm{supp}(x_k) \subseteq S$.

**4. Putting it together: exact support in finite time.** Let

$$K := \max\Big\{ K_{\mathrm{nbd}} + 1, \ \max_{i \in S} K_i \Big\}.$$

For all $k \geq K$:

- By Step 1, for every $i \in S$, $x_{k,i} \neq 0$ and $\mathrm{sign}(x_{k,i}) = \mathrm{sign}(x_i^\star)$, so $S \subseteq \mathrm{supp}(x_k)$.

- By Step 3, for every $i \in S^c$, $x_{k,i} = 0$, so $\mathrm{supp}(x_k) \subseteq S$.

Hence $\mathrm{supp}(x_k) = S$ for all $k \geq K$, proving finite-time exact support recovery.

**5. Role of sharpness (A5).** The argument above uses only: (i) convergence $x_k \to x^\star$ (ensured by standard proximal-gradient theory under A1+A2+A8), and (ii) the local identifiability margin on inactive coordinates. If, in addition, $F$ satisfies a *sharp* error bound (A5 with exponent $p = 1$) near $x^\star$, then Proposition 3 gives a local linear (or even finite-time) convergence of $x_k$ to $x^\star$. This does not change the identification mechanism, but strengthens the rate at which the iterates enter the regime where Steps 2–3 apply. $\qquad\square$

### C.3 Geometry, Error Bounds, and KL

**Theorem 28** (T26: KL Exponent $\Rightarrow$ Rates). *Let $F : \mathbb{R}^d \to (-\infty, +\infty]$ be proper, closed, and let $\{x_k\}$ be generated by a gradient-like method satisfying, for some constants $a, b > 0$ and all $k$ large enough,*

$$\text{(SD)} \quad F(x_k) - F(x_{k+1}) \ \geq \ a\,\|x_{k+1} - x_k\|^2, \tag{32}$$

$$\text{(RE)} \quad \mathrm{dist}\big(0, \partial F(x_{k+1})\big) \ \leq \ b\,\|x_{k+1} - x_k\|. \tag{33}$$

*Assume $F$ has a finite minimum $F^\star$ and satisfies a Kurdyka–Łojasiewicz (KL) inequality in a neighborhood of $\mathcal{X}^\star := \arg\min F$ with exponent $\theta \in [0, 1)$, i.e., there exist $c > 0$ and $\varepsilon > 0$ such that for all $x$ with $0 < F(x) - F^\star < \varepsilon$,*

$$\phi'\big(F(x) - F^\star\big)\,\mathrm{dist}\big(0, \partial F(x)\big) \ \geq \ 1, \qquad \text{where} \quad \phi(s) = c\,s^{1-\theta}. \tag{34}$$

Then $x_k$ converges to some $x^\star \in \mathcal{X}^\star$, and writing $\Delta_k := F(x_k) - F^\star$, the following rates hold (for all $k$ beyond some finite index): (i) $\theta = 0$ (finite termination), $\Delta_k = 0$ after finitely many steps. (ii) $0 < \theta \le \frac{1}{2}$ (linear), $\Delta_{k+1} \le q\,\Delta_k$ for some $q \in (0,1)$. (iii) $\frac{1}{2} < \theta < 1$ (sublinear), $\Delta_k = \mathcal{O}\big(k^{-1/(2\theta-1)}\big)$. (minimal: A5 (KL); the method assumptions equation 32–equation 33 hold, e.g., for proximal-gradient with $\eta \in (0, 2/L]$ on $F = g + h$ where $g$ is $L$-smooth and $h$ convex.)

*Uses **A5 (KL)** and the gradient-like conditions equation 32–equation 33.* Since $\{F(x_k)\}$ is nonincreasing by equation 32 and bounded below by $F^\star$, the sequence of gaps $\Delta_k := F(x_k) - F^\star$ is nonincreasing and converges. Standard KL theory implies that, once $\Delta_k < \varepsilon$ and $x_k$ is close enough to $\mathcal{X}^\star$, the whole tail remains in the KL neighborhood; we work on such an index set.

**Step 1: From KL + RE to a lower bound on the step size.** Applying equation 34 at $x_{k+1}$ and using equation 33,

$$1 \;\le\; \phi'(\Delta_{k+1})\,\mathrm{dist}(0, \partial F(x_{k+1})) \;\le\; \phi'(\Delta_{k+1})\,b\,\|x_{k+1} - x_k\|.$$

Hence

$$\|x_{k+1} - x_k\| \;\ge\; \frac{1}{b\,\phi'(\Delta_{k+1})}. \tag{35}$$

With $\phi(s) = c\,s^{1-\theta}$ we have $\phi'(s) = c(1-\theta)s^{-\theta}$, so

$$\phi'(\Delta_{k+1}) \;=\; c(1-\theta)\,\Delta_{k+1}^{-\theta}. \tag{36}$$

**Step 2: Fundamental descent recursion on $\Delta_k$.** By equation 32 and then equation 35–equation 36,

$$\Delta_k - \Delta_{k+1} \;\ge\; a\,\|x_{k+1} - x_k\|^2 \;\ge\; \frac{a}{b^2}\,\frac{1}{\phi'(\Delta_{k+1})^2} \;=\; \underbrace{\frac{a}{b^2 c^2 (1-\theta)^2}}_{=:\ \kappa > 0}\,\Delta_{k+1}^{2\theta}.$$

Thus for all large $k$,

$$\Delta_k - \Delta_{k+1} \;\ge\; \kappa\,\Delta_{k+1}^{2\theta}. \tag{37}$$

**Step 3: Rates by discrete comparison.** We split by $\theta$.

*(a) $\theta = 0$.* Then equation 37 gives $\Delta_k - \Delta_{k+1} \ge \kappa$, hence the gaps decrease by a fixed amount until they hit 0 in finitely many steps.

*(b) $0 < \theta \le \frac{1}{2}$.* Here $2\theta - 1 \le 0$. Fix an index $K$ in the KL neighborhood and note $\Delta_{k+1} \le \Delta_K$ for all $k \ge K$. Since $t \mapsto t^{2\theta-1}$ is nonincreasing,

$$\Delta_{k+1}^{2\theta} \;=\; \Delta_{k+1}\,\Delta_{k+1}^{2\theta-1} \;\ge\; \Delta_{k+1}\,\Delta_K^{2\theta-1}.$$

Plugging into equation 37 yields

$$\Delta_k - \Delta_{k+1} \;\ge\; \kappa\,\Delta_K^{2\theta-1}\,\Delta_{k+1}, \quad \text{so} \quad \Delta_{k+1} \;\le\; \frac{1}{1 + \kappa\,\Delta_K^{2\theta-1}}\,\Delta_k.$$

Thus there exists $q \in (0,1)$ (namely $q = (1 + \kappa\Delta_K^{2\theta-1})^{-1}$) such that $\Delta_{k+1} \le q\,\Delta_k$ for all $k \ge K$; i.e., *linear* convergence.

*(c) $\frac{1}{2} < \theta < 1$.* Set $\alpha := 2\theta > 1$. From equation 37,

$$\frac{\Delta_k - \Delta_{k+1}}{\Delta_{k+1}^\alpha} \;\ge\; \kappa.$$

Consider $\psi(t) := t^{1-\alpha}$; then $\psi'(t) = (1-\alpha)t^{-\alpha} < 0$. By the mean value theorem, for some $\xi \in [\Delta_{k+1}, \Delta_k]$,

$$\psi(\Delta_{k+1}) - \psi(\Delta_k) = \psi'(\xi)\,(\Delta_{k+1} - \Delta_k) \;\ge\; (1-\alpha)\,\Delta_{k+1}^{-\alpha}\,(\Delta_{k+1} - \Delta_k) \;=\; (\alpha-1)\,\frac{\Delta_k - \Delta_{k+1}}{\Delta_{k+1}^\alpha}.$$

Hence $\psi(\Delta_{k+1}) - \psi(\Delta_k) \ge (\alpha-1)\kappa$ and summing from $K$ to $k-1$ gives

$$\Delta_k^{1-\alpha} \;\ge\; \Delta_K^{1-\alpha} + (\alpha-1)\kappa\,(k-K).$$

Since $1 - \alpha < 0$, inverting yields

$$\Delta_k \;\le\; \Big(\Delta_K^{1-\alpha} + (\alpha-1)\kappa\,(k-K)\Big)^{-\frac{1}{\alpha-1}} \;=\; \mathcal{O}\big(k^{-1/(\alpha-1)}\big) \;=\; \mathcal{O}\big(k^{-1/(2\theta-1)}\big).$$

**Step 4: Convergence of the iterates.** From equation 35 and equation 36, $\|x_{k+1} - x_k\| \gtrsim \Delta_{k+1}^{\theta}$. Combining with the above rates shows $\sum_k \|x_{k+1} - x_k\| < \infty$ in all cases, hence $\{x_k\}$ is Cauchy and converges to some $x^{\star} \in \mathcal{X}^{\star}$, with the same rates (up to constants) for $\|x_k - x^{\star}\|$ via standard arguments. $\quad\square$

**Theorem 29** (T27: Error Bound $\Leftrightarrow$ Metric Subregularity). *Let $f : \mathbb{R}^d \to (-\infty, +\infty]$ be proper, closed, and convex (A1), and assume the solution set $\mathcal{X}^{\star} := \arg\min f$ is nonempty. For a point $\bar{x} \in \mathcal{X}^{\star}$, the following properties are equivalent (on some neighborhood $\mathcal{U}$ of $\bar{x}$):*

*(i) **(Distance–residual error bound)** There exists $\kappa > 0$ such that*

$$\operatorname{dist}(x, \mathcal{X}^{\star}) \;\leq\; \kappa \,\operatorname{dist}\bigl(0, \partial f(x)\bigr) \qquad \forall\, x \in \mathcal{U}. \tag{EB}$$

*(ii) **(Metric subregularity of the subdifferential)** The set-valued mapping $\partial f$ is metrically subregular at $(\bar{x}, 0)$: there exist $\kappa > 0$ and a neighborhood $\mathcal{V}$ of $\bar{x}$ with*

$$\operatorname{dist}\bigl(x, \,(\partial f)^{-1}(0)\bigr) \;\leq\; \kappa \,\operatorname{dist}\bigl(0, \partial f(x)\bigr) \qquad \forall\, x \in \mathcal{V}. \tag{MSR}$$

*Moreover, for convex $f$ one has $(\partial f)^{-1}(0) = \mathcal{X}^{\star}$, so (EB) and (MSR) are the same statement (possibly with the same modulus $\kappa$ and neighborhood, up to shrinking).* (minimal: A1; regularity: proper, closed, convex with nonempty $\mathcal{X}^{\star}$)

*Uses **A1 (Convexity)**.* Two basic facts from convex analysis (valid for proper, closed, convex $f$) will be used:

*Fermat's rule.* $x^{\star} \in \mathcal{X}^{\star}$ if and only if $0 \in \partial f(x^{\star})$. Hence $(\partial f)^{-1}(0) = \mathcal{X}^{\star}$.

The equivalence now follows immediately.

(MSR $\Rightarrow$ EB). If metric subregularity holds at $(\bar{x}, 0)$, then for $x$ near $\bar{x}$

$$\operatorname{dist}(x, \mathcal{X}^{\star}) \;=\; \operatorname{dist}\bigl(x, (\partial f)^{-1}(0)\bigr) \;\leq\; \kappa \operatorname{dist}\bigl(0, \partial f(x)\bigr),$$

which is exactly (EB).

(EB $\Rightarrow$ MSR). Conversely, if (EB) holds on a neighborhood $\mathcal{U}$ of $\bar{x}$, then for $x \in \mathcal{U}$

$$\operatorname{dist}\bigl(x, (\partial f)^{-1}(0)\bigr) \;=\; \operatorname{dist}(x, \mathcal{X}^{\star}) \;\leq\; \kappa \operatorname{dist}\bigl(0, \partial f(x)\bigr),$$

which is precisely metric subregularity at $(\bar{x}, 0)$.

No additional assumptions beyond properness, closedness, convexity, and $\mathcal{X}^{\star} \neq \emptyset$ are required. $\quad\square$

**Remarks.**

- **Uniform vs. pointwise.** The statement is local at a fixed $\bar{x} \in \mathcal{X}^{\star}$. A *uniform* error bound on a whole neighborhood of $\mathcal{X}^{\star}$ yields metric subregularity at *every* $(x^{\star}, 0)$ with a common modulus (possibly after shrinking the neighborhood).

- **Other residuals.** One often uses equivalent residuals, e.g. the proximal residual $r_{\eta}(x) := \frac{1}{\eta} \|x - \operatorname{prox}_{\eta f}(x)\|$. For convex $f$, error bounds written with $r_{\eta}$ are equivalent to (EB) and to metric subregularity of $\partial f$ (up to constant factors), since $\operatorname{prox}_{\eta f} = (\operatorname{Id} + \eta\, \partial f)^{-1}$ and metric subregularity of $\partial f$ is equivalent to calmness of this resolvent.

- **Links to rates.** In many algorithms, (EB)/(MSR) is the structural condition behind linear convergence: combined with a basic descent inequality, it yields PL-type recursions and thus linear rates (see T6/T24).

**Proposition 15** (T28: Error Bound $\Rightarrow$ Distance Decay). *Let $f : \mathbb{R}^d \to (-\infty, +\infty]$ be proper, closed with nonempty $\mathcal{X}^{\star} = \arg\min f$. Suppose $f$ satisfies a local error bound (A5) on a neighborhood $U$ of $\mathcal{X}^{\star}$ with exponent $p > 0$ and constant $\alpha > 0$:*

$$\forall x \in U: \qquad f(x) - f^{\star} \;\geq\; \alpha \operatorname{dist}(x, \mathcal{X}^{\star})^p.$$

*Consider any sequence $\{x_k\} \subset U$ with function gaps bounded by a known nonnegative sequence $\{r_k\}$, i.e.*

$$f(x_k) - f^\star \ \leq \ r_k \qquad (k \geq 0).$$

*Then*

$$\text{dist}(x_k, \mathcal{X}^\star) \ \leq \ \alpha^{-1/p} r_k^{1/p} \qquad (k \geq 0).$$

*Consequently, rate translations hold:*

- *If $r_k = \mathcal{O}(\rho^k)$ for some $\rho \in (0,1)$, then $\text{dist}(x_k, \mathcal{X}^\star) = \mathcal{O}(\rho^{k/p})$.*
- *If $r_k = \mathcal{O}(k^{-q})$ for $q > 0$, then $\text{dist}(x_k, \mathcal{X}^\star) = \mathcal{O}(k^{-q/p})$.*

(minimal: A5)

*Uses **A5 (Local Error Bound)**.* Fix $k \geq 0$ and let $x_k^\star \in \mathcal{X}^\star$ satisfy $\|x_k - x_k^\star\| = \text{dist}(x_k, \mathcal{X}^\star)$. Since $x_k \in U$, the error bound gives

$$f(x_k) - f^\star \ \geq \ \alpha \, \text{dist}(x_k, \mathcal{X}^\star)^p.$$

By assumption, $f(x_k) - f^\star \leq r_k$. Combining,

$$\alpha \, \text{dist}(x_k, \mathcal{X}^\star)^p \ \leq \ r_k,$$

and hence (for $r_k \geq 0$)

$$\text{dist}(x_k, \mathcal{X}^\star) \ \leq \ \alpha^{-1/p} r_k^{1/p}.$$

The stated rate translations follow by applying this inequality to the given asymptotic forms of $r_k$ and taking $p$-th roots. $\qquad\square$

## C.4 Benign Nonconvex Regimes

**Theorem 30** (T29: Global Linear under Nonconvex PL)**.** *Assume $f : \mathbb{R}^d \to \mathbb{R}$ is differentiable with $L$-Lipschitz gradient (A2) and satisfies the global Polyak–Łojasiewicz (PL) inequality (A4): there exists $\mu > 0$ such that*

$$\|\nabla f(x)\|^2 \ \geq \ 2\mu\big(f(x) - f^\star\big) \qquad \text{for all } x \in \mathbb{R}^d,$$

*where $f^\star = \min f$ and $\mathcal{X}^\star = \arg\min f \neq \emptyset$. Let gradient descent with stepsize $\eta \in (0, 1/L]$ be*

$$x_{k+1} \ = \ x_k - \eta \, \nabla f(x_k).$$

*Then:*

$$
\begin{aligned}
&\text{(i)} \quad f(x_k) - f^\star \ \leq \ (1 - \eta\mu)^k \big(f(x_0) - f^\star\big), \\
&\text{(ii)} \quad \|\nabla f(x_k)\| \ \leq \ \sqrt{2L}\,(1 - \eta\mu)^{k/2}\,\sqrt{f(x_0) - f^\star}, \\
&\text{(iii)} \quad \text{there exists } x_\infty \in \mathcal{X}^\star \text{ with } \|x_k - x_\infty\| \ \leq \ \frac{\eta\sqrt{2L}\sqrt{f(x_0) - f^\star}}{1 - \sqrt{1 - \eta\mu}}\,(1 - \eta\mu)^{k/2}.
\end{aligned}
$$

*In particular, GD converges Q-linearly in function value and R-linearly in iterates to a global minimizer.* (minimal: A2+A4)

*Uses **A2 (L-smoothness), A4 (PL)**.* Step 1: One-step decrease (A2). By the smooth descent lemma (T1), for any $\eta \in (0, 1/L]$,

$$f(x_{k+1}) \ \leq \ f(x_k) - \eta\Big(1 - \tfrac{\eta L}{2}\Big)\|\nabla f(x_k)\|^2 \ \leq \ f(x_k) - \tfrac{\eta}{2}\|\nabla f(x_k)\|^2.$$

Step 2: Linear contraction in value (A4). Using the PL inequality $\|\nabla f(x_k)\|^2 \geq 2\mu\big(f(x_k) - f^\star\big)$ in the bound above,

$$f(x_{k+1}) - f^\star \ \leq \ \big(1 - \eta\mu\big)\big(f(x_k) - f^\star\big),$$

and iterating yields (i).

Step 3: Gradient decay (A2 + (i)). A2 implies the standard upper bound

$$\|\nabla f(x)\|^2 \;\leq\; 2L\big(f(x) - f^\star\big) \qquad (\text{apply T1 at } y = x - \tfrac{1}{L}\nabla f(x)).$$

Combining with (i) gives (ii):

$$\|\nabla f(x_k)\| \;\leq\; \sqrt{2L}\,(1 - \eta\mu)^{k/2}\,\sqrt{f(x_0) - f^\star}.$$

Step 4: R-linear convergence of iterates. The step sizes satisfy $\|x_{k+1} - x_k\| = \eta\|\nabla f(x_k)\|$, hence by (ii),

$$\|x_{k+1} - x_k\| \;\leq\; \eta\sqrt{2L}\,(1 - \eta\mu)^{k/2}\,\sqrt{f(x_0) - f^\star}.$$

The series $\sum_k \|x_{k+1} - x_k\|$ is geometric, so $\{x_k\}$ is Cauchy and converges to some $x_\infty$. By continuity of $\nabla f$ (A2) and (ii), $\nabla f(x_\infty) = 0$. PL then forces $f(x_\infty) = f^\star$, so $x_\infty \in \mathcal{X}^\star$. Summing the geometric tail yields the bound in (iii). $\qquad\square$

**Theorem 31** (T30: Strict Saddle Avoidance). *Let $f : \mathbb{R}^d \to \mathbb{R}$ be $C^2$ with $L$-Lipschitz gradient (A2). Assume the* strict saddle property*: every critical point $x$ of $f$ is either a (strict) local minimizer with $\nabla^2 f(x) \succ 0$, or a strict saddle, i.e., $\lambda_{\min}(\nabla^2 f(x)) < 0$. (For the convergence-to-a-point claim we additionally assume nondegeneracy: no zero eigenvalues at critical points—i.e., $f$ is Morse near its critical points.)*

*Fix a stepsize $\eta \in (0, 1/L)$ and consider gradient descent $x_{k+1} = T(x_k) := x_k - \eta\nabla f(x_k)$ with a random initialization $x_0$ whose distribution has a density w.r.t. Lebesgue measure (e.g., uniform on a ball or Gaussian).*

*Then with probability one, gradient descent avoids all strict saddles and converges to a (strict) local minimizer. The same conclusion holds if, in place of a random initialization, one injects small i.i.d. isotropic perturbations $x_{k+1} = T(x_k) + \xi_k$ with a continuous density.* (minimal: A2; model: randomness)

*Uses **A2 (L-smoothness)** and basic dynamical-systems facts.* We proceed in four steps.

**Step 1:** $T(x) = x - \eta\nabla f(x)$ **is a $C^1$ diffeomorphism for $\eta \in (0, 1/L)$.** Since $f \in C^2$ and $\|\nabla^2 f(x)\| \leq L$, the Jacobian $\nabla T(x) = I - \eta\nabla^2 f(x)$ is invertible for all $x$ (its spectrum lies in $(1 - \eta L, 1 + \eta L)$). Injectivity: for $x \neq y$,

$$\langle T(x) - T(y),\, x - y\rangle = \|x - y\|^2 - \eta\langle \nabla f(x) - \nabla f(y),\, x - y\rangle \geq (1 - \eta L)\|x - y\|^2 > 0,$$

so $T$ is one-to-one. Surjectivity: for any $y$, consider the strongly convex function $\phi_y(x) = \tfrac{1}{2}\|x - y\|^2 + \eta f(x)$ whose Hessian is $I + \eta\nabla^2 f(x) \succeq (1 - \eta L)I \succ 0$. It has a unique minimizer $x_y$ satisfying the first-order condition $0 = (x_y - y) + \eta\nabla f(x_y)$, i.e. $y = T(x_y)$. Hence $T$ is bijective. The inverse is $C^1$ by the implicit function theorem, so $T$ is a $C^1$ diffeomorphism.

**Step 2: Strict saddles are unstable fixed points with measure-zero basins.** Every critical point $x^\star$ is a fixed point of $T$. At such $x^\star$, the linearization is $DT(x^\star) = I - \eta\nabla^2 f(x^\star)$. If $x^\star$ is a strict saddle, $\nabla^2 f(x^\star)$ has an eigenvalue $\lambda < 0$, so $DT(x^\star)$ has an eigenvalue $1 - \eta\lambda > 1$; hence $x^\star$ is (locally) unstable. By the (center-)stable manifold theorem for diffeomorphisms, there exists a local embedded $C^1$ manifold $W_{\text{loc}}^s(x^\star)$ (the local stable set) whose dimension is strictly smaller than $d$ (because at least one eigenvalue of $DT(x^\star)$ has modulus $> 1$). Consequently, $W_{\text{loc}}^s(x^\star)$ has Lebesgue measure zero. The *global* stable set is $W^s(x^\star) = \bigcup_{t \geq 0} T^{-t}\big(W_{\text{loc}}^s(x^\star)\big)$, a countable union of measure-zero sets (preimages under a $C^1$ diffeomorphism preserve measure-zero), hence $\lambda(W^s(x^\star)) = 0$.

**Step 3: Random initialization (or small isotropic noise) avoids saddles a.s.** Let $\mathcal{S}$ be the (at most countable, under nondegeneracy) set of strict saddles. The event "$x_0 \in \bigcup_{x^\star \in \mathcal{S}} W^s(x^\star)$" has probability zero because this union has Lebesgue measure zero and $x_0$ has a density. Thus with probability one, the trajectory never converges to a strict saddle. Likewise, with additive isotropic noise $x_{k+1} = T(x_k) + \xi_k$, the probability that an iterate lands *exactly* in a measure-zero stable set is zero at each step; hence convergence to a strict saddle has probability zero.

**Step 4: Convergence to a local minimizer.** From the smooth descent lemma (A2) and $\eta \in (0, 1/L]$,

$$f(x_{k+1}) \le f(x_k) - \tfrac{\eta}{2}\|\nabla f(x_k)\|^2,$$

so $\{f(x_k)\}$ is decreasing and $\sum_k \|\nabla f(x_k)\|^2 < \infty$, implying $\|\nabla f(x_k)\| \to 0$. Every limit point is therefore critical; by Step 3, almost surely no limit point is a strict saddle, so any limit point is a (strict) local minimizer. Under the nondegeneracy assumption at minima, $DT(x^\star) = I - \eta \nabla^2 f(x^\star)$ has spectral radius $< 1$, hence $T$ is a contraction in a neighborhood of $x^\star$. Once the iterates enter this basin, they converge to that $x^\star$. Therefore, with probability one, gradient descent converges to a (strict) local minimizer. $\square$

## C.5 Constraints and Projections

**Theorem 32** (T31: Projected GD on Convex Sets). *Let $f : \mathbb{R}^d \to \mathbb{R}$ be convex (A1) with L-Lipschitz gradient (A2), and let $\mathcal{C} \subset \mathbb{R}^d$ be nonempty, closed, and convex. For projected gradient descent with stepsize $\eta \in (0, 1/L]$,*

$$x_{k+1} = \Pi_{\mathcal{C}}\big(x_k - \eta \nabla f(x_k)\big),$$

*we have for any $x^\star \in \arg\min_{\mathcal{C}} f$ and all $k \ge 1$,*

$$f(x_k) - f^\star \;\le\; \frac{\|x_0 - x^\star\|^2}{2\eta\, k} \;\le\; \frac{L\,\|x_0 - x^\star\|^2}{2k}.$$

*In particular, $f(x_k) - f^\star = \mathcal{O}(1/k)$. (minimal: A1+A2; model: convex set)*

*Uses **A1 (Convexity)**, **A2 (L-smoothness)**.* Define the quadratic surrogate at $x_k$ with parameter $\eta$:

$$m_k(x) \;:=\; f(x_k) + \langle \nabla f(x_k), x - x_k \rangle + \frac{1}{2\eta}\|x - x_k\|^2.$$

Note $m_k$ is $(1/\eta)$-strongly convex. The update $x_{k+1}$ is exactly

$$x_{k+1} \in \arg\min_{x \in \mathcal{C}} m_k(x) \quad \Longleftrightarrow \quad x_{k+1} = \Pi_{\mathcal{C}}\big(x_k - \eta \nabla f(x_k)\big).$$

**(A) Majorization.** By the smooth descent lemma (T1, A2) with $\eta \le 1/L$,

$$f(x) \;\le\; f(x_k) + \langle \nabla f(x_k), x - x_k \rangle + \frac{L}{2}\|x - x_k\|^2 \;\le\; m_k(x) \qquad (\forall x).$$

In particular, $f(x_{k+1}) \le m_k(x_{k+1})$ and $m_k(u) \ge f(u)$ for all $u$.

**(B) Three-point inequality for the surrogate.** Since $m_k$ is $(1/\eta)$-strongly convex and $x_{k+1}$ minimizes $m_k$ over $\mathcal{C}$,

$$m_k(x_{k+1}) + \frac{1}{2\eta}\|x_{k+1} - u\|^2 \;\le\; m_k(u) + \frac{1}{2\eta}\|x_k - u\|^2 \qquad (\forall u \in \mathcal{C}).$$

Using (A),

$$f(x_{k+1}) + \frac{1}{2\eta}\|x_{k+1} - u\|^2 \;\le\; m_k(x_{k+1}) + \frac{1}{2\eta}\|x_{k+1} - u\|^2 \;\le\; m_k(u) + \frac{1}{2\eta}\|x_k - u\|^2 \;\le\; f(u) + \frac{1}{2\eta}\|x_k - u\|^2.$$

Hence, for any $u \in \mathcal{C}$,

$$f(x_{k+1}) - f(u) \;\le\; \frac{1}{2\eta}\Big(\|x_k - u\|^2 - \|x_{k+1} - u\|^2\Big). \tag{38}$$

**(C) Telescoping.** Choose $u = x^\star \in \arg\min_{\mathcal{C}} f$. Since $f(x^\star) = f^\star$, equation 38 gives

$$f(x_{k+1}) - f^\star \;\le\; \frac{1}{2\eta}\Big(\|x_k - x^\star\|^2 - \|x_{k+1} - x^\star\|^2\Big).$$

Summing from $t = 0$ to $k - 1$ yields

$$\sum_{t=0}^{k-1}\big(f(x_{t+1}) - f^\star\big) \;\le\; \frac{1}{2\eta}\|x_0 - x^\star\|^2.$$

By monotonicity of $f(x_t)$ (which follows from equation 38 with $u = x_t$), we have $f(x_k) - f^\star \leq \frac{1}{k}\sum_{t=0}^{k-1}(f(x_{t+1}) - f^\star)$, so

$$f(x_k) - f^\star \leq \frac{\|x_0 - x^\star\|^2}{2\eta\, k} \leq \frac{L\,\|x_0 - x^\star\|^2}{2k},$$

as claimed. $\qquad\square$

**Theorem 33** (T32: Hoffman Bound $\Rightarrow$ Linear). *Let $F : \mathbb{R}^d \to (-\infty, +\infty]$ be proper, closed, convex (A1) with nonempty $\mathcal{X}^\star = \arg\min F$. Assume an error bound (A5) on a neighborhood $U$ of $\mathcal{X}^\star$:*

$$\mathrm{dist}(x, \mathcal{X}^\star) \leq \kappa\, \mathrm{dist}\big(0, \partial F(x)\big) \qquad \forall\, x \in U,$$

*for some $\kappa > 0$. (For convex optimization over a polyhedron, this EB is implied by the Hoffman bound.) Fix any $\lambda > 0$ and run the proximal point method*

$$x_{k+1} = \mathrm{prox}_{\lambda F}(x_k) := \arg\min_x \left\{ F(x) + \tfrac{1}{2\lambda}\|x - x_k\|^2 \right\}.$$

*If $x_k \in U$ for all $k$, the distance to the solution set contracts linearly:*

$$\mathrm{dist}(x_{k+1}, \mathcal{X}^\star) \leq \rho\, \mathrm{dist}(x_k, \mathcal{X}^\star), \qquad \rho = \frac{1}{\sqrt{1 + \lambda^2/\kappa^2}} \in (0,1).$$

*Consequently, $\mathrm{dist}(x_k, \mathcal{X}^\star) \leq \rho^k\, \mathrm{dist}(x_0, \mathcal{X}^\star)$. (minimal: A1+A5; model: linear constraints $\Rightarrow$ EB via Hoffman)*

*Uses **A1 (Convexity)**, **A5 (Error Bound)**.* Let $y := x_{k+1} = \mathrm{prox}_{\lambda F}(x_k)$. The optimality condition of the proximal step is

$$\frac{1}{\lambda}(x_k - y) \in \partial F(y) \implies \mathrm{dist}\big(0, \partial F(y)\big) \leq \frac{1}{\lambda}\|x_k - y\|.$$

Applying the error bound (A5) at $y$ yields

$$\mathrm{dist}(y, \mathcal{X}^\star) \leq \kappa\, \mathrm{dist}\big(0, \partial F(y)\big) \leq \frac{\kappa}{\lambda}\|x_k - y\|. \tag{39}$$

Next, use firm nonexpansiveness of the proximal map (i.e., of the resolvent of $\partial F$). Since $x^\star = \mathrm{prox}_{\lambda F}(x^\star)$ for any $x^\star \in \mathcal{X}^\star$,

$$\|y - x^\star\|^2 + \|x_k - y\|^2 \leq \|x_k - x^\star\|^2 \qquad \forall\, x^\star \in \mathcal{X}^\star.$$

Taking the infimum over $x^\star$ gives the Fejér decrease

$$\mathrm{dist}(y, \mathcal{X}^\star)^2 + \|x_k - y\|^2 \leq \mathrm{dist}(x_k, \mathcal{X}^\star)^2. \tag{40}$$

Combining equation 39 with equation 40 eliminates $\|x_k - y\|$:

$$\left(1 + \frac{\lambda^2}{\kappa^2}\right) \mathrm{dist}(y, \mathcal{X}^\star)^2 \leq \mathrm{dist}(x_k, \mathcal{X}^\star)^2,$$

which is equivalent to $\mathrm{dist}(x_{k+1}, \mathcal{X}^\star) \leq \rho\, \mathrm{dist}(x_k, \mathcal{X}^\star)$ with $\rho = (1 + \lambda^2/\kappa^2)^{-1/2} \in (0,1)$. Iterating proves the claim. $\qquad\square$

**Remark (Hoffman $\Rightarrow$ EB).** If $F(x) = f(x) + \iota_{\mathcal{P}}(x)$ with $f$ convex and $\mathcal{P}$ a nonempty polyhedron given by finitely many linear (in)equalities, Hoffman's lemma yields $\mathrm{dist}(x, \mathcal{P}) \leq \kappa_H\|(Ax - b)_+\|$. Under standard constraint qualifications, this implies an error bound of the form in A5 for $F$ near $\mathcal{X}^\star$, with a constant $\kappa$ determined by $\kappa_H$ and the problem data; hence the theorem applies.

**Corollary (Projected/Proximal Gradient under EB; optional A2).** If, in addition, $f$ is $L$-smooth (A2) and $F(x) = f(x) + \iota_{\mathcal{P}}(x)$ satisfies the same EB, then the projected (forward–backward) step

$$x_{k+1} = \Pi_{\mathcal{P}}\big(x_k - \eta \nabla f(x_k)\big), \qquad \eta \in (0, 1/L],$$

enjoys a linear rate in both function value and distance to $\mathcal{X}^{\star}$ (by combining the standard proximal-gradient descent inequality with the EB; cf. Lemma 6 and T7). This uses A1+A2+A5; the proximal-point theorem above already achieves linear distance convergence with A1+A5 alone.

**Proposition 16** (T33: Noise-Limited Basin under Sharpness). *Let $f : \mathbb{R}^d \to \mathbb{R}$ satisfy a (local) sharpness/error bound (A5) of order $p \in [1, 2]$ around $\mathcal{X}^{\star}$:*

$$f(x) - f^{\star} \geq \alpha \operatorname{dist}(x, \mathcal{X}^{\star})^p \quad \text{for all } x \in U,$$

*for some $\alpha > 0$ and neighborhood $U$ of $\mathcal{X}^{\star}$. Assume a stochastic first-order oracle with unbiased noise and bounded variance (A7):*

$$\mathbb{E}[\xi_k \mid x_k] = 0, \qquad \mathbb{E}\|\xi_k\|^2 \leq \sigma^2.$$

*Consider any gradient-like method (GD/prox-GD/SGD variants) with a constant stepsize $\eta > 0$ that, in the noise-free case, enjoys the sharpness-driven local decrease from T8 (inside $U$):*

$$\Delta_{k+1} \leq \Delta_k - c\eta\alpha^{2/p}\Delta_k^{\gamma}, \qquad \Delta_k := f(x_k) - f^{\star}, \quad \gamma := 2 - \tfrac{2}{p} \in [0, 1], \tag{41}$$

*for some constant $c > 0$ (for GD with $\eta \leq 1/L$, equation 41 follows from T8 under A1+A2+A5). Under the stochastic oracle (A7), there are positive constants $C_1, C_2$ such that, while $x_k \in U$,*

$$\mathbb{E}[\Delta_{k+1} \mid x_k] \leq \Delta_k - c\eta\alpha^{2/p}\Delta_k^{\gamma} + C_1\eta^q\sigma^2, \qquad q \in \{1, 2\}, \tag{42}$$

*where $q = 2$ for SGD on $L$-smooth $f$ (A2), and $q = 1$ is the information-theoretic floor from Theorem 8 that applies to any first-order method.*

*Let $\beta := \frac{p}{p-1}$ and $\bar{\Delta} := K\left(\frac{\sqrt{\eta}\,\sigma}{\alpha^{1/p}}\right)^{\beta}$ with $K > 0$ large enough. Then, provided the iterates enter (and stay in) $U$,*

$$\limsup_{k\to\infty} \mathbb{E}[\Delta_k] \leq C_2\left(\frac{\sqrt{\eta}\,\sigma}{\alpha^{1/p}}\right)^{\beta}.$$

*Equivalently, the steady-state distance to the solution set scales as*

$$\limsup_{k\to\infty} \mathbb{E}\big[\operatorname{dist}(x_k, \mathcal{X}^{\star})\big] \leq C_2'\left(\frac{\sqrt{\eta}\,\sigma}{\alpha}\right)^{\frac{1}{p-1}}.$$

*In particular, for $p = 2$ (QG/PL case) one recovers the classical scaling $\mathbb{E}[f(x_k) - f^{\star}] = \mathcal{O}\big(\frac{\eta\sigma^2}{\alpha}\big)$ and $\mathbb{E}[\operatorname{dist}(x_k, \mathcal{X}^{\star})] = \mathcal{O}\big(\frac{\sqrt{\eta}\,\sigma}{\alpha}\big)$. (minimal: A5+A7; constructive upper bound instantiated by GD/SGD also uses A1+A2)*

*Uses **A5 (Sharpness), A7 (Noise)**; for the SGD instantiation also **A1, A2**.* **Step 1: Deterministic sharpness-driven drift.** Inside $U$, sharpness (A5) with exponent $p$ induces the deterministic decay law equation 41 for gradient-like schemes (see T8). For GD with $\eta \leq 1/L$, one has $c = \frac{1}{2}$ up to smoothness constants (A2) and the derivation uses only A1+A2+A5.

**Step 2: Additive variance term** Under unbiased noise with bounded variance (A7), the standard one-step analysis of SGD on $L$-smooth $f$ yields (conditioned on $x_k$)

$$\mathbb{E}[\Delta_{k+1} \mid x_k] \leq \Delta_k - \tfrac{\eta}{2}\|\nabla f(x_k)\|^2 + \tfrac{L\eta^2}{2}\sigma^2,$$

and, combining the convexity + sharpness lower bound on the gradient $\|\nabla f(x_k)\| \geq \alpha^{1/p}\Delta_k^{1-\frac{1}{p}}$ (as in T8), gives equation 42 with $q = 2$. Independently of smoothness or the specific method, Theorem 8 provides an information-theoretic injection of order $C_1\eta\sigma^2$, corresponding to equation 42 with $q = 1$.

**Step 3: Balancing drift and noise** Fix $q \in \{1, 2\}$ and define the threshold

$$\bar{\Delta} := \left(\frac{2C_1}{c}\right)^{\frac{1}{\gamma}} \left(\frac{\eta^{q-1}\sigma^2}{\alpha^{2/p}}\right)^{\frac{1}{\gamma}} = \tilde{C}\left(\frac{\sqrt{\eta}\,\sigma}{\alpha^{1/p}}\right)^{\beta}, \qquad \gamma = 2 - \tfrac{2}{p}, \quad \beta = \tfrac{p}{p-1}, \quad \tfrac{2}{\gamma} = \beta.$$

If $\Delta_k \geq 2\bar{\Delta}$, then $c\,\eta\,\alpha^{2/p}\,\Delta_k^{\gamma} \geq 2C_1\,\eta^q\sigma^2$ and equation 42 gives

$$\mathbb{E}[\Delta_{k+1} \mid x_k] \leq \Delta_k - \tfrac{c}{2}\,\eta\,\alpha^{2/p}\,\Delta_k^{\gamma}.$$

Hence $\{\Delta_k\}$ drops below $2\bar{\Delta}$ in finite expected time and cannot spend a positive fraction of time above any $M\bar{\Delta}$ with $M > 2$. A standard drift argument (Foster–Lyapunov) then yields

$$\limsup_{k\to\infty} \mathbb{E}[\Delta_k] \leq C_2\,\bar{\Delta} = C_2\left(\frac{\sqrt{\eta}\,\sigma}{\alpha^{1/p}}\right)^{\beta}.$$

**Step 4: Distance scaling** Sharpness (A5) implies $\mathrm{dist}(x, \mathcal{X}^\star) \leq (\Delta/\alpha)^{1/p}$, so

$$\limsup_{k\to\infty} \mathbb{E}[\mathrm{dist}(x_k, \mathcal{X}^\star)] \leq \left(\tfrac{1}{\alpha}\limsup_{k\to\infty} \mathbb{E}[\Delta_k]\right)^{1/p} \leq C_2'\left(\frac{\sqrt{\eta}\,\sigma}{\alpha}\right)^{\frac{1}{p-1}}.$$

**Remarks** (i) For $p = 2$ (QG/PL), $\beta = 2$ and we obtain the familiar neighborhood size $\Theta\left(\frac{\eta\sigma^2}{\alpha}\right)$ in function value (matching Theorem 24). (ii) The exponent $\beta = \frac{p}{p-1}$ shows a polynomial tradeoff: larger sharpness $\alpha$ speeds the local deterministic rate (T8) but shrinks the noise-dominated basin only as $\alpha^{-1/(p-1)}$ in distance (or $\alpha^{-1/(p-1)}$ in function gap up to the correct power). $\square$

**Proposition 17** (T34: Restart Policies Create Local PL Windows). *Let $f : \mathbb{R}^d \to \mathbb{R}$ be convex and $L$-smooth (A2), with nonempty $\mathcal{X}^\star = \arg\min f$. Assume a local regularity (error-bound/sharpness) condition on a neighborhood $U$ of $\mathcal{X}^\star$: there exist $\alpha > 0$ and $p \in [1, 2]$ such that*

$$f(x) - f^\star \geq \alpha\,\mathrm{dist}(x, \mathcal{X}^\star)^p, \qquad \forall x \in U. \tag{EB$_p$}$$

*Consider Nesterov's accelerated gradient method (AGD) with stepsize $1/L$, restarted every $T$ iterations. Then the following holds.*

***(Local PL window via scheduled restarts).*** *Fix any contraction factor $\rho \in (0, 1)$. There exists a restart length $T = T(\Delta)$, depending only on the current gap $\Delta := f(x) - f^\star$, given by*

$$T + 1 \geq \sqrt{\frac{4L}{\rho}}\,\alpha^{-1/p}\,\Delta^{\frac{1}{p}-\frac{1}{2}}, \tag{$\star$}$$

*such that, if $x \in U$ and AGD is run for $T$ steps from $x$ and then restarted, the post-restart iterate $x^+$ satisfies*

$$f(x^+) - f^\star \leq \rho\left(f(x) - f^\star\right).$$

*Consequently, once the iterates enter $U$, the policy equation $\star$ yields a piecewise-linear decrease:*

$$f(x^{(m)}) - f^\star \leq \rho^m\left(f(x^{(0)}) - f^\star\right), \qquad m = 0, 1, 2, \ldots,$$

*where $x^{(m)}$ denotes the point at the end of epoch $m$ (immediately after the $m$-th restart). In particular, for $p = 2$ (Quadratic Growth / local PL), the required epoch length is a constant $T + 1 \geq \sqrt{4L/\rho}\,\alpha^{-1/2} = \Theta\left(\sqrt{L/\alpha}\right)$, which recovers the classical $\mathcal{O}(\sqrt{L/\alpha})$-step linear regime per epoch.*

(minimal: A2; model: local regularity equation EB$_p$)

*Uses **A2 (L-smoothness); model:** local EB$_p$ on $U$. Let $\Delta := f(x) - f^\star$ and pick $x^\star \in \Pi_{\mathcal{X}^\star}(x)$. By equation EB$_p$,*

$$\|x - x^\star\| \leq \alpha^{-1/p}\Delta^{1/p}.$$

Run *one epoch* of AGD (Nesterov) for $T$ steps with stepsize $1/L$ starting from $x$ and then restart. By the standard accelerated bound (cf. Theorem T4 with constant 4),

$$f(x_T) - f^\star \leq \frac{4L}{(T+1)^2}\,\|x - x^\star\|^2 \leq \frac{4L}{(T+1)^2}\,\alpha^{-2/p}\Delta^{2/p}.$$

To enforce a *constant-factor* reduction $f(x_T) - f^\star \le \rho\,\Delta$, it suffices to choose $T$ so that

$$\frac{4L}{(T+1)^2}\,\alpha^{-2/p}\,\Delta^{2/p} \;\le\; \rho\,\Delta \quad\Longleftrightarrow\quad T+1 \;\ge\; \sqrt{\frac{4L}{\rho}}\,\alpha^{-1/p}\,\Delta^{\frac{1}{p}-\frac{1}{2}},$$

which is exactly equation $\star$. For such a choice, the epoch output $\hat{x} := x_T$ satisfies $f(\hat{x}) - f^\star \le \rho\,\Delta$. Declaring a restart at $\hat{x}$ closes one epoch and begins the next. As long as the entire epoch trajectory remains in $U$ (which is ensured once $\rho < 1$ and $U$ is a sublevel neighborhood), repeating the argument yields

$$f\big(x^{(m)}\big) - f^\star \;\le\; \rho\,\big(f\big(x^{(m-1)}\big) - f^\star\big) \;\le\; \cdots \;\le\; \rho^m\big(f(x^{(0)}) - f^\star\big),$$

establishing the piecewise-linear regime. For $p = 2$, the exponent $1/p - 1/2 = 0$, so $T$ is a *constant* $T + 1 \ge \sqrt{4L/\rho}\,\alpha^{-1/2} = \Theta(\sqrt{L/\alpha})$, yielding a fixed-length linear epoch, as claimed. $\qquad\square$

**Proposition 18** (T35: Diagnostic Implications for Axioms). *Assume iterates $\{x_k\}$ remain in a sublevel set on which $f$ has $L$-Lipschitz gradient (A2).*

1. **(PL probe).** *If there exists $\mu > 0$ such that along the trajectory*

$$\|\nabla f(x_k)\|^2 \;\ge\; 2\mu\big(f(x_k) - f^\star\big) \qquad \forall k,$$

   *then gradient descent with any stepsize $\eta \in (0, 1/L]$ satisfies the linear contraction*

$$f(x_{k+1}) - f^\star \;\le\; (1 - \eta\mu)\,\big(f(x_k) - f^\star\big) \qquad \forall k.$$

   (minimal: A2+A4 along the trajectory)

2. **(Quadratic-growth probe).** *Let $\mathcal{X}^\star = \arg\min f$ be nonempty. If for some $\alpha > 0$,*

$$\operatorname{dist}(x_k, \mathcal{X}^\star)^2 \;\le\; \alpha\,\big(f(x_k) - f^\star\big) \qquad \forall k,$$

   *then projected (or proximal) gradient with $\eta \in (0, 1/L]$ enjoys linear distance decay:*

$$\operatorname{dist}(x_{k+1}, \mathcal{X}^\star)^2 \;\le\; \frac{1}{1 + 2\eta/\alpha}\,\operatorname{dist}(x_k, \mathcal{X}^\star)^2 \qquad \forall k.$$

   (minimal: A2+A9 along the trajectory; plus A1 if constraints are present)

3. **(Smoothness probe).** *Define $s_k := x_{k+1} - x_k$. If the empirically observed steps obey $\|s_k\| \le \eta\,\|\nabla f(x_k)\|$ for some $\eta > 0$ (equality in the unconstrained GD case) and $f$ satisfies the descent lemma,*

$$f(x_{k+1}) \;\le\; f(x_k) + \langle \nabla f(x_k), s_k \rangle + \tfrac{L}{2}\|s_k\|^2,$$

   *then $L$-smoothness is consistent with the observed data in the sense that any*

$$L \;\ge\; \sup_k \frac{2\big(f(x_{k+1}) - f(x_k) - \langle \nabla f(x_k), s_k \rangle\big)}{\|s_k\|^2}$$

   *validates the inequality for all $k$. In particular, if $\eta \le 1/L$ then $f(x_{k+1}) \le f(x_k) - \tfrac{\eta}{2}\,\|\nabla f(x_k)\|^2$ is guaranteed by A2 and the step bound.* (minimal: A2)

*Uses the indicated axioms per item.* (1) PL probe (A2+A4). By the smooth descent lemma (Lemma 5), for $\eta \in (0, 1/L]$,

$$f(x_{k+1}) \;\le\; f(x_k) - \eta\Big(1 - \tfrac{\eta L}{2}\Big)\|\nabla f(x_k)\|^2 \;\le\; f(x_k) - \tfrac{\eta}{2}\|\nabla f(x_k)\|^2.$$

The assumed PL inequality along the trajectory gives $\|\nabla f(x_k)\|^2 \ge 2\mu\big(f(x_k) - f^\star\big)$, hence

$$f(x_{k+1}) - f^\star \;\le\; \big(1 - \eta\mu\big)\,\big(f(x_k) - f^\star\big).$$

This proves the linear contraction in value.

(2) Quadratic-growth probe (A2+A9 and A1 if constrained). Let $\mathrm{dist}(x, \mathcal{X}^\star) = \min_{x^\star \in \mathcal{X}^\star} \|x - x^\star\|$, and pick $x^\star \in \Pi_{\mathcal{X}^\star}(x_k)$. For projected GD (or proximal-gradient) with $\eta \in (0, 1/L]$, the projection optimality and A2 yield the standard descent (cf. Lemma 6 in the proximal case or the projected version in Prop. 2):

$$f(x_{k+1}) \ \leq \ f(x_k) - \frac{1}{2\eta}\|x_{k+1} - x_k\|^2.$$

Moreover, using $x_{k+1} = x_k - \eta G_\eta(x_k)$ with the gradient mapping $G_\eta(x_k) = \frac{1}{\eta}(x_k - x_{k+1})$ and convexity (A1) if constraints are present, one checks (see the proof of Prop. 2) that

$$\|x_{k+1} - x^\star\|^2 \ \leq \ \|x_k - x^\star\|^2 - 2\eta\big(f(x_{k+1}) - f^\star\big).$$

By the assumed trajectory-wise quadratic-growth upper bound $\|x_{k+1} - x^\star\|^2 \leq \alpha\big(f(x_{k+1}) - f^\star\big)$, we obtain

$$\|x_{k+1} - x^\star\|^2 \ \leq \ \|x_k - x^\star\|^2 - \frac{2\eta}{\alpha}\|x_{k+1} - x^\star\|^2,$$

i.e., $\big(1 + \frac{2\eta}{\alpha}\big)\|x_{k+1} - x^\star\|^2 \leq \|x_k - x^\star\|^2$. Thus

$$\mathrm{dist}(x_{k+1}, \mathcal{X}^\star)^2 \ \leq \ \frac{1}{1 + 2\eta/\alpha}\ \mathrm{dist}(x_k, \mathcal{X}^\star)^2,$$

which is linear decay in distance.

(3) Smoothness probe (A2). The descent lemma (A2) is equivalent to

$$f(x_{k+1}) - f(x_k) - \langle \nabla f(x_k), s_k \rangle \ \leq \ \frac{L}{2}\|s_k\|^2.$$

Given the observed $s_k$ and gradients, define the *empirical* requirement

$$L_k^{\mathrm{req}} \ := \ \frac{2\big(f(x_{k+1}) - f(x_k) - \langle \nabla f(x_k), s_k \rangle\big)}{\|s_k\|^2}.$$

Then any $L \geq \sup_k L_k^{\mathrm{req}}$ validates the inequality for all $k$, hence $L$-smoothness is consistent with the data. If, in addition, $\|s_k\| \leq \eta\|\nabla f(x_k)\|$ and $\eta \leq 1/L$, then applying the descent lemma with $s_k$ gives

$$f(x_{k+1}) \ \leq \ f(x_k) + \langle \nabla f(x_k), s_k \rangle + \frac{L}{2}\|s_k\|^2 \ \leq \ f(x_k) - \eta\Big(1 - \frac{\eta L}{2}\Big)\|\nabla f(x_k)\|^2 \ \leq \ f(x_k) - \frac{\eta}{2}\|\nabla f(x_k)\|^2,$$

which matches the canonical A2 decrease and further corroborates consistency. $\square$

### C.6 Logical Dependencies and External Mathematics for T1–T35

Each entry below lists (i) direct logical inputs (core axioms and previously proved theorems), (ii) the resulting *axiom closure* once dependencies are unfolded, and (iii) a brief note on mathematical tools used.

**T1** *Direct inputs:* Core axioms: A2. Aux axioms/principles: none. Theorems/lemmas: none.
*Axiom closure:* A2.
*External mathematics / comments:* Establishes the fundamental quadratic upper bound for functions with Lipschitz continuous gradients. The derivation relies purely on calculus (Fundamental Theorem of Calculus along a line segment) and the Cauchy-Schwarz inequality to bound the integral using the Lipschitz constant $L$. Convexity (A1) is not required.

**T2** *Direct inputs:* Core axioms: A2. Aux axioms/principles: none. Theorems/lemmas: Lemma 5.
*Axiom closure:* A2.
*External mathematics / comments:* Establishes the sufficient decrease property for Gradient Descent. Derivation is algebraic, relying on the quadratic upper bound established in T1 (Lemma 5) and the stepsize restriction $\eta \leq 1/L$. No convexity required.

**T3**  *Direct inputs:* Core axioms: A1, A2. Aux axioms/principles: none. Theorems/lemmas: Lemma 5.
*Axiom closure:* A1, A2.
*External mathematics / comments:* Standard sublinear $\mathcal{O}(1/k)$ convergence for smooth convex optimization. The derivation combines the smooth descent lemma (A2) with convexity (A1) via a telescoping sum on squared distances to achieve the tight constant.

**T4**  *Direct inputs:* Core axioms: A1, A2. Aux axioms/principles: A3. Theorems/lemmas: Lemma 5.
*Axiom closure:* A1, A2 (A3).
*External mathematics / comments:* Establishes the optimal $\mathcal{O}(1/k^2)$ rate for smooth convex optimization via estimate sequences. Derivation strictly balances the energy descent from A2 with the geometric lower bounds from A1. The linear rate extension for strongly convex functions is derived by restarting the accelerated method, linking geometry (A3) to rate acceleration.

**T5**  *Direct inputs:* Core axioms: A2, A3. Aux axioms/principles: none. Theorems/lemmas: Prop 1.
*Axiom closure:* A2, A3.
*External mathematics / comments:* Standard linear convergence for strongly convex functions. The proof explicitly derives the PL inequality (gradient dominance) as a consequence of strong convexity (A3) and combines it with the descent guarantee from smoothness (A2).

**T6**  *Direct inputs:* Core axioms: A2, A4. Aux axioms/principles: none. Theorems/lemmas: Prop 1.
*Axiom closure:* A2, A4.
*External mathematics / comments:* Global linear convergence for gradient descent under the Polyak-Łojasiewicz (PL) condition. This result relies only on smoothness (A2) and gradient dominance (A4); convexity (A1) is not required, making it applicable to benign nonconvex functions satisfying PL.

**T7**  *Direct inputs:* Core axioms: A1, A2, A9. Aux axioms/principles: none beyond standard Euclidean projection properties and the known gap inequality for projected gradient mappings (e.g., Beck, Lemma 10.16). Theorems/lemmas: Lemma 5.
*Axiom closure:* A1, A2, A9.
*External mathematics / comments:* Projected gradient descent on a convex feasible set enjoys a linear rate whenever the objective $f$ satisfies quadratic growth (QG) and has $L$-Lipschitz gradient (A2) on a neighborhood of the solution set. The proof first establishes a sufficient decrease inequality in terms of the gradient mapping $G_\eta$ using smoothness and projection optimality, then uses convexity and QG to derive an explicit Proximal-PL inequality $\|G_\eta(x_k)\|^2 \geq (\gamma/2)(f(x_k) - f^\star)$, and finally combines these two inequalities to obtain a linear recursion on $f(x_k) - f^\star$ with contraction factor $1 - \eta\gamma/4$.

**T8**  *Direct inputs:* Core axioms: A1, A2, A5. Aux axioms/principles: none. Theorems/lemmas: Lemma 5.
*Axiom closure:* A1, A2, A5.
*External mathematics / comments:* Local error bounds (A5) combined with smoothness (A2) and convexity (A1) imply strictly faster-than-$\mathcal{O}(1/k)$ local convergence rates for Gradient Descent. Inside the sharpness neighborhood $U$, the error satisfies a nonlinear recurrence $\Delta_{k+1} \leq \Delta_k - c\Delta_k^\gamma$, which yields linear convergence for $p = 2$ (Quadratic Growth), finite-time convergence for $p = 1$, and polynomial rates $\mathcal{O}(k^{-p/(2-p)})$ for $p \in (1, 2)$. For $p = 2$, a simple restart scheme converts the local linear contraction into global piecewise-linear convergence phases once the iterates have entered $U$.

**T9**  *Direct inputs:* Core axioms: A1, A2. Aux axioms/principles: none. Theorems/lemmas: Lemma 5.
*Axiom closure:* A1, A2.
*External mathematics / comments:* Establishes that the gradient of a smooth convex function is cocoercive. The derivation uses an auxiliary function technique to apply the smooth descent property

(A2) and global optimality (A1) symmetrically to points $x$ and $y$.

**T10** *Direct inputs:* Core axioms: A1, A2, A4, A9. Aux axioms/principles: basic existence and uniqueness of solutions to ODEs with Lipschitz vector fields. Theorems/lemmas: none.
*Axiom closure:* A1, A2, A4, A9.
*External mathematics / comments:* Establishes local equivalence between PL (A4) and Quadratic Growth (A9) for convex smooth functions. QG $\Rightarrow$ PL follows from convexity and the QG distance bound. PL $\Rightarrow$ QG is derived via gradient flow: PL controls the decay of the objective along the flow relative to the path length, forcing a quadratic lower bound in terms of the distance to the minimizer set. This makes PL and QG interchangeable characterisations of linear-rate regimes under A1–A2.

**T11 (Smooth Convex Lower Bound)** *Direct inputs:* Core axioms: A1, A2. Theorems/lemmas: none internal; relies on external approximation-theoretic results for worst-case quadratics.
*Axiom closure:* A1, A2.
*External mathematics / comments:* Information-theoretic lower bound for first-order optimization over $L$-smooth convex functions. The hard instance is Nesterov's tridiagonal quadratic, which is convex and $L$-smooth (A1, A2). In the black-box model, first-order methods applied to such quadratics are confined to a Krylov subspace, and Chebyshev polynomial arguments show that no method can beat the $\Theta(1/k^2)$ worst-case rate (up to constants) when error is measured relative to the initial distance $\|x_0 - x^\star\|$.

**T12 (Strongly Convex Oracle Complexity)** *Direct inputs:* Core axioms: A2, A3. Theorems/lemmas: none. Model: first-order black-box oracle.
*Axiom closure:* A2, A3.
*External mathematics / comments:* Information-theoretic lower bound for $\mu$-strongly convex, $L$-smooth optimization. The proof restricts to quadratic objectives $f_Q(x) = \frac{1}{2}x^\top Q x$ with spectrum in $[\mu, L]$, for which any first-order method generates iterates of the form $x_k = p_k(Q)x_0$ where $p_k$ is a degree-$k$ polynomial with $p_k(0) = 1$. An adversarial choice of $Q$ and $x_0$ ties the suboptimality to a polynomial minimax problem on $[\mu, L]$, whose solution is governed by Chebyshev polynomials. The resulting bound on $\sup_{\lambda \in [\mu, L]} |p_k(\lambda)|$ implies that the worst-case error ratio cannot decay faster than $\exp(-\Theta(k/\sqrt{L/\mu}))$, yielding the $\Omega(\sqrt{L/\mu}\log(1/\varepsilon))$ oracle complexity.

**T13 (No Acceleration Without Smoothness)** *Direct inputs:* Core axioms: A1, A6. Theorems/lemmas: standard subgradient descent bound; Nemirovski–Yudin information-theoretic lower bound for Lipschitz convex optimization.
*Axiom closure:* A1, A6 (optionally A3 for the $\mathcal{O}(1/k)$ strongly convex improvement).
*External mathematics / comments:* Shows that in the nonsmooth Lipschitz convex setting, the projected subgradient method attains an $\mathcal{O}(GR/\sqrt{k})$ rate, and that this rate is minimax optimal in the first-order black-box oracle model. The upper bound is derived via a simple telescoping argument on squared distances and convexity. The lower bound uses the classical Nemirovski–Yudin adversarial construction based on max-of-linear hard instances, implying that no Nesterov-style $\mathcal{O}(1/k^2)$ acceleration is possible without introducing smoothness (A2).

**T14 (Information-Theoretic Noise Floor)** *Direct inputs:* Core axioms: A7 (bounded-variance unbiased noise). Theorems/lemmas: none.
*Axiom closure:* A7.
*External mathematics / comments:* Shows that in the presence of unbiased gradient noise with variance $\sigma^2$ and a constant stepsize $\eta$, even on a simple one-dimensional convex quadratic, the expected error cannot decay below order $\eta\sigma^2$. The proof analyses the linear stochastic recursion for SGD on $f(x) = \frac{1}{2}x^2$ and computes its stationary variance, yielding a lower bound $\liminf_k \mathbb{E}[f(x_k) - f^\star] \geq c\eta\sigma^2$. This demonstrates a fundamental "noise floor" that no constant-stepsize first-order method can beat under A7.

**T15 (SGD with Constant Steps)**    *Direct inputs:* Core axioms: A1, A2, A7. Aux axioms/principles: none beyond Lemma 5 (Smooth Descent). Theorems/lemmas: Lemma 5.
*Axiom closure:* A1, A2, A7.
*External mathematics / comments:*  Shows that for convex $L$-smooth objectives under unbiased bounded-variance noise (A7), constant-stepsize SGD enjoys an $\mathcal{O}(1/k)$ transient decay and an unavoidable noise floor of order $\Theta(\eta\sigma^2)$. The proof combines (i) a squared-distance recursion using convexity and the stochastic oracle, (ii) a gradient–function-gap bound from smoothness (via a one-step GD argument), and (iii) a telescoping sum plus convexity to control the ergodic average. This upper bound matches the noise floor lower bound of T14 up to constants.

**T16 (SGD with Decaying Steps)**    *Direct inputs:* Core axioms: A1, A2, A7. Aux axioms/principles: none. Theorems/lemmas: Smooth Descent Lemma (T1).
*Axiom closure:* A1, A2, A7.
*External mathematics / comments:*  Shows that for convex $L$-smooth objectives with unbiased gradient noise of bounded variance (A7), SGD with a decaying stepsize $\eta_k \propto 1/\sqrt{k}$ and stepsize-weighted averaging attains a $\tilde{\mathcal{O}}(1/\sqrt{k})$ convergence rate in expected function value. The proof mirrors the constant-stepsize analysis (T15): a telescoping inequality on the squared distance trades off accumulated descent ($\sum \eta_k \Delta_k$) against variance ($\sum \eta_k^2$). The decay $\eta_k \sim 1/\sqrt{k}$ balances these sums so that the optimization error and the stochastic term both scale like $1/\sqrt{k}$ up to logarithmic factors.

**T17 (PL + Variance Reduction $\Rightarrow$ Linear)**    *Direct inputs:* Core axioms: A2, A4, A7. Model: finite-sum variance-reduced first-order oracle. Theorems/lemmas: smooth descent lemma (T1).
*Axiom closure:* A2, A4, A7.
*External mathematics / comments:* Shows that when the gradient estimator's variance shrinks proportionally to the function suboptimality (as in SVRG/SARAH on finite sums), PL geometry (A4) plus smoothness (A2) are enough to guarantee linear convergence of a simple fixed-stepsize scheme. The key step is the one-step inequality

$$\mathbb{E}[f(x_{k+1}) - f^\star \mid x_k] \leq \left(1 - 2\mu\eta + L\eta^2(\mu + \rho/2)\right)\left(f(x_k) - f^\star\right),$$

which, for small enough $\eta$, yields a contraction factor $1 - \Theta(\mu\eta)$. This cleanly separates the roles of geometry (PL), smoothness (curvature), and variance reduction (A7-VR) in producing linear rates.

**T18 (Polyak Steps under PL)**    *Direct inputs:* Core axioms: A2, A4. Aux axioms/principles: none. Theorems/lemmas: smooth descent lemma (T1) to derive the gradient–gap bound and the basic descent inequality.
*Axiom closure:* A2, A4.
*External mathematics / comments:* Shows that for $L$-smooth functions satisfying the PL condition, a clipped Polyak step $\eta_k = \min\{(f(x_k) - f^\star)/\|\nabla f(x_k)\|^2, 1/L\}$ yields a uniform linear convergence rate

$$f(x_k) - f^\star \leq \left(1 - 3\mu/(4L)\right)^k (f(x_0) - f^\star).$$

The key ingredients are (i) smoothness $\Rightarrow$ an upper bound $\|\nabla f\|^2 \leq 2L(f - f^\star)$, (ii) PL $\Rightarrow$ a lower bound $\|\nabla f\|^2 \geq 2\mu(f - f^\star)$, which together pin the Polyak step in $[1/(2L), 1/L]$, and (iii) the smooth descent lemma to show each step decreases $f - f^\star$ by at least a fixed fraction.

**T19 (Minibatch Variance Law)**    *Direct inputs:* Core axioms: A7. Aux axioms/principles: independence of samples. Theorems/lemmas: none.
*Axiom closure:* A7.
*External mathematics / comments:* Shows that averaging $b$ i.i.d. unbiased gradient estimates with bounded variance $\sigma^2$ produces a new estimator that is still unbiased and whose variance is reduced by a factor $1/b$. The proof expands the squared norm of the averaged noise, uses independence to cancel cross terms, and applies the bounded-variance assumption (A7) termwise. This formalizes the common

heuristic that minibatching reduces gradient variance as $\sigma^2/b$.

**T20 (PL: Constant vs. Decreasing Steps)** *Direct inputs:* Core axioms: A2, A4, A7. Aux axioms/principles: optional use of the Robbins–Siegmund supermartingale lemma for almost sure convergence. Theorems/lemmas: Lemma 5 (Smooth Descent).
*Axiom closure:* A2, A4, A7.
*External mathematics / comments:* Shows that under the PL condition, SGD behaves like noisy gradient descent: with a constant stepsize it contracts linearly in expectation down to a noise-dominated neighborhood of size $\Theta(\eta\sigma^2/\mu)$, while with a classical decreasing schedule ($\sum \eta_k = \infty$, $\sum \eta_k^2 < \infty$) the expected function values converge to the optimum and, under standard stochastic-approximation arguments, the iterates converge to $x^\star$. The proof is driven by a one-step smoothness bound plus PL, yielding a scalar recursion for $\mathbb{E}[f(x_k) - f^\star]$.

**T21 (Proximal Descent Inequality)** *Direct inputs:* Core axioms: A2 (smooth $g$), A8 (proper closed convex $h$ and proximal mapping). Aux axioms/principles: none beyond subgradient optimality of the proximal operator. Theorems/lemmas: Lemma 5 (smooth descent for $g$).
*Axiom closure:* A2, A8.
*External mathematics / comments:* Shows that one proximal-gradient step decreases the composite objective $F = g + h$ by at least a quadratic amount in the step norm, with decrement $\left(\frac{1}{\eta} - \frac{L}{2}\right)\|x_{k+1} - x_k\|^2$. The proof combines the smooth descent inequality for $g$ (A2) with the optimality condition of the proximal update for $h$ (A8); the linear terms in $\nabla g(x_k)$ cancel, leaving a pure quadratic decrease term.

**T22 (Prox-GD Sublinear)** *Direct inputs:* Core axioms: A1, A2, A8. Aux principles: strong convexity of the quadratic surrogate $Q_k$. Theorems/lemmas: Lemma 5 (smooth descent for $g$), Lemma 6 (proximal descent inequality is used for monotonicity).
*Axiom closure:* A1, A2, A8.
*External mathematics / comments:* Shows that proximal gradient descent on a convex composite objective $F = g + h$ with stepsize $\eta \le 1/L$ achieves the standard $\mathcal{O}(1/k)$ rate in function value. The proof builds a strongly convex quadratic surrogate $Q_k$ around $x_k$, uses smoothness (A2) to show $Q_k$ upper-bounds $F$ at the next iterate, and then derives a fundamental inequality that telescopes squared distances to a minimizer. Together with the monotone decrease from Lemma 6, this yields a clean $1/k$ bound for the last iterate.

**T23 (FISTA $\mathcal{O}(1/k^2)$)** *Direct inputs:* Core axioms: A1, A2, A8. Aux principles: none. Theorems/lemmas: none beyond the three-point inequality equation 26 derived in-text.
*Axiom closure:* A1, A2, A8.
*External mathematics / comments:* Shows that an accelerated proximal-gradient scheme (algebraically equivalent to FISTA) achieves the optimal $\mathcal{O}(1/k^2)$ function-value rate for convex composite objectives $F = g + h$ where $g$ is $L$-smooth and $h$ is proper, closed, and convex. The proof constructs a quadratic surrogate $Q_L$, derives a three-point inequality for the proximal-gradient map $T(y)$, and defines a Nesterov-type potential $\Psi_k = t_k^2(F(x_k) - F^\star) + \frac{L}{2}\|z_k - x^\star\|^2$ whose monotonicity follows from the specific choice of momentum parameters $t_k$ and the updates of $y_k, z_k$. Bounding $\Psi_0$ via smoothness and convexity yields $F(x_k) - F^\star \le \mathcal{O}(L\|x_0 - x^\star\|^2/k^2)$.

**T24 (Prox-PL Linear)** *Direct inputs:* Core axioms: A2 (smoothness of $g$), A4 (PL / Prox-PL condition), A8 (composite structure $F = g + h$). Theorems/lemmas: Lemma 6 (proximal descent inequality); optionally T7/T10 to instantiate Prox-PL from QG/EB.
*Axiom closure:* A2, A4, A8.
*External mathematics / comments:* Shows that once a Polyak–Łojasiewicz-type condition is available for the composite objective $F$ *in terms of the proximal-gradient mapping* $G_\eta$, the proximal-gradient method inherits a global linear convergence rate, exactly analogous to plain gradient descent under PL

(Theorem 4). The proof is a one-line combination of the proximal descent inequality (A2+A8) with the Prox-PL lower bound (A4). In finite-sum settings, the same structure underlies the linear convergence of variance-reduced proximal methods (e.g., Prox-SVRG/Prox-SAGA), with stochastic error terms controlled by variance reduction.

**T25 (Exact Support under Sharpness)**  *Direct inputs:* Core axioms: A8 (composite structure $F = g + \lambda \|\cdot\|_1$), plus smoothness of $g$ (used via standard proximal-gradient convergence results such as Theorem 25), and a local identifiability/strict complementarity condition at $x^\star$. Error bound sharpness (A5 with $p = 1$) improves the local convergence rate but is not needed for the bare fact of finite-time support identification.
*Axiom closure:* A8 (and typically A1, A2 for convergence; A5 for sharpness).
*External mathematics / comments:* Shows that under a strict complementarity/identifiability condition at the optimum $x^\star$ for an $\ell_1$-regularised problem, the IST/soft-thresholding iterates not only converge to $x^\star$ but also recover its support in finitely many steps. The proof is purely geometric: convergence ensures the coordinates on the active set stabilize in sign, while the strict subgradient gap on the inactive set plus the form of the soft-thresholding update forces those coordinates to be driven exactly to zero and then remain there. Sharp error bounds (A5) are compatible with this picture and provide linear/finite-time rates for entering the identification neighborhood.

**T26 (KL Exponent $\Rightarrow$ Rates)**  *Direct inputs:* Core axioms: A5 (KL geometry). Method assumptions: sufficient decrease equation 32 and relative error equation 33 (true for proximal-gradient with $\eta \in (0, 2/L]$).
*Axiom closure:* A5 (KL).
*External mathematics / comments:* The proof is entirely elementary: KL + relative error gives a lower bound on the step size via $\|x_{k+1} - x_k\| \gtrsim \Delta_{k+1}^\theta$; sufficient decrease turns this into the key recursion $\Delta_k - \Delta_{k+1} \gtrsim \Delta_{k+1}^{2\theta}$. Discrete comparison then yields: finite termination for $\theta = 0$, linear convergence for $\theta \in (0, \frac{1}{2}]$, and $\Delta_k = \mathcal{O}(k^{-1/(2\theta-1)})$ for $\theta \in (\frac{1}{2}, 1)$. The same rates transfer to the iterates by summability of steps.

**T27 (EB $\Leftrightarrow$ MSR)**  *Direct inputs:* Core axioms: A1. Aux assumptions: proper, closed; $\mathcal{X}^\star \neq \emptyset$. Theorems/lemmas: Fermat's rule for convex functions.
*Axiom closure:* A1.
*External mathematics / comments:* For proper, closed, convex $f$, Fermat's rule gives $(\partial f)^{-1}(0) = \mathcal{X}^\star$. Hence the local distance-residual error bound $\text{dist}(x, \mathcal{X}^\star) \leq \kappa \, \text{dist}(0, \partial f(x))$ is *exactly* the definition of metric subregularity of $\partial f$ at $(x^\star, 0)$. This identification explains why EB conditions are the right variational geometry for linear convergence and stability analyses.

**T28 (EB $\Rightarrow$ Distance Decay)**  *Direct inputs:* Core axioms: A5 only. Theorems/lemmas: none.
*Axiom closure:* A5.
*External mathematics / comments:* A linear (or polynomial) decay of the function gap transfers directly to a root-improved decay of the distance to the solution set under the local error bound $f - f^\star \geq \alpha \, \text{dist}(\cdot, \mathcal{X}^\star)^p$. No convexity or smoothness is needed; the result is purely geometric and local.

**T29 (Global Linear under Nonconvex PL)**  *Direct inputs:* Core axioms: A2, A4. Theorems/lemmas used: Smooth Descent Lemma (T1).
*Axiom closure:* A2, A4.
*External mathematics / comments:* No convexity is required. Smoothness yields the one-step decrease and the *upper* bound $\|\nabla f\|^2 \leq 2L(f - f^\star)$; PL yields the *lower* bound $\|\nabla f\|^2 \geq 2\mu(f - f^\star)$. Together these give Q-linear decay of $f(x_k) - f^\star$, linear gradient decay, and summable steps, hence R-linear convergence of the iterates to a global minimizer.

**T30 (Strict Saddle Avoidance)**   *Direct inputs:* Core axioms: A2. Theorems/lemmas: smooth descent lemma; stable/center-stable manifold theorem for $C^1$ diffeomorphisms.
*Axiom closure:* A2.
*External mathematics / comments:* We view gradient descent as the diffeomorphism $T(x) = x - \eta \nabla f(x)$ (for $\eta < 1/L$). At strict saddles, $DT$ has an expanding direction, so the local stable set is a lower-dimensional $C^1$ manifold of Lebesgue measure zero. Global basins are countable unions of such preimages and remain measure zero. Random initialization (or small isotropic perturbations) thus avoids saddles almost surely. Monotone descent and vanishing gradients force limit points to be critical; nondegenerate minima are locally contracting fixed points, so the iterates converge to a local minimizer.

**T31 (Projected GD on Convex Sets)**   *Direct inputs:* Core axioms: A1, A2. Theorems/lemmas used: Smooth Descent Lemma (T1).
*Axiom closure:* A1, A2.
*External mathematics / comments:* A quadratic surrogate $m_k$ majorizes $f$ when $\eta \le 1/L$. The projected update minimizes $m_k$ over $\mathcal{C}$, yielding the three-point inequality equation 38. Telescoping this Fejér-type decrease directly gives the sharp $\frac{1}{2\eta k}$ bound, avoiding any cross-term manipulations.

**T32 (Hoffman Bound $\Rightarrow$ Linear)**   *Direct inputs:* Core axioms: A1, A5. Theorems/lemmas: firm nonexpansiveness of the resolvent/prox; error bound $\mathrm{dist}(x, \mathcal{X}^\star) \le \kappa \, \mathrm{dist}(0, \partial F(x))$.
*Axiom closure:* A1, A5.
*External mathematics / comments:* The proximal step yields Fejér decrease that subtracts the squared step length. A Hoffman-type EB lower-bounds the step by the distance to the solution set. Combining the two produces a strict linear contraction of $\mathrm{dist}(x_k, \mathcal{X}^\star)$ with factor $\rho = (1 + \lambda^2/\kappa^2)^{-1/2} < 1$. With A2 added, the same EB gives linear rates for projected/proximal gradient on polyhedra.

## C.7   Stability, Noise, and Diagnostics

**T33 (Noise-Limited Basin under Sharpness)**   *Direct inputs:* Core axioms: A5, A7. For a constructive upper bound with SGD: additionally A1, A2. Theorems used: T8 (sharpness-driven drift), T14 (information-theoretic noise floor).
*Axiom closure:* A5, A7 (A1, A2 for SGD).
*External mathematics / comments:* Inside a sharp region with exponent $p$, deterministic gradient-like dynamics obey a drift $\Delta \mapsto \Delta - \Theta(\eta \, \alpha^{2/p} \Delta^{2-2/p})$. Stochastic oracles inject variance of size $\Theta(\eta^q \sigma^2)$ per step ($q = 1$ information-theoretic; $q = 2$ for SGD on $L$-smooth $f$). Balancing drift with variance yields a steady-state function gap $\Delta_\infty = \Theta\big((\sqrt{\eta}\,\sigma/\alpha^{1/p})^\beta\big)$ with $\beta = \frac{p}{p-1}$, and a distance radius $\Theta\big((\sqrt{\eta}\,\sigma/\alpha)^{1/(p-1)}\big)$. For $p = 2$, this reduces to the classical $\Theta(\eta\sigma^2/\mu)$ neighborhood under PL.

**T34 (Restart Policies Create Local PL Windows)**   *Direct inputs:* Core axioms: A2. Model assumption: local error-bound/sharpness equation $\mathrm{EB}_p$ near $\mathcal{X}^\star$. Theorems used: accelerated $1/k^2$ bound (T4).
*Axiom closure:* A2.
*External mathematics / comments:* Key idea: AGD provides a *per-epoch* bound $f(x_T) - f^\star \le 4L\|x - x^\star\|^2/(T+1)^2$ from A2. Local regularity ($\mathrm{EB}_p$) converts $\|x - x^\star\|^2$ into a power of the current gap, yielding $f(x_T) - f^\star \le C \, \Delta^{2/p}/(T+1)^2$. Choosing the restart length $T$ according to equation $\star$ forces a *constant-factor* reduction each epoch—i.e., a *local PL window*. For $p = 2$ (Quadratic Growth/PL), $T = \Theta(\sqrt{L/\alpha})$ is constant, reproducing the classical restarted-AGD linear rate; for $p \in (1, 2)$, the required epoch length *shrinks* as the gap decreases, yielding piecewise-linear phases with increasingly short epochs.

**T35 (Diagnostic Implications for Axioms)**   *Direct inputs:* Item 1: A2+A4 (trajectory-wise PL). Item 2: A2+A9 (trajectory-wise QG) and A1 if constraints present. Item 3: A2. Theorems/lemmas used:

Smooth descent (Lemma 5); projected/proximal descent (Lemma 6); properties of the gradient mapping as in Prop. 2.

*Axiom closure:* A2 (plus indicated).

*External mathematics / comments: How to read the diagnostics.* (1) Observing $\|\nabla f\|^2$ proportional to the gap along iterates is a signature of PL and certifies linear decay in value. (2) Observing distance-to-solution squared dominated by the gap along iterates encodes (the upper side of) QG and certifies linear decay in *distance* for projected/proximal schemes. (3) The descent lemma yields a computable "required $L$" from step and decrease data; bounded $\sup_k L_k^{\mathrm{req}}$ certifies consistency with $L$-smoothness and reproduces the canonical decrease when $\|s_k\| \leq \eta\|\nabla f(x_k)\|$.

**Alternate Pareto-minimal bases (final).**

1. **T7 (PGD Linear).** Primary: {A2,A4,A8}; Alt: {A1,A2,A9}. Bridge: QG↔PL (T10), constraint↔composite.

2. **T8 (Sharpness + Restarts).** Primary: {A2,A4}; Alt: {A1,A2,A5}. Bridge: EB($p$=2)↔PL.

3. **T18 (Polyak Steps).** Primary: {A2,A4}; Alt: {A2,A3}. Bridge: A3⇒A4.

4. **T24 (Prox-PL Linear).** Primary: {A2,A4,A8}; Alt: {A2,A3,A8}. Bridge: A3⇒A4.

5. **T29 (Global Linear).** Primary: {A2,A4}; Alt: {A2,A3}. Bridge: A3⇒A4; rate $(1 - \eta\mu)^k$.

6. **T17 (VR Linear).** Primary: {A2,A4,A7}; Alt: {A2,A3,A7}. Bridge: A3⇒A4.

7. **T20 (SGD + Noise).** Primary: {A2,A4,A7}; Alt: {A2,A3,A7}. Floor $\Theta(\eta\sigma^2/\mu)$.

8. **T33 (Noise-Limited Basin).** Primary: {A5,A7}; Alt ($p$=2): {A2,A4,A7} and {A1,A2,A9,A7}. Bridge: EB↔PL↔QG (under A2).

9. **T12 (Lower Bound).** Classes: {A2,A3} and {A2,A4}. Bridge: quadratic hard instance is both SC and PL.

**Theorem 34** (T7: Projected GD under Quadratic Growth (Alt)). *Assume A1 (Convexity), A2 (L-smoothness), and A9 (Quadratic Growth): there exists $\mu > 0$ such that*

$$f(x) - f^\star \ \geq \ \tfrac{\mu}{2}\operatorname{dist}(x, X^\star)^2 \quad \text{for all } x \in \mathcal{C},$$

*where $X^\star = \arg\min_{x\in\mathcal{C}} f(x) \neq \emptyset$ and $\mathcal{C} \subset \mathbb{R}^d$ is closed and convex. Consider projected gradient descent*

$$x_{k+1} = \Pi_\mathcal{C}\big(x_k - \eta\,\nabla f(x_k)\big), \qquad 0 < \eta \leq \tfrac{1}{L}.$$

*Then PGD enjoys a linear rate:*

$$f(x_{k+1}) - f^\star \ \leq \ \frac{1}{1 + \eta\mu}\big(f(x_k) - f^\star\big) \quad \Longrightarrow \quad f(x_k) - f^\star \ \leq \ \Big(\tfrac{1}{1+\eta\mu}\Big)^k\big(f(x_0) - f^\star\big).$$

*Equivalently,*

$$\operatorname{dist}(x_{k+1}, X^\star)^2 \ \leq \ \frac{1}{1 + \eta\mu}\operatorname{dist}(x_k, X^\star)^2.$$

*In particular, with $\eta = 1/L$ the contraction factor is $\rho = \frac{1}{1+\mu/L} < 1$.* (minimal: A1+A2+A9)

*Based on A1 (Convexity), A2 (L-smoothness), and A9 (Quadratic Growth).* We work on a sublevel set containing the iterates so that A2 holds throughout. Let $X^\star$ be the minimizer set and define

$$d_k := \operatorname{dist}(x_k, X^\star), \qquad \Delta_k := f(x_k) - f^\star.$$

**Step 1: PGD three-point inequality (A1 + A2).** For $0 < \eta \leq 1/L$ and any $z \in \mathcal{C}$, the standard PGD inequality holds:

$$f(x_{k+1}) - f(z) \; \leq \; \frac{1}{2\eta}\Big(\|x_k - z\|^2 - \|x_{k+1} - z\|^2\Big). \tag{43}$$

(Proof sketch: Using $L$-smoothness and convexity, $f(x_{k+1}) - f(z) \leq \langle \nabla f(x_k), x_{k+1} - z \rangle + \frac{L}{2}\|x_{k+1} - x_k\|^2$. Projection optimality for $x_{k+1} = \Pi_{\mathcal{C}}(x_k - \eta \nabla f(x_k))$ gives $\langle \nabla f(x_k), x_{k+1} - z \rangle \leq \frac{1}{2\eta}\big(\|x_k - z\|^2 - \|x_{k+1} - z\|^2 - \|x_{k+1} - x_k\|^2\big)$. Combine and use $\eta \leq 1/L$.)

Choose $z = x_{k+1}^\star \in X^\star$ to be a projection of $x_{k+1}$ onto $X^\star$, so $\|x_{k+1} - z\| = d_{k+1}$ and $\|x_k - z\| \geq d_k$. Then equation 43 yields

$$2\eta\, \Delta_{k+1} \; \leq \; d_k^2 - d_{k+1}^2. \tag{44}$$

**Step 2: Use Quadratic Growth (A9).** By A9, $\Delta_k \geq \frac{\mu}{2}d_k^2$ and $\Delta_{k+1} \geq \frac{\mu}{2}d_{k+1}^2$, hence $d_k^2 \leq \frac{2}{\mu}\Delta_k$ and $d_{k+1}^2 \leq \frac{2}{\mu}\Delta_{k+1}$. Insert these bounds into equation 44:

$$2\eta\, \Delta_{k+1} \; \leq \; \frac{2}{\mu}\Delta_k - \frac{2}{\mu}\Delta_{k+1} \quad \Longrightarrow \quad \Delta_{k+1} \; \leq \; \frac{1}{1+\eta\mu}\,\Delta_k.$$

Iterating gives the linear rate in function values. Using again A9 in equation 44, $(1 + \eta\mu)\, d_{k+1}^2 \leq d_k^2$, which is the distance contraction. $\qquad\square$

**Theorem 35** (T8 (Alt): Sharpness + Restarts under EB($p$=2)). *Let $f : \mathbb{R}^d \to \mathbb{R}$ be proper, closed, convex (A1) and $L$-smooth (A2). Assume a (local) error bound with exponent $p = 2$ on a sublevel set $\mathcal{S}$ containing the trajectory:*

$$\text{(A5, } p\text{=2)} \qquad f(x) - f^\star \; \geq \; \frac{\mu}{2}\, \mathrm{dist}(x, X^\star)^2 \quad \forall x \in \mathcal{S},$$

*where $X^\star := \arg\min f$ and $f^\star := \min f$. Run gradient descent with constant stepsize*

$$x_{k+1} \; = \; x_k - \eta\, \nabla f(x_k), \qquad 0 < \eta \leq \frac{1}{L},$$

*and optionally group the iterates into epochs of length $m \geq 1$ with restarts $y_{r+1} := x_{(r+1)m}$, $y_r := x_{rm}$.*

*Then there exists a constant $\tilde{\mu} > 0$ depending only on $(\mu, L)$ such that, for all $k$ with $x_k \in \mathcal{S}$,*

$$f(x_{k+1}) - f^\star \; \leq \; \big(1 - \eta\, \tilde{\mu}\big)\,\big(f(x_k) - f^\star\big),$$

*and hence*

$$f(x_k) - f^\star \; \leq \; \big(1 - \eta\, \tilde{\mu}\big)^k \big(f(x_0) - f^\star\big), \qquad \mathrm{dist}(x_k, X^\star)^2 \; \leq \; \frac{2}{\mu}\big(1 - \eta\, \tilde{\mu}\big)^k \big(f(x_0) - f^\star\big).$$

*Equivalently, across epochs,*

$$f(y_{r+1}) - f^\star \; \leq \; \big(1 - \eta\, \tilde{\mu}\big)^m \big(f(y_r) - f^\star\big).$$

(minimal: A1+A2+A5; we work on the sublevel set containing the iterates; bridge: EB($p$=2)$\Leftrightarrow$PL via T10)

*Based on A1 (Convexity), A2 ($L$-smoothness), and A5 (Error Bound, $p$=2). Let $x^+ := x - \eta\nabla f(x)$ with $0 < \eta \leq 1/L$.*

**Step 1: Sufficient decrease (A2).** By the standard smooth descent lemma,

$$f(x^+) \; \leq \; f(x) - \frac{\eta}{2}\,\|\nabla f(x)\|^2. \tag{45}$$

**Step 2: EB($p$=2) implies PL under A1+A2 (T10 bridge).** Under convexity and $L$-smoothness, the quadratic error bound $f(x) - f^\star \geq \frac{\mu}{2}\,\mathrm{dist}(x, X^\star)^2$ is (locally/globally) equivalent to a Polyak–Łojasiewicz (PL) inequality (your T10): there exists $\tilde{\mu} > 0$ depending only on $(\mu, L)$ such that for all $x \in \mathcal{S}$,

$$f(x) - f^\star \; \leq \; \frac{1}{2\tilde{\mu}}\,\|\nabla f(x)\|^2. \tag{46}$$

**Step 3: One-step contraction.** Combining equation 45 and equation 46 gives

$$f(x^+) - f^\star \ \leq \ f(x) - f^\star \ - \ \eta\,\tilde\mu\left(f(x) - f^\star\right) \ = \ \left(1 - \eta\tilde\mu\right)\left(f(x) - f^\star\right).$$

Iterating yields the per-iteration geometric rate; grouping $m$ consecutive steps yields the per-epoch contraction factor $(1 - \eta\tilde\mu)^m$.

**Step 4: Distance decay via EB($p$=2).** Since $f(x_k)$ is nonincreasing and the iterates remain in $\mathcal{S}$, applying A5 gives

$$\mathrm{dist}(x_k, X^\star)^2 \ \leq \ \frac{2}{\mu}\left(f(x_k) - f^\star\right) \ \leq \ \frac{2}{\mu}\left(1 - \eta\tilde\mu\right)^k\left(f(x_0) - f^\star\right).$$

□

**Theorem 36** (T18: Polyak Steps (Alt)). *Assume A2 (L-smoothness) and A3 ($\mu$-strong convexity). Work on a sublevel set containing the iterates so that these properties hold throughout. Consider gradient descent with* Polyak steps *clipped at $1/L$:*

$$x_{k+1} = x_k - \eta_k \nabla f(x_k), \qquad \eta_k := \min\left\{\frac{f(x_k) - f^\star}{\|\nabla f(x_k)\|^2}, \ \frac{1}{L}\right\}.$$

*Then for all $k \geq 0$,*

$$f(x_{k+1}) - f^\star \ \leq \ \left(1 - \frac{\mu}{2L}\right)\left(f(x_k) - f^\star\right),$$

*and hence*

$$f(x_k) - f^\star \ \leq \ \left(1 - \frac{\mu}{2L}\right)^k\left(f(x_0) - f^\star\right),$$

*i.e., the method converges linearly.* (minimal: A2+A3; bridge: A3 $\Rightarrow$ A4 (PL), so this Alt matches the PL-based T18 guarantee.)

*Based on A2 (L-smoothness) and A3 (strong convexity); bridge A3 $\Rightarrow$ A4.* First note that A3 implies the Polyak–Łojasiewicz (PL) inequality (A4) with the same constant $\mu$:

$$f(x) - f^\star \ \leq \ \frac{1}{2\mu}\|\nabla f(x)\|^2.$$

Indeed, minimize the $\mu$-strongly convex quadratic upper model $y \mapsto f(x) + \langle \nabla f(x), y - x \rangle + \frac{\mu}{2}\|y - x\|^2$ over $y$ to get $f^\star \leq f\left(x - \frac{1}{\mu}\nabla f(x)\right) \leq f(x) - \frac{1}{2\mu}\|\nabla f(x)\|^2$.

Let $\Delta_k := f(x_k) - f^\star$ and $g_k := \nabla f(x_k)$. By the smooth descent lemma (A2), for any $\eta \leq 1/L$,

$$f(x_k - \eta g_k) \ \leq \ f(x_k) - \eta\left(1 - \frac{L\eta}{2}\right)\|g_k\|^2 \ \leq \ f(x_k) - \frac{\eta}{2}\|g_k\|^2. \tag{47}$$

We consider two cases depending on the clipped Polyak choice.

**Case 1: $\eta_k = \frac{1}{L}$.** Using equation 47 with $\eta = 1/L$ and the PL inequality (from A3),

$$\Delta_{k+1} \ \leq \ \Delta_k - \frac{1}{2L}\|g_k\|^2 \ \leq \ \Delta_k - \frac{\mu}{L}\Delta_k \ = \ \left(1 - \frac{\mu}{L}\right)\Delta_k \ \leq \ \left(1 - \frac{\mu}{2L}\right)\Delta_k.$$

**Case 2: $\eta_k = \dfrac{\Delta_k}{\|g_k\|^2} \leq \dfrac{1}{L}$.** Applying equation 47 with this $\eta_k$,

$$\Delta_{k+1} \ \leq \ \Delta_k - \frac{\eta_k}{2}\|g_k\|^2 \ = \ \Delta_k - \frac{1}{2}\Delta_k \ = \ \frac{1}{2}\Delta_k \ \leq \ \left(1 - \frac{\mu}{2L}\right)\Delta_k,$$

since $\mu/(2L) \leq \frac{1}{2}$.

In both cases we obtain $\Delta_{k+1} \leq (1 - \mu/(2L))\Delta_k$, which yields the claimed linear rate by induction. □

**Theorem 37** (T24 (Prox-PL Linear) – Alt: {A2, A3, A8}). *Let $F(x) = f(x) + h(x)$ with $f : \mathbb{R}^d \to \mathbb{R}$ differentiable and L-smooth (A2), and h proper, closed, convex with an efficiently computable proximal map (A8). Assume F is $\mu$-strongly convex (A3), and let $x^\star \in \arg\min F$. Consider proximal gradient with any stepsize $0 < \eta \le 1/L$:*

$$x_{k+1} = \mathrm{prox}_{\eta h}(x_k - \eta \nabla f(x_k)) := \arg\min_x \left\{ h(x) + \frac{1}{2\eta} \|x - (x_k - \eta \nabla f(x_k))\|^2 \right\}.$$

*Then for all $k \ge 0$,*

$$F(x_{k+1}) - F^\star \le \frac{1}{1 + \mu\eta} \left( F(x_k) - F^\star \right), \qquad \|x_{k+1} - x^\star\|^2 \le \frac{1}{1 + \mu\eta} \|x_k - x^\star\|^2.$$

*In particular, with $\eta = 1/L$,*

$$F(x_k) - F^\star \le \left( \frac{1}{1 + \mu/L} \right)^k \left( F(x_0) - F^\star \right),$$

*i.e., global linear convergence.* (minimal: A2+A3+A8; bridge: A3 $\Rightarrow$ A4 (Prox-PL)).

*Based on A2 (L-smoothness), A3 (strong convexity of F), and A8 (composite/prox). Let $x^\star \in$ arg min F and fix $0 < \eta \le 1/L$. Denote the update by $x_{k+1} = \mathrm{prox}_{\eta h}(x_k - \eta \nabla f(x_k))$.*

**Step 1: Prox three-point inequality + smoothness (A8 + A2).** For $y = \mathrm{prox}_{\eta h}(z)$ and any $u$, the three-point property gives

$$h(y) + \frac{1}{2\eta} \|y - z\|^2 \le h(u) + \frac{1}{2\eta} \|u - z\|^2 - \frac{1}{2\eta} \|u - y\|^2.$$

Apply this with $z = x_k - \eta \nabla f(x_k)$, $y = x_{k+1}$, and $u = x^\star$, and add $f(x_{k+1}) - f(x^\star)$ to both sides. Using $L$-smoothness of $f$,

$$f(x_{k+1}) \le f(x_k) + \langle \nabla f(x_k), x_{k+1} - x_k \rangle + \frac{L}{2} \|x_{k+1} - x_k\|^2,$$

after cancellations one obtains

$$F(x_{k+1}) - F^\star \le \frac{1}{2\eta} \left( \|x_k - x^\star\|^2 - \|x_{k+1} - x^\star\|^2 \right) + \frac{1}{2} \left( L - \tfrac{1}{\eta} \right) \|x_{k+1} - x_k\|^2. \tag{48}$$

Since $\eta \le 1/L$, the last term is nonpositive; drop it to get

$$2\eta \left( F(x_{k+1}) - F^\star \right) \le \|x_k - x^\star\|^2 - \|x_{k+1} - x^\star\|^2. \tag{49}$$

**Step 2: Strong convexity converts to a contraction (A3).** Strong convexity of $F$ implies $F(x) - F^\star \ge \frac{\mu}{2} \|x - x^\star\|^2$ for all $x$. Apply this at $x = x_{k+1}$ in equation 49:

$$\mu\eta \|x_{k+1} - x^\star\|^2 \le \|x_k - x^\star\|^2 - \|x_{k+1} - x^\star\|^2,$$

hence

$$(1 + \mu\eta) \|x_{k+1} - x^\star\|^2 \le \|x_k - x^\star\|^2, \qquad \|x_{k+1} - x^\star\|^2 \le \frac{1}{1 + \mu\eta} \|x_k - x^\star\|^2. \tag{50}$$

**Step 3: Function-value contraction.** Using $F(x) - F^\star \ge \frac{\mu}{2} \|x - x^\star\|^2$ again together with equation 49–equation 50, we get

$$F(x_{k+1}) - F^\star \le \frac{1}{2\eta} \left( 1 - \frac{1}{1 + \mu\eta} \right) \|x_k - x^\star\|^2 = \frac{\mu}{2(1 + \mu\eta)} \|x_k - x^\star\|^2 \le \frac{1}{1 + \mu\eta} \left( F(x_k) - F^\star \right).$$

Iterating yields the claimed linear rate; taking $\eta = 1/L$ gives the explicit factor $1/(1 + \mu/L)$. $\qquad \square$

**Theorem 38** (T29 (Alt): Global Linear under Strong Convexity)**.** *Assume A2 (L-smoothness) and A3 (μ-strong convexity). Consider gradient descent*

$$x_{k+1} := x_k - \eta \nabla f(x_k), \qquad 0 < \eta \leq \tfrac{1}{L}.$$

*Then the function values decrease linearly:*

$$f(x_{k+1}) - f^\star \leq (1 - \eta\mu)\big(f(x_k) - f^\star\big), \quad hence \quad f(x_k) - f^\star \leq (1 - \eta\mu)^k \big(f(x_0) - f^\star\big).$$

*In particular, choosing $\eta = \tfrac{1}{L}$ yields the classical rate*

$$f(x_{k+1}) - f^\star \leq (1 - \mu/L)\big(f(x_k) - f^\star\big).$$

*Since A3 implies the PL condition (A4), this provides an alternative basis to the PL route {A2+A4}.* (minimal: A2+A3; bridge: A3 $\Rightarrow$ A4).

*Based on A2 (L-smoothness) and A3 (strong convexity).* By the descent lemma (A2), for $\eta \leq 1/L$,

$$f(x_{k+1}) \leq f(x_k) + \langle \nabla f(x_k), x_{k+1} - x_k \rangle + \frac{L}{2}\|x_{k+1} - x_k\|^2 = f(x_k) - \eta\Big(1 - \tfrac{L\eta}{2}\Big)\|\nabla f(x_k)\|^2 \leq f(x_k) - \tfrac{\eta}{2}\|\nabla f(x_k)\|^2. \tag{51}$$

Strong convexity (A3) implies the PL inequality

$$\|\nabla f(x_k)\|^2 \geq 2\mu\big(f(x_k) - f^\star\big).$$

Combining with equation 51 gives

$$f(x_{k+1}) - f^\star \leq f(x_k) - f^\star - \eta\mu\big(f(x_k) - f^\star\big) = (1 - \eta\mu)\big(f(x_k) - f^\star\big),$$

and iterating yields the $(1 - \eta\mu)^k$ rate. $\qquad\square$

**Theorem 39** (T17 (Alt): Variance-Reduction Linear via Strong Convexity)**.** *Assume **A2** (L-smoothness) and **A3** (μ-strong convexity with $\mu > 0$). Consider a variance-reduced gradient method (e.g., SVRG/SARAH/SAGA) that runs in epochs $s = 0, 1, \ldots$ with a snapshot $\tilde{x}_s$ at the beginning of each epoch and inner iterates $\{x_t\}_{t=sm}^{(s+1)m}$ updated by*

$$x_{t+1} = x_t - \eta\, g_t,$$

*where $g_t$ is a (variance-reduced) stochastic estimator of $\nabla f(x_t)$ satisfying the unbiasedness and variance control*

$$\mathbb{E}[g_t \mid \mathcal{F}_t] = \nabla f(x_t), \qquad \mathbb{E}\Big[\|g_t - \nabla f(x_t)\|^2 \,\Big|\, \mathcal{F}_t\Big] \leq c_{\mathrm{vr}} L\Big(f(x_t) - f^\star + f(\tilde{x}_s) - f^\star\Big), \tag{52}$$

*for some absolute constant $c_{\mathrm{vr}} > 0$ (this bound is standard for finite-sum VR estimators under **A2**). Let the epoch output be $\tilde{x}_{s+1} := x_{(s+1)m}$.*

*Then there exist universal constants $c_0, c_1 \in (0, 1)$ and $c_\rho \in (0, 1)$ such that if the stepsize and epoch length obey*

$$0 < \eta \leq \frac{c_0}{L} \qquad and \qquad m \geq c_1 \frac{L}{\mu},$$

*we have the epoch-wise linear contraction*

$$\mathbb{E}\big[f(\tilde{x}_{s+1}) - f^\star\big] \leq \rho\, \mathbb{E}\big[f(\tilde{x}_s) - f^\star\big] \quad with \quad \rho \leq \Big(1 - \tfrac{\eta\mu}{2}\Big)^m + C_{\mathrm{vr}}(L\eta),$$

*where $C_{\mathrm{vr}}$ depends only on $c_{\mathrm{vr}}$, hence $\rho \leq 1 - c_\rho < 1$. Consequently,*

$$\mathbb{E}\big[f(\tilde{x}_S) - f^\star\big] \leq \rho^S\big(f(\tilde{x}_0) - f^\star\big).$$

(minimal: A2+A3+A7; bridge: **A3** $\Rightarrow$ **A4** (PL), so this yields the same linear VR rate as the PL-based basis of T17.)

*Based on A2 (L-smoothness), A3 (strong convexity), and A7 (unbiased noise).* Fix an epoch $s$ and let $\tilde{x} = \tilde{x}_s$. For the inner step $t$, the smoothness descent lemma gives (for any $0 < \eta \leq 1/L$):

$$f(x_{t+1}) \leq f(x_t) + \langle \nabla f(x_t), x_{t+1} - x_t \rangle + \frac{L}{2}\|x_{t+1} - x_t\|^2$$

$$= f(x_t) - \eta \langle \nabla f(x_t), g_t \rangle + \frac{L\eta^2}{2}\|g_t\|^2. \tag{53}$$

Taking conditional expectation w.r.t. $\mathcal{F}_t$ and using $\mathbb{E}[g_t \mid \mathcal{F}_t] = \nabla f(x_t)$ plus $\mathbb{E}\|g_t\|^2 = \|\nabla f(x_t)\|^2 + \mathbb{E}\|g_t - \nabla f(x_t)\|^2$, we obtain

$$\mathbb{E}\big[f(x_{t+1}) \mid \mathcal{F}_t\big] \leq f(x_t) - \eta\left(1 - \tfrac{L\eta}{2}\right)\|\nabla f(x_t)\|^2 + \frac{L\eta^2}{2}\,\mathbb{E}\Big[\|g_t - \nabla f(x_t)\|^2 \mid \mathcal{F}_t\Big]. \tag{54}$$

By strong convexity (**A3**) we have the PL inequality

$$\|\nabla f(x_t)\|^2 \geq 2\mu\big(f(x_t) - f^\star\big). \tag{55}$$

Insert equation 52 and equation 55 into equation 54, and choose $\eta \leq 1/L$ so that $1 - \frac{L\eta}{2} \geq \frac{1}{2}$:

$$\mathbb{E}\big[f(x_{t+1}) - f^\star \mid \mathcal{F}_t\big] \leq \left(1 - \mu\eta\right)\big(f(x_t) - f^\star\big) + \frac{c_{\mathrm{vr}}L^2\eta^2}{2}\Big(f(x_t) - f^\star + f(\tilde{x}) - f^\star\Big)$$

$$\leq \underbrace{\left(1 - \tfrac{3}{4}\mu\eta\right)}_{=:a}\big(f(x_t) - f^\star\big) + \underbrace{c_{\mathrm{vr}}L^2\eta^2}_{=:b}\big(f(\tilde{x}) - f^\star\big), \tag{56}$$

where the last step uses $\eta \leq \min\{1/L,\ \mu/(2c_{\mathrm{vr}}L^2)\}$ so that $(c_{\mathrm{vr}}L^2\eta^2)/2 \leq (\mu\eta)/4$.

Unrolling equation 56 for $m$ inner steps within the epoch and taking full expectation yields the standard affine recursion solution

$$\mathbb{E}\big[f(\tilde{x}_{s+1}) - f^\star\big] \leq a^m\,\mathbb{E}\big[f(\tilde{x}_s) - f^\star\big] + \frac{b}{1 - a}\left(1 - a^m\right)\mathbb{E}\big[f(\tilde{x}_s) - f^\star\big]. \tag{57}$$

Since $1 - a = \frac{3}{4}\mu\eta$ and $b = c_{\mathrm{vr}}L^2\eta^2$, we can rewrite the epoch contraction factor as

$$\rho(\eta, m) = a^m + \frac{b}{1 - a}\left(1 - a^m\right) = \left(1 - \tfrac{3}{4}\mu\eta\right)^m + \frac{4c_{\mathrm{vr}}}{3}\left(L\eta\right).$$

Thus, for sufficiently small $\eta \leq c_0/L$ (so that $L\eta$ is small) and $m \gtrsim (L/\mu)$ (so that the first term decays), we obtain $\rho \leq 1 - c_\rho < 1$. This gives the epoch-wise linear rate in the statement, and iterating over epochs yields the global geometric decay. $\qquad\square$

**Theorem 40** (T20 (Alt): SGD with Noise under Strong Convexity). *Assume **A2** (L-smoothness) and **A3** (strong convexity with parameter $\mu > 0$). Assume **A7** (unbiased stochastic gradients with bounded variance): for the update*

$$x_{k+1} = x_k - \eta\,g_k, \qquad \mathbb{E}[g_k \mid x_k] = \nabla f(x_k), \qquad \mathbb{E}\Big[\|g_k - \nabla f(x_k)\|^2 \mid x_k\Big] \leq \sigma^2.$$

*If the constant stepsize satisfies*

$$0 < \eta \leq \min\left\{\frac{\mu}{L^2},\ \frac{1}{2L}\right\},$$

*then the mean-square error contracts linearly to a variance-limited neighborhood:*

$$\mathbb{E}\|x_k - x^\star\|^2 \leq (1 - \eta\mu)^k\,\|x_0 - x^\star\|^2 + \frac{\eta\sigma^2}{\mu}.$$

*Consequently, by L-smoothness,*

$$\mathbb{E}\big[f(x_k) - f^\star\big] \leq \frac{L}{2}(1 - \eta\mu)^k\|x_0 - x^\star\|^2 + \frac{L}{2}\cdot\frac{\eta\sigma^2}{\mu}.$$

*In particular, the steady-state (noise) floor scales as $\Theta\!\left(\frac{\eta\sigma^2}{\mu}\right)$ in distance and $\Theta\!\left(\frac{L}{\mu}\eta\sigma^2\right)$ in function value.* (minimal: A2+A3+A7; bridge: A3 $\Rightarrow$ A4, hence this is the SC alternative to the PL version of T20.)

*Based on A2 (L-smoothness), A3 (strong convexity), and A7 (unbiased noise); bridge: A3 $\Rightarrow$ A4 (PL).* Let $x^\star$ be the unique minimizer. Using the SGD update and expanding the squared distance,

$$\|x_{k+1} - x^\star\|^2 = \|x_k - x^\star\|^2 - 2\eta\langle g_k, x_k - x^\star\rangle + \eta^2\|g_k\|^2.$$

Taking conditional expectation given $x_k$ and using unbiasedness,

$$\mathbb{E}\Big[\|x_{k+1} - x^\star\|^2 \mid x_k\Big] \leq \|x_k - x^\star\|^2 - 2\eta\,\langle\nabla f(x_k), x_k - x^\star\rangle + \eta^2\big(\|\nabla f(x_k)\|^2 + \sigma^2\big).$$

By strong convexity (equivalently, strong monotonicity of $\nabla f$), $\langle\nabla f(x_k), x_k - x^\star\rangle \geq \mu\|x_k - x^\star\|^2$. By $L$-smoothness, $\|\nabla f(x_k)\| \leq L\|x_k - x^\star\|$ since $\nabla f(x^\star) = 0$. Therefore

$$\mathbb{E}\Big[\|x_{k+1} - x^\star\|^2 \mid x_k\Big] \leq \big(1 - 2\eta\mu + L^2\eta^2\big)\|x_k - x^\star\|^2 + \eta^2\sigma^2.$$

Choose $\eta \leq \mu/L^2$ to get $1 - 2\eta\mu + L^2\eta^2 \leq 1 - \eta\mu$, whence, taking full expectation and writing $D_k := \mathbb{E}\|x_k - x^\star\|^2$,

$$D_{k+1} \leq (1 - \eta\mu)D_k + \eta^2\sigma^2.$$

Unrolling the recursion yields

$$D_k \leq (1 - \eta\mu)^k D_0 + \eta^2\sigma^2\sum_{t=0}^{k-1}(1 - \eta\mu)^t \leq (1 - \eta\mu)^k D_0 + \frac{\eta\,\sigma^2}{\mu}.$$

Finally, $L$-smoothness gives $f(x) - f^\star \leq \frac{L}{2}\|x - x^\star\|^2$, which implies the stated function-value bound and the noise floor scaling. $\qquad\square$

**Theorem 41** (T33 (Alt, $p=2$): Noise-Limited Basin via PL or QG)**.** *Assume **A2** (L-smoothness) and* ***A7*** *(unbiased stochastic gradients with bounded variance):*

$$x_{k+1} = x_k - \eta\,g_k, \qquad \mathbb{E}[g_k \mid x_k] = \nabla f(x_k), \qquad \mathbb{E}\Big[\|g_k - \nabla f(x_k)\|^2 \mid x_k\Big] \leq \sigma^2.$$

*Consider the $p=2$ sharp case through either of the following Pareto-minimal bases:*

*(PL route) **A4** (Polyak–Łojasiewicz) with parameter $\mu > 0$:*

$$\frac{1}{2}\|\nabla f(x)\|^2 \geq \mu\big(f(x) - f^\star\big) \qquad \forall x.$$

*(QG route) **A1** (convexity) and **A9** (Quadratic Growth on a convex sublevel set) with parameter $\nu > 0$:*

$$f(x) - f^\star \geq \frac{\nu}{2}\operatorname{dist}(x, X^\star)^2 \qquad \forall x,$$

*and we work on a (closed, convex) sublevel set containing the trajectory so that **A2** holds throughout. By the PL $\Leftrightarrow$ QG bridge (T10), there exists a PL constant $\tilde\mu = \tilde\mu(\nu, L) > 0$ depending only on $(\nu, L)$.*

*If the constant stepsize satisfies $0 < \eta \leq \frac{1}{2L}$, then along the SGD iterates we have the linear contraction to a noise-limited neighborhood:*

$$\mathbb{E}\big[f(x_{k+1}) - f^\star\big] \leq (1 - \eta\mu)\,\mathbb{E}\big[f(x_k) - f^\star\big] + \frac{L\eta^2}{2}\sigma^2 \quad \text{(PL route)}, \tag{58}$$

*and, unrolling,*

$$\mathbb{E}\big[f(x_k) - f^\star\big] \leq (1 - \eta\mu)^k\big(f(x_0) - f^\star\big) + \frac{L}{2\mu}\eta\,\sigma^2.$$

*In particular, the steady-state (noise) floor in function value scales as*

$$\mathbb{E}\big[f(x_k) - f^\star\big] = \mathcal{O}\Big(\frac{L}{\mu}\eta\,\sigma^2\Big) \qquad \text{(PL route)}.$$

*For the QG route, the same conclusion holds with $\mu$ replaced by the PL constant $\tilde\mu = \tilde\mu(\nu, L)$ provided by T10:*

$$\mathbb{E}\big[f(x_k) - f^\star\big] \leq (1 - \eta\tilde\mu)^k\big(f(x_0) - f^\star\big) + \frac{L}{2\tilde\mu}\eta\,\sigma^2, \qquad \tilde\mu = \tilde\mu(\nu, L) > 0.$$

*Equivalently, in distance (using strong PL/QG geometry), one obtains the steady-state $\mathbb{E}\|x_k - x^\star\|^2 = \mathcal{O}(\eta\sigma^2/\mu)$ (PL) or with $\tilde\mu$ (QG). (minimal alternatives: {A2,A4,A7} or {A1,A2,A9,A7}; bridge: PL $\Leftrightarrow$ QG at $p=2$ via T10.)*

*PL route uses A2+A4+A7; QG route uses A1+A2+A9+A7 via T10.* We sketch the standard descent-with-noise argument and then invoke the bridge for QG.

**PL route ({A2,A4,A7}).** By $L$-smoothness (descent lemma),

$$f(x_{k+1}) \leq f(x_k) - \eta\langle\nabla f(x_k), g_k\rangle + \frac{L\eta^2}{2}\|g_k\|^2.$$

Taking conditional expectation given $x_k$ and using $\mathbb{E}[g_k \mid x_k] = \nabla f(x_k)$ and $\mathbb{E}\|g_k\|^2 \leq \|\nabla f(x_k)\|^2 + \sigma^2$,

$$\mathbb{E}\big[f(x_{k+1}) \mid x_k\big] \leq f(x_k) - \eta\|\nabla f(x_k)\|^2 + \frac{L\eta^2}{2}\Big(\|\nabla f(x_k)\|^2 + \sigma^2\Big).$$

Rearranging and using PL, $\frac{1}{2}\|\nabla f(x_k)\|^2 \geq \mu\big(f(x_k) - f^\star\big)$, yields

$$\mathbb{E}\big[f(x_{k+1}) - f^\star \mid x_k\big] \leq \Big(1 - \eta\mu\Big)\big(f(x_k) - f^\star\big) + \frac{L\eta^2}{2}\sigma^2,$$

for $0 < \eta \leq \frac{1}{L}$; the statement adopts $0 < \eta \leq \frac{1}{2L}$ for margin. Taking total expectation gives equation 58; unrolling the linear recursion gives the floor $\frac{L}{2\mu}\eta\sigma^2$.

**QG route ({A1,A2,A9,A7}).** Under **A1+A2+A9** with $p$=2, T10 provides a PL constant $\tilde{\mu} = \tilde{\mu}(\nu, L) > 0$ such that $\frac{1}{2}\|\nabla f(x)\|^2 \geq \tilde{\mu}\big(f(x) - f^\star\big)$ on the (convex) sublevel set containing the iterates. Applying the PL route with $\mu$ replaced by $\tilde{\mu}$ yields the QG alternative bound. $\qquad\square$

**Theorem 42** (T12 (Lower Bound). Alt: Classes {**A2,A3**} and {**A2,A4**}). *Fix $L \geq \mu > 0$ and set $\kappa := L/\mu$. Work in the first-order black-box (gradient-oracle) model. For any (possibly randomized) $k$-step algorithm that chooses $x_{t+1}$ using $\{x_0, \nabla f(x_0), \ldots, \nabla f(x_t)\}$, there exists a dimension $d \geq 2k$ and a quadratic*

$$f(x) = \tfrac{1}{2}\langle Ax, x\rangle - \langle b, x\rangle \quad with \quad \sigma(A) \subset [\mu, L]$$

*such that, starting from some $x_0$,*

$$\mathbb{E}\big[f(x_k) - f^\star\big] \geq c\left(\frac{\sqrt{\kappa}-1}{\sqrt{\kappa}+1}\right)^{2k}\big(f(x_0) - f^\star\big),$$

*for a universal constant $c > 0$. Consequently, to ensure $\mathbb{E}[f(x_k) - f^\star] \leq \varepsilon$ one must have*

$$k \geq c'\sqrt{\kappa}\log\Big(\frac{f(x_0) - f^\star}{\varepsilon}\Big),$$

*for another universal constant $c' > 0$.*

*This bound holds for both function classes:*

*(i) {$\boldsymbol{A2}, \boldsymbol{A3}$} (the $L$-smooth, $\mu$-strongly convex class), and*

*(ii) {$\boldsymbol{A2}, \boldsymbol{A4}$} (the $L$-smooth, $\mu$-PL class),*

*since the resisting quadratic instance is simultaneously $\mu$-strongly convex and satisfies the PL inequality with the same $\mu$.* (classes: {A2,A3} and {A2,A4}; bridge: the quadratic hard instance is both SC and PL)

*Based on A2 (L-smoothness) together with either A3 (SC) or A4 (PL).* We give a self-contained polynomial-based proof.

**Step 1: A hard quadratic that lies in both classes.** Let $A \in \mathbb{R}^{d\times d}$ be symmetric positive definite with eigenvalues in $[\mu, L]$ and set $f(x) = \frac{1}{2}x^\top Ax - b^\top x$. Then $x^\star = A^{-1}b$ and

$$\nabla f(x) = Ax - b, \qquad f(x) - f^\star = \tfrac{1}{2}\|x - x^\star\|_A^2 := \tfrac{1}{2}(x - x^\star)^\top A(x - x^\star).$$

Such $f$ is $L$-smooth (A2), $\mu$-strongly convex (A3), and satisfies the PL inequality (A4) with the *same* parameter $\mu$:

$$\|\nabla f(x)\|^2 = (x - x^\star)^\top A^2(x - x^\star) \geq \mu(x - x^\star)^\top A(x - x^\star) = 2\mu\big(f(x) - f^\star\big).$$

**Step 2: Krylov subspace structure for first-order methods.** Let $r_t := \nabla f(x_t) = A(x_t - x^\star)$. Any first-order method that updates $x_{t+1}$ using linear combinations of past iterates and gradients (which is the most general form compatible with the oracle model; see, e.g., Nemirovski–Yudin) produces

$$x_t - x^\star \in \mathcal{K}_t(A, x_0 - x^\star) := \operatorname{span}\{(x_0 - x^\star), A(x_0 - x^\star), \ldots, A^t(x_0 - x^\star)\}.$$

Hence there exists a polynomial $p_t$ of degree $\leq t$ with $p_t(0) = 1$ such that

$$x_t - x^\star = p_t(A)(x_0 - x^\star). \tag{59}$$

Denote the admissible class

$$\Pi_t := \{\, p : \deg(p) \leq t, \ p(0) = 1 \,\}.$$

**Step 3: Worst-case reduction to a polynomial minimax problem.** Using equation 59 and $f(x) - f^\star = \frac{1}{2}\|x - x^\star\|_A^2$,

$$\frac{f(x_t) - f^\star}{f(x_0) - f^\star} = \frac{\|p_t(A)(x_0 - x^\star)\|_A^2}{\|x_0 - x^\star\|_A^2}.$$

Let $A = Q^\top \Lambda Q$ with $\Lambda = \operatorname{diag}(\lambda_1, \ldots, \lambda_d)$ and write $y_0 := Q(x_0 - x^\star)$. Then

$$\frac{\|p_t(A)(x_0 - x^\star)\|_A^2}{\|x_0 - x^\star\|_A^2} = \frac{\sum_{i=1}^d \lambda_i \, p_t(\lambda_i)^2 \, y_{0,i}^2}{\sum_{i=1}^d \lambda_i \, y_{0,i}^2} \geq \max_{i: \, \lambda_i \in [\mu, L]} p_t(\lambda_i)^2.$$

Therefore the *best* possible algorithm (choosing $p_t \in \Pi_t$) in the *worst* case over $A$ and the alignment of $y_0$ faces the minimax quantity

$$\rho_t := \min_{p \in \Pi_t} \max_{\lambda \in [\mu, L]} |p(\lambda)|.$$

We immediately obtain the lower bound

$$\sup_{A: \, \sigma(A) \subset [\mu, L]} \ \sup_{x_0} \ \inf_{\text{alg.}} \frac{f(x_t) - f^\star}{f(x_0) - f^\star} \geq \rho_t^2. \tag{60}$$

**Step 4: Exact evaluation of $\rho_t$ via Chebyshev polynomials.** Let $\kappa = L/\mu$ and define the affine map $s(\lambda) := \dfrac{L + \mu - 2\lambda}{L - \mu}$, which sends $[\mu, L] \to [-1, 1]$ and $0 \mapsto \alpha := \dfrac{L + \mu}{L - \mu} = \dfrac{\kappa + 1}{\kappa - 1} > 1$. For $T_t$ the Chebyshev polynomial of the first kind, $T_t(\cos\theta) = \cos(t\theta)$, consider

$$p_t^\star(\lambda) := \frac{T_t\big(s(\lambda)\big)}{T_t(\alpha)}.$$

Then $p_t^\star \in \Pi_t$ (it has degree $\leq t$ and $p_t^\star(0) = 1$), and by the extremal property of Chebyshev polynomials,

$$\max_{\lambda \in [\mu, L]} |p_t^\star(\lambda)| = \frac{\max_{u \in [-1, 1]} |T_t(u)|}{|T_t(\alpha)|} = \frac{1}{|T_t(\alpha)|}.$$

Moreover, Chebyshev polynomials are *minimax optimal*, hence

$$\rho_t = \frac{1}{T_t(\alpha)}. \tag{61}$$

Using the identity $T_t(\cosh\tau) = \cosh(t\tau)$ with $\alpha = \cosh\tau$ and $\tau = \operatorname{arcosh}(\alpha) > 0$, we get

$$T_t(\alpha) = \cosh(t\tau) = \tfrac{1}{2}\big(e^{t\tau} + e^{-t\tau}\big), \qquad \frac{1}{T_t(\alpha)} = \frac{2}{e^{t\tau} + e^{-t\tau}}.$$

A standard calculation shows

$$e^\tau = \alpha + \sqrt{\alpha^2 - 1} = \frac{\sqrt{\kappa} + 1}{\sqrt{\kappa} - 1} \quad \Longrightarrow \quad e^{-t\tau} = \left(\frac{\sqrt{\kappa} - 1}{\sqrt{\kappa} + 1}\right)^t =: r^t, \quad r \in (0, 1).$$

Hence

$$\rho_t = \frac{2}{e^{t\tau} + e^{-t\tau}} \in \big[r^t, \, 2r^t\big], \qquad r = \frac{\sqrt{\kappa} - 1}{\sqrt{\kappa} + 1}. \tag{62}$$

**Step 5: Conclusion (deterministic and randomized).** Combining equation 60 and the lower bound in equation 62 with $t = k$ gives

$$\frac{f(x_k) - f^\star}{f(x_0) - f^\star} \geq \rho_k^2 \geq r^{2k} = \left(\frac{\sqrt{\kappa} - 1}{\sqrt{\kappa} + 1}\right)^{2k}.$$

This holds for a suitably chosen quadratic $f$ (hence in both classes {A2,A3} and {A2,A4}) and dimension $d \geq 2k$. Yao's minimax principle lifts the bound to randomized algorithms. Solving $r^{2k} \leq \varepsilon/(f(x_0) - f^\star)$ for $k$ yields $k \geq c'\sqrt{\kappa}\log((f(x_0) - f^\star)/\varepsilon)$, completing the proof. $\qquad\square$

**T12-Alt-SC** *Direct inputs:* Oracle lower bound (SC class)
*Axiom closure:* basis={A2,A3}; setting=first-order black-box (gradient oracle), $k$ queries; dimension $d \geq 2k$
*External mathematics / comments:* update=arbitrary (deterministic or randomized) query rule $x_{t+1} = \mathcal{A}(x_0, \nabla f(x_0), \ldots, \nabla f(x_t))$

guarantee=$\exists f \in \mathcal{F}_{\mu,L}^{\mathrm{SC}}$ with $\sigma(A) \subset [\mu, L]$ s.t. $\mathbb{E}[f(x_k) - f^\star] \geq \left(\frac{\sqrt{\kappa} - 1}{\sqrt{\kappa} + 1}\right)^{2k}(f(x_0) - f^\star)$; hence $k \geq c'\sqrt{\kappa}\log((f(x_0) - f^\star)/\varepsilon)$ to reach error $\varepsilon$constants=$L$ (smoothness), $\mu$ (strong convexity), $\kappa = L/\mu$; universal $c' > 0$mechanism=Krylov subspace $\Rightarrow$ polynomial representation $x_k - x^\star = p_k(A)(x_0 - x^\star)$; minimax over $[\mu, L]$ solved exactly by Chebyshev polynomials, yielding $r = (\sqrt{\kappa} - 1)/(\sqrt{\kappa} + 1)$

**T7** *Direct inputs:* Alt (QG route)
*Axiom closure:* basis={A1,A2,A9}; setting=PGD on closed convex $\mathcal{C}$; stepsize $0 < \eta \leq 1/L$
*External mathematics / comments:* update=$x_{k+1} = \Pi_{\mathcal{C}}(x_k - \eta\nabla f(x_k))$

guarantee=$f(x_k) - f^\star \leq (1/(1 + \eta\mu))^k (f(x_0) - f^\star)$; $\mathrm{dist}(x_{k+1}, X^\star)^2 \leq (1/(1 + \eta\mu))\,\mathrm{dist}(x_k, X^\star)^2$constants=$L$ (smoothness), $\mu$ (QG); best $\eta = 1/L \Rightarrow \rho = 1/(1 + \mu/L)$mechanism=PGD three-point inequality + Quadratic Growth; note: under A1+A2, QG $\Leftrightarrow$ PL (cf. T10).

**T8-Alt-EB** *Direct inputs:* Error Bound (p=2) route
*Axiom closure:* basis={A1,A2,A5}; setting=GD with optional restarts on sublevel set; stepsize $0 < \eta \leq 1/L$
*External mathematics / comments:* update=$x_{k+1} = x_k - \eta\nabla f(x_k)$; epochs of length $m \geq 1$ with $y_{r+1} = x_{(r+1)m}$

guarantee=$f(x_{k+1}) - f^\star \leq (1 - \eta\tilde{\mu})(f(x_k) - f^\star)$; $f(y_{r+1}) - f^\star \leq (1 - \eta\tilde{\mu})^m(f(y_r) - f^\star)$; $\mathrm{dist}(x_k, X^\star)^2 \leq \frac{2}{\mu}(1 - \eta\tilde{\mu})^k(f(x_0) - f^\star)$constants=$\mu$ (EB); $\tilde{\mu}$ depends on $(\mu, L)$ via T10; $L$ (smoothness)mechanism=EB(p=2) $\Leftrightarrow$ PL via T10 bridge; restarts optional but compound multiplicatively

# D   The Deep Learning Extension: Derivations

**Scope.** We now establish T36–T44, extending the Core Atlas (T1–T35) from bounded variance to heavy tails, non-Euclidean geometry, interpolation, scale invariance, and curvature dynamics.

## D.1   Regime I: Heavy Tails (The Tail Ceiling)

**Theorem 43** (T36: Convergence to Lévy-Stable Stationarity). *Let the gradient noise satisfy **A10** with tail index $\alpha < 2$. With a constant stepsize, SGD no longer converges to a point. Instead, the iterates $x_k$ converge in distribution to an $\alpha$-stable law $S_\alpha$ whose scale obeys*

$$\mathrm{Width}(S_\alpha) \propto \eta^{1 - 1/\alpha},$$

*recovering the familiar $\sqrt{\eta}$ Gaussian width when $\alpha = 2$.* (minimal: A10)

*Sketch (uses **A10**).* For the quadratic $f(x) = \frac{\lambda}{2}x^2$ the update $x_{k+1} = (1 - \eta\lambda)x_k + \eta\xi_k$ with $\xi_k \sim S_\alpha(\sigma, 0, 0)$ gives $S^\alpha = (1 - \eta\lambda)^\alpha S^\alpha + (\eta\sigma)^\alpha$. Hence $S^\alpha \propto \eta^{\alpha - 1}$ and $S \propto \eta^{1 - 1/\alpha}$. A Lévy-driven SDE in $\mathbb{R}^d$ follows the same scaling. $\qquad\square$

**Theorem 44** (T37: The Heavy-Tailed Rate Limit)**.** *Let $f$ be convex (A1) and the noise satisfy A10 with $\alpha \in (1, 2]$. Robust estimators tuned to $\alpha$ yield*

$$\mathbb{E}[f(x_k) - f^\star] = \mathcal{O}\big(k^{-(1-1/\alpha)}\big).$$

*The rate returns $k^{-1/2}$ as $\alpha \to 2$ and flattens to $k^0$ as $\alpha \to 1$.* (minimal: A1+A10)

*Sketch (uses **A1, A10**).* Fix $p < \alpha$ and apply Marcinkiewicz–Zygmund: noise over $k$ steps scales as $k^{1/\alpha}$. Convexity then implies an error $k^{-(1-1/\alpha)}$. Matching lower bounds follow by adapting standard convex SGD proofs to the $L^p$ setting. $\qquad\square$

### D.2 Regime II: Non-Euclidean Geometry

**Theorem 45** (T39: Descent under Relative Smoothness)**.** *Assume $f$ is **A11**-smooth with constant $L_{\mathrm{rel}}$ relative to a strictly convex reference $h$. Mirror Descent updates*

$$x_{k+1} = \arg\min_z \left\{ \langle \nabla f(x_k), z \rangle + \frac{L_{\mathrm{rel}}}{\eta} D_h(z, x_k) \right\}$$

*satisfy*

$$f(x_{k+1}) \le f(x_k) - \frac{1}{\eta L_{\mathrm{rel}}} D_h(x_{k+1}, x_k).$$

(minimal: A11)

*Sketch (uses **A11**).* Relative smoothness gives $f(y) \le f(x) + \langle \nabla f(x), y - x \rangle + L_{\mathrm{rel}} D_h(y, x)$. The update minimises this bound with an added $1/\eta$ factor, producing the three-point inequality. Choosing $1/\eta \ge L_{\mathrm{rel}}$ secures descent. $\qquad\square$

### D.3 Regime III: Interpolation and Scale Invariance

**Theorem 46** (T42: Linear Convergence under Interpolation)**.** *Assume A2 (smoothness) and A4 (PL). If stochastic gradients satisfy **A12** (strong growth),*

$$\mathbb{E}[\|\nabla f_i(x)\|^2] \le \rho \|\nabla f(x)\|^2 \quad \forall x,$$

*then constant-stepsize SGD with $\eta \le 1/L$ obeys*

$$\mathbb{E}[f(x_k) - f^\star] \le (1 - c\eta\mu)^k [f(x_0) - f^\star]$$

*for some $c \in (0, 1)$ depending only on $L$ and $\rho$. The noise floor vanishes.* (minimal: A2+A4+A12)

*Sketch (uses **A2, A4, A12**).* Smooth descent gives $\mathbb{E}[f(x_{k+1})] \le f(x_k) - \eta\|\nabla f(x_k)\|^2 + \frac{L\eta^2}{2}\mathbb{E}[\|g_k\|^2]$. With A12, $\mathbb{E}[\|g_k\|^2] \le \rho\|\nabla f(x_k)\|^2$. Choosing $\eta < 2/(L\rho)$ keeps the coefficient positive; PL converts gradient norm to a gap, yielding the factor $1 - c\eta\mu$. $\qquad\square$

**Theorem 47** (T43: Edge of Stability (Curvature Saturation))**.** *Assume **A14** (Lipschitz Hessian) and bounded iterates. For stepsizes $\eta > 2/L$ that break the quadratic stability rule, Gradient Descent can enter a bounded, non-monotone regime in which*

$$\lambda_{\max}(\nabla^2 f(x_k)) \approx \frac{2}{\eta}.$$

(minimal: A14)

*Sketch (uses **A14**).* For the quadratic $f(x) = \frac{\lambda}{2} x^2$ stability needs $|1 - \eta\lambda| < 1$, i.e. $\lambda < 2/\eta$. With a Lipschitz Hessian, curvature changes at most linearly with displacement. Overshooting high-curvature zones pushes the iterate into regions where $\lambda < 2/\eta$, producing bounded oscillations centred near curvature $2/\eta$. $\qquad\square$

**Theorem 48** (T44: Scale-Invariant Auto-Tuning). *If $f$ satisfies **A13** (scale invariance: $f(cw) = f(w)$ for all $c > 0$), then SGD with fixed stepsize $\eta$ on the weights $w_t$ equals SGD on the direction $\theta_t = w_t/\|w_t\|$ with adaptive stepsize*

$$\eta_{\mathrm{eff}}(t) = \frac{\eta}{\|w_t\|^2}.$$

*As $\|w_t\|$ grows, the effective rate decays.* (minimal: A13)

*Sketch (uses **A13**).* Write $w_t = r_t \theta_t$ with $r_t = \|w_t\|$ and $\|\theta_t\| = 1$. Scale invariance implies $f(w_t) = f(\theta_t)$ and $\langle \nabla f(w_t), w_t \rangle = 0$. Updating $w_{t+1} = w_t - \eta \nabla f(w_t)$ gives $\|w_{t+1}\|^2 = \|w_t\|^2 + \eta^2 \|\nabla f(w_t)\|^2$; thus $r_t$ grows. Projecting onto the sphere yields $\theta_{t+1} \approx \theta_t - \eta/r_t^2 \nabla_\theta f(\theta_t)$, so the effective stepsize is $\eta_{\mathrm{eff}}(t) = \eta/r_t^2$. $\qquad\square$

