# OpenReview forum: "An Axiomatic Atlas for Optimization"
_TMLR — Rejected by TMLR_

### Review · Reviewer_AzX8 · 2026-03-24

**Summary Of Contributions:**

The starting point of this paper could be summarized as: *there are too many different assumptions ruling the behavior of optimization algorithms and something needs to be done about it*. The authors remain quite vague about why this would be a problem.

Then the authors propose a compilation of existing convergence/complexity results (dubbed: the atlas), which allows them to observe that changing one assumption (say: removing smoothness, or strong convexity) changes drastically the theoretical bound. This observation leads to grandiloquent sentences such as *"The atlas exposes inclusion-minimal assumption sets that suffice for a desired outcome, and it delineates sharp frontiers where a single assumption change alters the attainable regime"*. As opposed to what is claimed, the set of hypotheses/axioms is quite redundant, and the authors claim to discover so through a nebulous low-rank approximation of the binary matrix encoding the assumptions. I want to emphasize that I love survey papers, but this is **not** a survey paper. For instance the appendix contains a long list of known results, with proofs, without a single citation. I am further convinced that the proofs there were straight copy/pasted from existing works.

From this atlas, the authors propose their framework : while running an algorithm, detect which assumption holds in view to tune the parameters properly (sections 5-6). I cannot stress how much those sections are a pain to read. It is just nonsensical. I am not even sure what the authors are trying to do there.

**Additional Comments:**

## More comments on the paper

- About the "problem" of a literature with too many assumptions

  - *"practitioners must choose among many algorithmic variants"* -> i do not really see why this is a problem. if a problem is composite yes of course you should use a proximal method ; if a problem writes as an intractable sum / integral of course you should use a stochastic/incremental method.

  - *"[practitioners] tune interacting hyperparameters with limited guidance about which assumptions actually matter"* -> the fact that theoretical results are not always useful to help tuning parameters in practice is a well-know issue. But what does this have to do with the fact that many assumptions are available? This is not explained.

- About proving bounds under different assumptions

  - *"we prove that the nine axioms are logically independent by giving separating counterexamples"* All this is well known, and the fact that the authors do not event try to properly cite previous works is a red enough flag for me.

  - *"we show a form of completeness: many standard first-order guarantees can be derived using only subsets of these axioms"* I think that what the authors mean here is: they provide guarantees with the smallest set possible of assumptions. Which is, by the way, already the case for most state-of-the -art literature. Do they provide something new? They do not say so, and I would say that they do not: I personally skimmed the appendix and found nothing which wasn't known already. Worse, some results are stated with assumptions which are *stronger than necessary* : for instance it is now well known that bounds for SGD can be derived without assuming bounded variance (axiom A7).

  - why is the composite structure an axiom at all remains a mystery to me

  - *"[PL] interacts differently with stochastic noise (A7): constant-step-size SGD contracts to a neighborhood rather than to the optimum"* are the authors discovering that strongly convex functions have a unique minimizer?

  - *"Under smoothness and convexity, quadratic growth is equivalent to the Polyak-Łojasiewicz inequality"* Better than that, the implication PL --> quadratic growth is always true. Therefore I do not understand why is PL part of the axioms, given that the authors are looking for minimal assumptions.

  - *"The second observation is that many conclusions admit more than one inclusion minimal proof route. Several linear convergence statements have both a strong convexity route and a PL route, and several results that rely on metric regularity admit alternative routes through error bounds, PL geometry, or quadratic growth."* Somehow the authors are discovering something which is well-known in the community since a decade

  - *"In the atlas, these alternatives matter because they describe distinct ways to reach the same behavioral regime,"* what does this even mean?

  - *"and they explain why different papers and different communities sometimes impose different conditions yet obtain essentially the same qualitative guarantee"* No. First of all, I would like to stress that this paper does not include such "explanation". Second, the fact that those geometric inequalities are equivalent is well-known and well-understood since the early 2010's (and was less known since the 90's at least)
  - Section A in the appendix contains a long list of (known) complexity results , whose proofs are deferred into section C. I am baffled to say that there is not a single mention of where those proofs were "inspired" from, nor an acknowledgement that those results were known. For instance Theorem 28 and its proof mimic **very** closely (notations, style of proof) what was done by Bolte and Attouch around 2010, which remain the standard for non convex proofs using KL inequality.

- About exploiting the "atlas"

  - *"the relevant question is whether an axiom holds on the region explored by the iterates, not necessarily on all of Rd."* this topic was extensively studied in the literature

  - Figure 2 is, in my opinion, symptomatic of the content of the paper. Looks shiny at first glance, but is completely void of any interest when looking closely. (what does even mean moving from SGD to SVRG, or GD to prox-GD?)

- Section 5 : I am completely clueless in regard to what is happening here.
  - At a high level, I feel like the authors want to track certain relevant quantities in view to assert, along iterations, what assumption is verified or not. What quantities? *"Normalized distances to curvature, nonsmooth, accelerated, and noise templates"* How are defined those quantities? I have no idea, the paper neither. More information is supposed to be provided in the appendix, which happens to be a nonsensical mess.
  - The paper suggests, without an ounce of clarity, that those quantities could be used along iterations to change the parameters of the method. *"(...) turns these distances into a binding ceiling decision using bootstrap resampling of [the set of assumptions]. A ceiling is declared binding only when it fits well and clearly dominates competing explanations"*
  - how exactly are those parameters modified? No idea.
  - Section 6 contains numerical experiments which are supposed to validate the overall approach, but I will preserve my sanity by not delving into it.

**Audience:**

No

**Audience Explanation:**

It is my general opinion that this paper is at best very poorly written, at worst AI-generated, and should be rejected without wasting anybody else's time.

**Claims And Evidence:**

No

**Claims Explanation:**

It is my general opinion that this paper is at best very poorly written, at worst AI-generated, and should be rejected without wasting anybody else's time. I refuse to spend too much time reviewing it, and will simply focus on the major issues this paper has.

**Requested Changes:**

I recommend rejection without any changes.

---

### Review · Reviewer_DeBG · 2026-04-05

**Summary Of Contributions:**

**Summary**

The authors present a set of assumptions that are indeed commonly used in convex and first order optimization. The proposed "atlas" is quite simple: it maps a subset of assumptions into "convergence". The work then does some analysis to demonstrate that theorem assumptions are sparse. The authors then propose to use some diagnostic information when optimizing in order to find a better optimizer, given by a prescribed subset of steps. An extensive collection of proofs and definitions are in the appendix.

A lot of the results given in the article are clear or known to those doing optimization, which is already a very wide literature, covered by many popular books. This work has potential as a concise survey-type article that detailed exactly what theorem one can use if one can assume certain assumptions. However, I believe that the authors intended this work in a different way.

**Strengths**
* Axioms given are reasonable, and the introduction is nicely written for researchers outside optimization to understand the principle behind proving convergence results.
* Might be useful as a quick reference once references are added.

**Weaknesses**

* What is a roofline? There appears to be not much discussion on what this is nor how it is used, and subsequent notation, e.g. "diagnostics", "control plane", "logged trace" are given as definitions without motivation.
* What counts as a convergence result in this work? Is it convergence in function value $f(x_k) \rightarrow x^* $, or convergence in iteration $x_k \rightarrow x^* $ ? What about in the stochastic case?
* (Critical) Section 4.3 presents some "practically motivated intersections that are absent from the core theorem set", which are then dismissed as "not absent from the literature, but not represented in our pipeline". A7 + A8 occurs in stochastic composite optimization, which a quick search revealed [Thm. 6, Pham20] to give a gradient convergence result. A5 + A8 is actually frequently used in the Plug-and-Play literature in imaging, giving convergence to a critical point of a nonconvex functional.
* Related: there are many instances where the authors essentially write "we do not mean this, but instead mean that". For example, end of Section 3, Section 4.1's first paragraph, Section 4.2's first and last paragraphs, discussion of Q2 in Section 5. Better writing could avoid this.
* (Critical) What is the purpose of Section 4.2? The low rank factorization is especially puzzling. The fact that many theorems use a small subset of assumptions is likely due to the desire for proofs to have a minimal number of assumptions for the desired result. Moreover, assumptions in proofs are usually discovered through the process of proving them, which aids this minimal construction.
* Experiments showing the usage of the atlas are not convincing. Main text does not adequately provide exposition for experiments in Section 6.1. My understanding of Section 6.3 is that when trying to find a better optimizer through these diagnostics, one of the options is "switch to Adam", which is accepted to be an empirically better optimizer than SGD for these tasks. However, Adam does not have convergence guarantees for training neural networks (that I am aware of), and thus should not fit into this atlas.
* Is there an explanation for the figures? There are lot of subfigures and I could not find an explanation or intuition in the surrounding text.
* (Critical) There are no references in the appendix, after checking at least up to page 30 and skimming through the rest. Please explain.

**Minor issues**

* A5 could mention "Kurdyka-Lojasiewicz". This is in the appendix but I am not sure why it is not included in the main text.

[Pham20] Pham, N. H., Nguyen, L. M., Phan, D. T., & Tran-Dinh, Q. (2020). ProxSARAH: An efficient algorithmic framework for stochastic composite nonconvex optimization. _Journal of Machine Learning Research_, _21_(110), 1-48.

**Additional Comments:**

Flag for academic integrity problems, namely lack of references in the appendix.

**Audience:**

No

**Audience Explanation:**

Survey of theorems and their assumptions are likely well-known by optimization experts. For non-experts, secondary references, e.g. for convergence results given a particular subset of given assumptions, are not readily accessible.

**Broader Impact Concerns:**

-

**Claims And Evidence:**

No

**Claims Explanation:**

Atlas is incomplete. Experiments are not sufficiently detailed in the main text, and the experimental setup in neural network training does not seem reasonable. Possible advantages of this setup as opposed to more common methods such as Bayesian hyperparameter tuning are also not provided. Thus, the atlas has not been demonstrated to be practical.

**Requested Changes:**

See weaknesses. A major rewrite is also necessary in order to properly provide exposition in the main text.

---

### Review · Reviewer_wW76 · 2026-04-17

**Summary Of Contributions:**

The paper:

1. Introduces an _Optimization Atlas_ comprising of 9 "axioms" intended to capture common assumptions in first-order optimization theory, together with a _dependency ledger_ listing a collection of theorem statements and the (inclusion minimal) list of assumptions required for obtaining that result. The main aim of this part of the paper is organizational: to map assumptions to behavior (eg: sublinear versus linear convergence), and to make explicit which assumptions are required for each result.

2. In the second part, it then proposes to use this atlas in the reverse direction, namely to go from observed behavior in a training trace back to a diagnosis of the mechanism limiting progress. This is motivated by hyperparameter tuning in machine learning applications.
The idea is that, from the tail of a run, one fits a small family of canonical rate templates, infers which regime appears to be active, and then uses that diagnosis to guide parameter search or method changes, for example by adjusting batch size, smoothing, regularization, or switching algorithms.

**Audience:**

Yes

**Audience Explanation:**

Yes, I think at least some individuals in the TMLR audience would be interested in the paper. The topic is clearly relevant to researchers working on optimization theory, first-order methods, and the broader question of how to make optimization guarantees more interpretable and usable in practice. The idea of organizing a scattered literature through a common axiomatic framework, and then attempting to connect that framework to optimizer diagnostics, is a reasonable one and addresses a problem that many readers will recognize.

In that sense, the paper is not lacking in relevance or potential interest. My concerns are mainly about the execution: I am not convinced that the atlas is the right abstraction, nor that the evidence supports the practical diagnostic claims. But the underlying question the paper is trying to address is certainly one that parts of the TMLR audience would care about.

**Claims And Evidence:**

No

**Claims Explanation:**

The requested changes below are primarily smaller clarifications and presentation fixes. [While addressing them would improve the paper, they would not resolve the more fundamental concerns in my review, and for that reason I do not currently see a path to publication.]

At a high level, the _Optimization Atlas_ is best viewed as a synthesis and repackaging of existing optimization theory, rather than as a contribution of new results. Given the vast literature on the subject, I do think that a systematic framework for organizing assumptions, rates, and algorithmic regimes could be valuable. My reservation is not with the goal, but with the particular formulation adopted here: I am not convinced that the proposed atlas provides a natural or robust way to structure this landscape.

Many of the alternative variants of the theorems do not seem to identify genuinely different frontiers. Rather, they appear to be different formulations of essentially the same condition. For example, Theorem 5 states the convex PL <=> QG equivalence with a factor-4 loss in the constants, and related equivalences between the error bound (p=2), PL, and QG are then used to generate the alternative forms of T7, T8, and T33. However, even without convexity, these equivalences are already known in the literature (e.g.: [1], [2] and references therein).

Also, some of the theorem statements also appear weaker than necessary, in the sense that they do not use the strongest known formulation of the underlying result. This is not a correctness issue, but it makes the atlas feel less sharp than advertised, especially given the paper's emphasis on identifying precise frontiers between regimes. The simplest example: the result showing monotone decrease for GD on $L$-smooth cost function is stated for $\eta \leq 1/L$, but the standard argument holds for $\eta \leq 2/L$.

I am also not convinced by the proposed workflow for using the atlas to guide optimization decisions in practice. The intended deep-learning applications lie precisely in regimes that are not captured by the current atlas. The special thing about modern neural network training is specifically that the training dynamics do not resemble the regimes present in classical first-order theory on which the atlas is built (the losses are usually non-convex, not $L$-smooth, and training is usually done with large step sizes, so the loss does not decrease monotonically).

More specifically, the provided CIFAR-10 study is not convincing. The proposed diagnosis workflow is inconclusive on CIFAR, yet the action stage still proceeds by evaluating two hand-picked interventions: switching to Adam and doubling the batch size. Both are standard choices that most practitioners would naturally try in such a setting, so it is unclear what additional value is provided by the atlas-based procedure. As a result, the experiment does not convincingly show that the method can identify non-obvious bottlenecks or lead to decisions beyond standard practice.



[1] Karimi, Nutini, Schmidt - Linear Convergence of Gradient and Proximal-Gradient Methods Under the Polyak-Lojasiewicz Condition (2020)

[2] Rebjock, Boumal - Fast convergence to non-isolated minima: four equivalent conditions for $C^2$ functions (2024)

**Requested Changes:**

The requested changes below are primarily smaller clarifications and presentation fixes. [While addressing them would improve the paper, they would not resolve the more fundamental concerns in my review, and for that reason I do not currently see a path to publication.]

1. Some references appear as “??” (on pages 15 and 20).

2. The theorem numbering appears to be inconsistent between Table 1 and the appendix. For example, in Table 1, T5 is marked with A2 and A3, so it seems intended to correspond to a result for $L$-smooth, $\mu$-strongly convex functions. However, Theorem 5 in the appendix is instead the equivalence between convex PL and quadratic growth. This makes the dependency ledger difficult to follow.

3. It is also unclear why some results are placed in their stated category. For example, T18 (Proposition 12) concerns gradient descent with Polyak steps on $L$-smooth functions, but does not appear to involve stochasticity, so it is not clear why it is listed under “C. Stochastic Optimization.”

4. The setup of the synthetic experiments is not sufficiently clear. In particular, it is difficult to understand what the underlying optimization task is.

5. The appendix contains proofs for the theorems from Table 1, but since these do not appear to be new proofs, the paper should provide theorem-level references to the existing literature. This is particularly important in a paper whose main contribution is the aggregation and organization of known results: clear attribution is needed so that readers can trace the origin of each statement, understand what is standard versus what is new, and assess whether the selected formulations are representative of the literature.

---

### Review · Reviewer_5Dcb · 2026-04-21

**Summary Of Contributions:**

The paper's main contribution can be summarized in two parts:

1. It tries to unify 35 first-order optimization theorems into a single, structured map. And by extracting 9 common axioms (such as smoothness, strong convexity, and bounded variance), it compresses a theoretical landscape into a finite assumption space. In short, it provides a summarized and structured view of the existing well-known theoretical results in first-order optimization.

2. Building on this foundation, the authors propose a diagnostic framework for practical model training. It tries to identify the bottleneck by analyzing the training process, and then evaluates and ranks the possible interventions by using a ROI criterion to guide how to adjust the training setups.

In my opinion, the core issue of this paper is that it sits in an awkward position between theory and practice. On the theory side, this atlas can indeed give a more clear view, but it does not give new insights beyond the existing well-known results. On the practice side, while the idea I think is somehow interesting, but the empirical validation is far from sufficient to support its effectiveness. Both relevant baselines such as random search and Bayesian optimization which are very basic, and modern benchmark tasks / models, are not considered by the authors in the evaluation.

Also, the writing seems a bit rough for me. The paper introduces a number of non-standard terms and stylistic expressions such as "roofline tag" without providing clear definitions and explanations in this specific scenario. This makes the paper more like it wants to use these seemingly fancy words, which often have conventional meaning, to impress readers. Also, the presentation of figures and tables is poorly integrated with the text. Please revise.

I think The paper has the potential to be a practically oriented contribution. However, this would require expanding the taxonomy of interventions and validating the framework on more complex, real-world training scenarios. In particular, the possible change should at least be at the following two directions:

1. Comprehensively analyze how modern training techniques influence the underlying axiomatic properties and form a systematic library of interventions and propose mechanism to adaptively select the most suitable intervention.

2. Replace toy examples with evaluations on real-world tasks to better demonstrate practical applicability.

**Audience:**

Yes

**Audience Explanation:**

Despite its limitations, the paper remains relevant for two main reasons. First, the proposed atlas offers a comprehensive and structured view of the theoretical landscape of first-order optimization. It could be helpful for researchers and practitioners who are not quite familiar. Second, it focuses on a practical problem to adaptively adjust and deploy hyperparameters during training in response to specific optimization bottlenecks. I think this is a meaningful and useful direction, as improving the efficiency and robustness of model training remains a central concern in modern machine learning, especially for large-scale models nowadays. Consequently, the problem setting aligns well with the interests of the TMLR community.

**Broader Impact Concerns:**

I have not found any discussions about the limitations and potential negative societal impact. But in my opinion, this may not be a problem, since the work only focuses on the optimization. Still, it is highly encouraged to add corresponding discussions.

**Claims And Evidence:**

No

**Claims Explanation:**

1. This atlas mainly works as a summary of existing results rather than introducing new theory. It organizes 35 standard first-order optimization theorems into a shared set of nine axioms, but this is more of a reorganization of well-known insights than a real advancement.
2. The current set of interventions seems somewhat basic, as such changes are often not needed in practice based on my experience. The paper should include more common engineering tricks (e.g., gradient clipping or learning rate warmup) and explain how these methods relate to the proposed axioms.
3. The empirical validation is insuffient as the framework is mainly tested on toy problems and shows inconclusive results on CIFAR-10. Moreover, without comparisons to standard methods such as Bayesian optimization or random search.

**Requested Changes:**

See above sections. Overall, while the conceptual direction is meaningful, the current paper is not yet mature. The core framework requires fundamental refinements, so a major reorganization and rewriting is necessary.

---

### Decision · Action_Editor_2CG3 · 2026-05-11

**Recommendation:** Reject

**Additional Comments:**

The authors did not react to any of the reviews or submit a revision during the rebuttal phase.

**Audience:**

No

**Audience Explanation:**

Even though the field of convex optimization is central to TMLR, this paper does not contribute significant empirical evidence, and its repackaging of existing theoretical results is flawed.

**Claims And Evidence:**

No

**Claims Explanation:**

As the reviewers all pointed out, the atlas of theorems presented here consists of known and important results, but these are often missing appropriate citations, or presented with overly restrictive hypotheses or conclusions. Thus, it cannot serve as a proper survey paper.
The empirical value of the atlas is also in doubt: its experimental validation is limited, especially since modern deep learning explores regimes where the conclusions of standard convex optimization are no longer relevant.